# Alternative platelet differentiation pathways initiated by nonhierarchically related hematopoietic stem cells

Joana Carrelha [1,2,3,14] ✉, Stefania Mazzi[4,14], Axel Winroth [4,14], Michael Hagemann-Jensen [5], Christoph Ziegenhain [5,6], Kari Högstrand[4], Masafumi Seki [4], Margs S. Brennan[4], Madeleine Lehander[4], Bishan Wu[1,2], Yiran Meng [2], Ellen Markljung[5], Ruggiero Norfo[1,2,7], Hisashi Ishida [4], Karin Belander Strålin [4,8], Francesca Grasso[4], Christina Simoglou Karali[1,2], Affaf Aliouat[1,2], Amy Hillen [5], Edwin Chari[4], Kimberly Siletti[9,10], Supat Thongjuea[11], Adam J. Mead [1,2,12], Sten Linnarsson [9], Claus Nerlov [2], Rickard Sandberg [5], Tetsuichi Yoshizato [4], Petter S. Woll [4] & Sten Eirik W. Jacobsen [1,2,4,5,13] ✉

Rare multipotent stem cells replenish millions of blood cells per second through a time-consuming process, passing through multiple stages of increasingly lineage-restricted progenitors. Although insults to the blood-forming system highlight the need for more rapid blood replenishment from stem cells, established models of hematopoiesis implicate only one mandatory differentiation pathway for each blood cell lineage. Here, we establish a nonhierarchical relationship between distinct stem cells that replenish all blood cell lineages and stem cells that replenish almost exclusively platelets, a lineage essential for hemostasis and with important roles in both the innate and adaptive immune systems. These distinct stem cells use cellularly, molecularly and functionally separate pathways for the replenishment of molecularly distinct megakaryocyte-restricted progenitors: a slower steady-state multipotent pathway and a fast-track emergency-activated platelet-restricted pathway. These findings provide a framework for enhancing platelet replenishment in settings in which slow recovery of platelets remains a major clinical challenge.

The blood system represents a paradigm for how multiple mature cell lineages in an adult tissue are replenished from self-renewing multipotent stem cells[1]. Almost 90% of >100 billion cells replaced daily in humans are blood cells[2], predominantly short-lived platelets, granulocytes and erythrocytes but also lymphoid lineages. In a steady state, the need for replenishment can be fulfilled through the amplification of multiple stages of increasingly lineage-restricted and highly proliferative progenitors until they become fully restricted to one blood cell lineage[1].

Common to all established models of the adult hematopoietic stem and progenitor cell (HSPC) hierarchy is the implication of only one mandatory differentiation pathway from hematopoietic stem cells (HSCs) to each short-lived mature blood cell lineage[1,3], translating into considerable time for an HSC to replenish fully lineage-restricted progenitors. While not representing a problem in steady-state hematopoiesis (as each progenitor stage is also continuously replenished), it could pose a challenge if intermediate progenitors are acutely eliminated in physiological settings or in response to therapeutic insults to

**Fig. 1 | Distinct platelet replenishment kinetics from single HSCs. a**, HSPC hierarchy reconstituted by single LSK*Gata1*-eGFP⁻CD34⁻CD150⁺CD48⁻ *Vwf*-tdTomato⁻ multi-HSCs (*n* = 8) or *Vwf*-tdTomato⁺ P-HSCs (*n* = 9). The numbers shown are the mean ± s.e.m. percentage contributions to each population. Orange, reconstitution in all mice. Pink, reconstitution in some mice (the fraction of reconstituted mice is indicated in the upper left of each circle); mean of positive mice. Only progenitor populations present in ≥1/3 of mice and with ≥0.1% average reconstitution are shown. For the full P-HSC hierarchy, see Extended Data Fig. 1d. Phenotypic definitions: LT-HSC, LSKFLT3⁻CD150⁺CD48⁻CD45.2⁺; ST-HSC, LSKFLT3⁻CD150⁻CD48⁻CD45.2⁺; MPP2, LSKFLT3⁻CD150⁺CD48⁺CD45.2⁺; MPP3, LSKFLT3⁻CD150⁻CD48⁺CD45.2⁺; MPP4, LSKFLT3⁺CD45.2⁺; MkP, LKCD150⁺CD41⁺CD45.2⁺; preMegE progenitor, LKCD41⁻CD16/32⁻CD150⁺CD105⁻CD45.2⁺; colony-forming unit-erythroid (CFU-E), LKCD41⁻CD16/32⁻CD150⁻CD105⁺*Gata1*-eGFP⁺; pregranulocyte–monocyte (preGM) progenitor, LKCD41⁻CD16/32⁻CD150⁻CD105⁻CD45.2⁺; GMP, LKCD

41⁻CD16/32⁺CD150⁻CD105⁻CD45.2⁺; platelets (P), CD150⁺CD41⁺TER119⁻*Vwf*-tdTomato⁺*Gata1*-eGFP⁺ for *Vwf*-tdTomato^tg/+ *Gata1*-eGFP^tg/+ donors; erythrocytes (E), TER119⁺CD150⁻CD41⁻*Vwf*-tdTomato⁻*Gata1*-eGFP⁺; myeloid (granulocyte and monocyte) cells (M), CD11b⁺NK1.1⁻CD19⁻CD4/CD8a⁻CD45.1⁻CD45.2⁺; donor-derived B lymphocytes (B), CD19⁺NK1.1⁻CD4/CD8a⁻CD11b⁻CD45.1⁻CD45.2⁺; donor-derived T lymphocytes (T), CD4/CD8a⁺NK1.1⁻CD11b⁻CD19⁻CD45.1⁻CD45.2⁺. **b**, HSCs and MPPs replenished by *Vwf*-tdTomato⁻ multi-HSCs (*n* = 8) or *Vwf*-tdTomato⁺ P-HSCs (*n* = 9). Representative profiles and mean ± s.e.m. percentages of the parent LSKCD45.2⁺ gate are shown. **c**, Granulocyte/monocyte (GM) and megakaryocyte (MK) in vitro lineage potentials (mean ± s.e.m.) of HSCs and MPPs replenished by *Vwf*-tdTomato⁻ multi-HSCs (*n* = 3). Data are from 580–720 plated wells per population with a similar distribution across three replicates. Each dot represents an independent experiment.

the bone marrow (BM), resulting in transient but critical reductions in short-lived platelets and granulocytes, which can lead to considerable morbidity, hospitalization and transfusion burden[4]. In addition to hemostasis and thrombosis[5], platelets have important roles in immune responses[6,7]. Substantial efforts have been made toward enhancing platelet replenishment following therapeutic and physiological challenges, but success has been limited[4].

Recently, platelet-biased and platelet-restricted HSCs (P-HSCs) were identified[8–10], and fast-track pathways for the replenishment of megakaryocyte-restricted progenitors (MkPs) have been implicated[11–14]. However, as relying on phenotypic or molecular (rather than functional) definitions of HSCs, it remains unclear whether the proposed accelerated pathways are initiated from true HSCs or downstream progenitor cells.

Although single-cell transplantations and steady-state lineage tracing have established HSC heterogeneity, there is evidence only for hierarchical relationships between HSCs with different lineage biases[8,15,16], implicating shared rather than separate pathways for blood lineage replenishment. Herein, we pursued multiple functional and molecular single-cell approaches, combined with genetic lineage tracing, to investigate whether HSCs with distinct lineage biases and restrictions have nonhierarchical relationships and/or use distinct platelet progenitor pathways.

## Results

### Nonhierarchically related distinct platelet-replenishing HSCs

We previously showed that a large fraction of phenotypic (lineage (LIN)⁻SCA1⁺cKIT⁺ (LSK)CD34⁻CD150⁺CD48⁻) HSCs express *von Willebrand factor* (*Vwf*) and are transcriptionally platelet primed; upon transplantation, *Vwf*⁺ HSCs replenish all blood lineages but in a platelet-biased manner[8]. Through a kinetic analysis of blood lineage replenishment in >1,000 mice transplanted with a single adult BM LSKCD34⁻CD150⁺CD48⁻ HSC, we identified *Vwf*-tdTomato⁺ long-term HSCs (LT-HSCs) that do not contribute to B or T lymphocytes and are platelet–erythroid–myeloid restricted, replenishing blood in a platelet-biased manner (platelet contribution ≥3-fold higher than erythroid and myeloid contributions); a smaller fraction of these HSCs are platelet-restricted, replenishing exclusively platelets in primary recipients[9]. However, platelet-restricted HSCs can also replenish low levels of erythroid and myeloid (granulocyte and monocyte) cells when transplanted into secondary recipients[9]. Nevertheless, they remain highly platelet-biased and fail to contribute to B and T lymphoid lineages[9], suggesting that platelet-bias is a stable and HSC-intrinsic property. In contrast, most single *Vwf*-tdTomato⁻ HSCs replenish all lympho-myeloid lineages upon transplantation (multilineage HSCs (multi-HSCs)), typically in a lineage-balanced or lymphoid-biased manner[9].

In the present study, *Vwf*-tdTomato[+] P-HSCs were defined as single HSCs that, upon transplantation, stably (at multiple analysis time points) contribute ≥50-fold more to platelets than to erythrocytes and myeloid cells, with little or no (≤0.01%) B and T lymphocyte contributions. Single *Vwf*-tdTomato[+] and *Vwf*-tdTomato[−] LSKCD34[−]CD150[+]CD48[−] cells were purified from the adult BM of CD45.2 *Vwf*-tdTomato/*Gata1*-eGFP mice[9] and transplanted into irradiated CD45.1 recipient mice (Extended Data Fig. 1a). Because LT-HSCs have been shown not to express *Gata1*-eGFP[17] but express the endothelial protein C receptor (EPCR/CD201)[18], we used *Gata1*-eGFP[−] and, in some instances, also CD201[+] gating to enhance HSC purity among LSKCD34[−]CD150[+]CD48[−] cells. As HSCs can be identified only through their functional properties[18], single P-HSCs and multi-HSCs were eventually identified through stable and distinct blood lineage output, as defined above.

We hypothesized that if alternative differentiation pathways for platelet replenishment exist downstream of HSCs, they should be used differently by *Vwf*-tdTomato[+] P-HSCs and *Vwf*-tdTomato[−] multi-HSCs replenishing blood cell lineages in a balanced or lymphoid-biased pattern (Extended Data Fig. 1b,c).

Flow cytometry analysis demonstrated that single *Vwf*-tdTomato[−] multi-HSCs robustly replenished all commonly defined phenotypic HSPC compartments[19–22] (Fig. 1a). In contrast, single *Vwf*-tdTomato[+] P-HSCs consistently reconstituted only LSK FMS-like tyrosine kinase 3 (FLT3)[−]CD150[+]CD48[−] LT-HSCs and LIN[−]SCA1[−]cKIT[+] (LK) CD150[+]CD41[+] MkPs, and, in a few instances, also LSKFLT3[−]CD150[+]CD48[+] multipotent progenitor 2 (MPP2) and LKCD150[+]CD41[−]CD105[−]CD16/32[−] premegakaryocyte−erythroid (preMegE) progenitors (Fig. 1a and Extended Data Fig. 1d). Three established subsets of LSK progenitors (LSKFLT3[−]CD150[−]CD48[+] MPP3, LSKFLT3[+] MPP4 and LSKFLT3[−]CD150[−]CD48[−] short-term HSCs (ST-HSCs))[21,22] were consistently and robustly replenished by *Vwf*-tdTomato[−] multi-HSCs but never by P-HSCs (Fig. 1a,b). Functional in vitro single-cell clonal analysis demonstrated that all MPP subsets replenished by *Vwf*-tdTomato[−] multi-HSCs possessed megakaryocyte potential (Fig. 1c and Extended Data Fig. 1e), suggesting that *Vwf*-tdTomato[−] multi-HSCs, at least in part, might use a different pathway with more progenitor intermediates for platelet replenishment compared to *Vwf*-tdTomato[+] P-HSCs.

We assessed whether a hierarchical relationship might exist between *Vwf*-tdTomato[−] multi-HSCs and *Vwf*-tdTomato[+] P-HSCs (Fig. 2a). The LSKCD150[+]CD48[−] phenotypic HSC compartment replenished in the BM of recipients of a single transplanted *Vwf*-tdTomato[−] multi-HSC remained exclusively or predominantly *Vwf*-tdTomato[−] (Fig. 2b). In contrast, when replenished by a single *Vwf*-tdTomato[+] P-HSC, the generated LSKCD150[+]CD48[−] cells remained predominantly *Vwf*-tdTomato[+], although often containing a fraction of *Vwf*-tdTomato[−] cells (Fig. 2b). As the HSC identity cannot be reliably defined by phenotype alone[18], we next performed gold-standard secondary long-term reconstitution experiments with *Vwf*-tdTomato[−] and *Vwf*-tdTomato[+] cells purified from the BM of primary recipients reconstituted by a single *Vwf*-tdTomato[−] multi-HSC or *Vwf*-tdTomato[+] P-HSC (Fig. 2a,b). In 11 primary recipients, single *Vwf*-tdTomato[−] multi-HSCs replenished *Vwf*-tdTomato[−] LSKCD150[+]CD48[−] cells capable of balanced or lymphoid-biased multilineage contribution in secondary recipients but not *Vwf*-tdTomato[+] LSKCD150[+]CD48[−] cells with secondary long-term contribution to platelets; therefore, no P-HSCs were produced (Fig. 2c,d and Supplementary Tables 1 and 2).

Although it is difficult to envision how P-HSCs could replenish multi-HSCs, given that replenishment remains stably platelet-biased and with little or no lymphoid contribution even upon secondary transplantation[9], we also performed secondary transplantations with purified *Vwf*-tdTomato[+] and *Vwf*-tdTomato[−] LSKCD150[+]CD48[−] cells replenished in primary recipients by single *Vwf*-tdTomato[+] P-HSCs. Single *Vwf*-tdTomato[+] P-HSCs replenished *Vwf*-tdTomato[+] LSKCD150[+]CD48[−] cells in primary recipients, which, upon secondary transplantation, replenished blood in a platelet-biased manner without detectable lymphoid output; in contrast, *Vwf*-tdTomato[−] LSKCD150[+]CD48[−] cells did not provide secondary long-term myelo-lymphoid reconstitution and, therefore, contained no multi-HSCs (Fig. 2e and Supplementary Table 3). In one unique case, in which a single *Vwf*-tdTomato[+] P-HSC also produced a sizable fraction of *Vwf*-tdTomato[−] LSKCD150[+]CD48[−] cells, both the *Vwf*-tdTomato[+] and *Vwf*-tdTomato[−] cells were capable of long-term secondary blood replenishment but exclusively in a platelet-biased manner without any lymphoid output (Extended Data Fig. 1f and Supplementary Table 3), confirming that P-HSCs are unable to replenish multi-HSCs. Together with *Vwf*-tdTomato[−] multi-HSCs being incapable of producing *Vwf*-tdTomato[+] P-HSCs, these findings demonstrate that *Vwf*-tdTomato[−] multi-HSCs and *Vwf*-tdTomato[+] P-HSCs are not hierarchically related and, therefore, should replenish platelets through different pathways.

**Molecularly distinct megakaryocyte replenishment pathways**

While the above experiments established that *Vwf*-tdTomato[−] multi-HSCs and *Vwf*-tdTomato[+] P-HSCs are nonhierarchically related and might use distinct progenitor pathways for platelet replenishment, they also suggested that these cells, in part, might pass through shared progenitor stages (Fig. 1a). To compare the cellular trajectories in a more unbiased and in-depth manner, we performed single-cell whole-transcriptome (Smart-seq3) analysis[23] of HSPCs

**Fig. 2 | Nonhierarchical relationship between distinct HSCs. a**, Experimental outline of hierarchical HSC transplantations. Further details of cell phenotypes and numbers are provided in Supplementary Tables 1–3. **b**, Top, percentage of *Vwf*-tdTomato[+] cells within LSKCD150[+]CD48[−]CD45.2[+] cells replenished in the BM of CD45.1 primary (1°) recipients transplanted with a single CD45.2 LSK*Gata1*-eGFP[−]CD34[−]CD150[+]CD48[−] *Vwf*-tdTomato[−] multi-HSC (*n* = 33) or *Vwf*-tdTomato[+] P-HSC (*n* = 17). Dots represent individual mice, and lines represent mean ± s.e.m. Bottom, representative flow cytometry profiles. **c**, Left, representative histogram (mean ± s.e.m., *n* = 4) of *Vwf*-tdTomato expression in LSKCD150[+]CD48[−]CD45.2[+] cells replenished in primary CD45.1 recipients by a single CD45.2 *Vwf*-tdTomato[−] multi-HSC. Middle, primary blood reconstitution at 16–37 weeks (wk) and secondary (2°) reconstitution at 16–18 weeks after transplantation of *Vwf*-tdTomato[−] LSKCD150[+]CD48[−]CD45.2[+] cells sorted from primary recipients (mean ± s.e.m., *n* = 4 from four experiments; in secondary recipients, each dot represents the average reconstitution of one to two mice per primary recipient). Right, interpretation of results regarding (non)hierarchical replenishment of multi-HSCs and P-HSCs. Further details of cell phenotypes and numbers are provided in Supplementary Table 1. **d**, Left, representative histogram (mean ± s.e.m., *n* = 7) of *Vwf*-tdTomato expression

in LSKCD150[+]CD48[−]CD45.2[+] cells replenished in primary recipients of a single transplanted LSK*Gata1*-eGFP[−]CD34[−]CD150[+]CD48[−] *Vwf*-tdTomato[−] multi-HSC. Middle, primary and secondary blood reconstitution 16–25 and 16–22 weeks after transplantation, respectively (mean ± s.e.m., *n* = 7 primary recipients from five experiments; in secondary recipients, each dot represents the average reconstitution of one to three mice per primary recipient). Right, interpretation of results regarding (non)hierarchical replenishment of multi-HSCs and P-HSCs. Further details of cell phenotypes and numbers are provided in Supplementary Table 2. **e**, Left, representative histogram (mean ± s.e.m., *n* = 3) of *Vwf*-tdTomato expression in LSKCD150[+]CD48[−]CD45.2[+] cells replenished in primary recipients of a single transplanted LSK*Gata1*-eGFP[−]CD34[−]CD150[+]CD48[−] *Vwf*-tdTomato[+] P-HSC. Middle, primary and secondary blood reconstitution 16–37 and 16–18 weeks after transplantation, respectively (mean ± s.e.m., *n* = 3 primary recipients from three experiments; in secondary recipients, each dot represents the average reconstitution of one to two mice per primary recipient). Right, interpretation of results regarding (non)hierarchical replenishment of multi-HSCs and P-HSCs. Further details of cell phenotypes and numbers are provided in Supplementary Table 3.

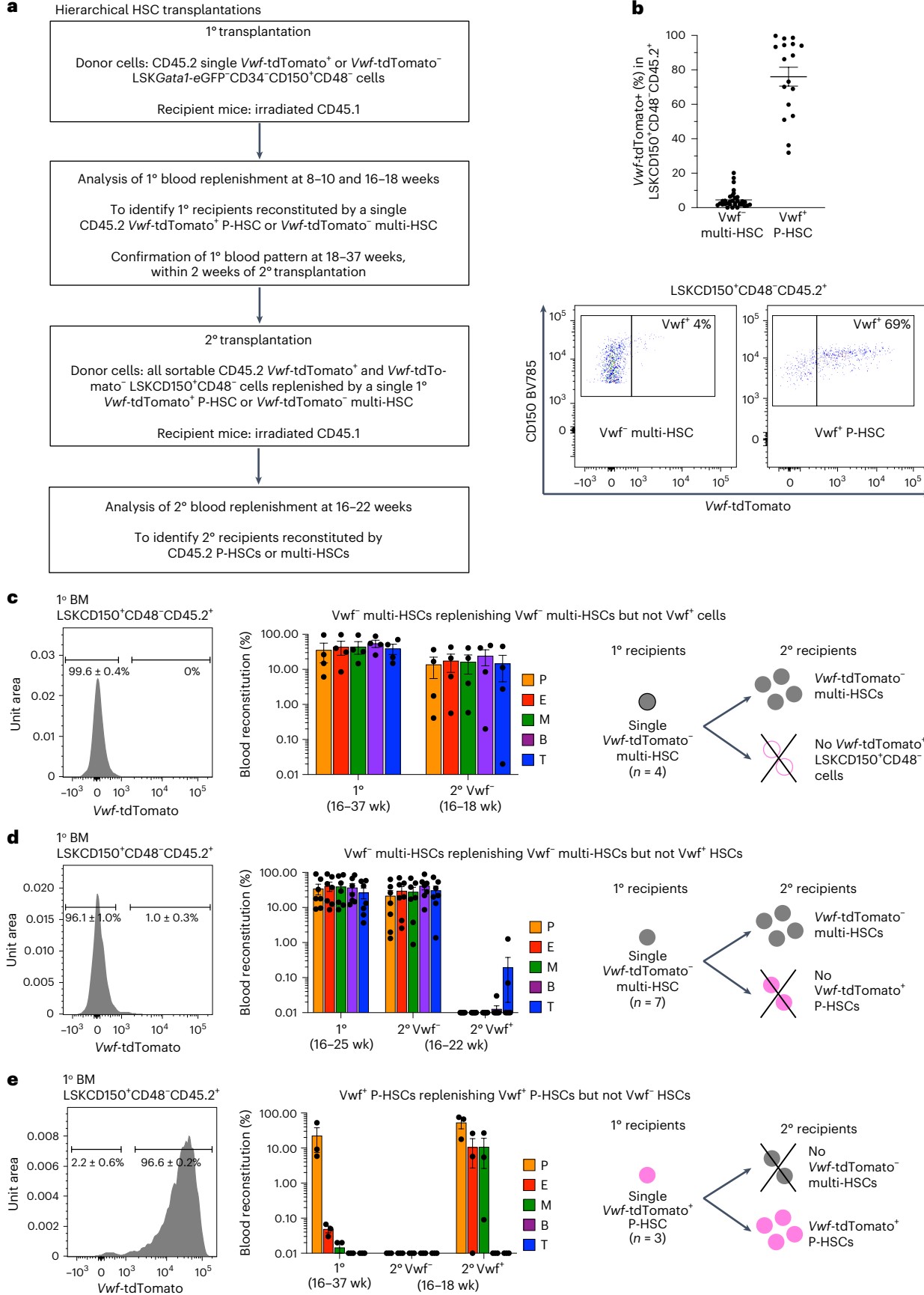

long-term replenished by single *Vwf*-tdTomato[+] P-HSCs (2,290 cells from seven reconstituted mice) and *Vwf*-tdTomato[−] multi-HSCs (2,478 cells from eight reconstituted mice) (Fig. 3a and Supplementary Fig. 1a). Single cells replenished by *Vwf*-tdTomato[+] P-HSCs and *Vwf*-tdTomato[−] multi-HSCs displayed similar quality control metrics (Supplementary Fig. 1b,c). After adjustment for batch effects using the mutual nearest-neighbor approach[24] (Supplementary Fig. 1d–g), we performed dimensional reduction using uniform manifold approximation and projection (UMAP) based on the 2,000 most variable genes (Fig. 3b and Supplementary Fig. 2a–c). The multiple replicate recipients of a single *Vwf*-tdTomato[+] P-HSC or *Vwf*-tdTomato[−] multi-HSC showed consistently distinct contributions to different HSPC compartments (Fig. 3b and Extended Data Fig. 2), with minimal overlap within shared phenotypically or molecularly defined HSC and progenitor compartments (Fig. 3b and Extended Data Figs. 2, 3a,b and 4a,b), using different ranges of highly variable genes (HVGs) (Extended Data Fig. 4c) and following dimensional reduction by *t*-distributed stochastic neighbor embedding (tSNE) (Extended Data Fig. 4d). This suggests that *Vwf*-tdTomato[+] P-HSCs and *Vwf*-tdTomato[−] multi-HSCs replenish molecularly distinct progenitor pathways for platelets.

In agreement with the strong and stable platelet-biased replenishment, no bipotent cells with shared expression of both MkP and erythroid gene signatures were detected in MkPs or any other HSPCs replenished by *Vwf*-tdTomato[+] P-HSCs (Fig. 3c,d). In contrast, and in agreement with previous studies[19], apparent bipotent preMegE progenitors with combined megakaryocyte and erythroid but no myeloid or lymphoid gene expression were replenished by *Vwf*-tdTomato[−] multi-HSCs (Fig. 3e).

After removing committed erythroid and myeloid progenitors replenished by *Vwf*-tdTomato[−] multi-HSCs, we identified cells with high area under the curve (AUC) scores for HSC ('molecular overlap population' (MolO))[25] or MkP signatures[19] (Methods), as expected at the start and end of pseudotime, respectively (Fig. 3f and Supplementary Fig. 2d,e). The MolO score decreased along pseudotime for cells replenished by *Vwf*-tdTomato[+] P-HSCs and *Vwf*-tdTomato[−] multi-HSCs, and this was also observed for the AUC score for HSCs with a low contribution to mature blood lineages and megakaryocyte bias[26] (Extended Data Fig. 5a,b). In contrast, the AUC scores for signatures enriched in multilineage HSCs and HSCs with high lineage output[26] followed the same pattern as the megakaryocyte signature (Extended Data Fig. 5a,b), increasing along pseudotime for both *Vwf*-tdTomato[+] P-HSCs and *Vwf*-tdTomato[−] multi-HSCs, compatible with also capturing progenitors. Although no major differences were observed when comparing the enrichment of published HSC signatures[21,25–29] in

replenished HSCs, *Vwf*-tdTomato[+] P-HSCs showed significantly higher AUC scores for functional HSC signatures (MolO and serial engrafter) and for restricted/biased lineage output (low output and megakaryocyte bias) (Extended Data Fig. 5c). MkP gene signatures[19] were also enriched in HSCs originating from *Vwf*-tdTomato[+] P-HSCs, whereas the granulocyte–monocyte progenitor (GMP) signature was higher in HSCs replenished from *Vwf*-tdTomato[−] multi-HSCs (Extended Data Fig. 5d).

We identified 375 differentially expressed genes (DEGs; absolute log₂(fold change) > 0.5, adjusted $P < 0.05$) within MolO-defined HSCs replenished by *Vwf*-tdTomato[−] multi-HSCs and *Vwf*-tdTomato[+] P-HSCs (Fig. 3g and Supplementary Table 4). Mammalian target of rapamycin complex 1 (mTORC1) signaling was among the pathways most enriched in MolO HSCs replenished by *Vwf*-tdTomato[+] P-HSCs (Fig. 3g,h), including *Fads1* (ref. 30), *Fads2* (ref. 31), *Ldha* (ref. 32) and *Nupr1* (ref. 33) (Fig. 3i and Supplementary Table 4), a pathway critical for HSC quiescence and self-renewal[34]. Other pathways associated with mTORC1 signaling[35] were also enriched, including interferon response and MYC targets (Fig. 3h).

TradeSeq analysis[36] established that 217 of 11,989 genes detected in at least 10% of all cells were differentially expressed between the two pathways along pseudotime (adjusted $P < 0.01$ and tradeSeq median absolute log₂(fold change) > 1) (Fig. 3j and Supplementary Table 4). Pearson correlation analysis of the 70 top-ranked DEGs along pseudotime showed a more similar gene expression profile for cells located at trajectory start (HSCs) and end (MkPs) than those at intermediate stages, whereas 70 randomly selected genes demonstrated consistent and very high correlation (Fig. 3k and Supplementary Table 4). Taken together, DEGs along pseudotime define the separation of the two pathways from HSCs to MkPs.

Genes encoding markers previously assigned to distinct HSPC stages were among the top-ranked DEGs (Fig. 3g), including *Vwf*, *Flt3* (encoding the receptor FLT3 expressed on MPP subsets, including those with little or no megakaryocyte potential)[37] and *Cd48* (encoding cell-surface CD48, suggested to define distinct MkP subsets)[14], each showing distinct separation along pseudotime (Fig. 3l). Differential RNA expression of *Flt3* and *Cd48* correlated closely with the corresponding protein expression (Figs. 1b and 3m and Extended Data Fig. 3a).

In further agreement with the replenishment of distinct MkPs, 345 DEGs (absolute log₂(fold change) > 0.5, false discovery rate (FDR)-adjusted $P < 0.05$) were identified when comparing molecularly defined MkPs replenished from *Vwf*-tdTomato[+] P-HSCs (P-MkPs) and *Vwf*-tdTomato[−] multi-HSCs (multi-MkPs) (Fig. 4a, Extended Data Fig. 6a and Supplementary Table 4) but, importantly, not in the gene

**Fig. 3 | Distinct molecular platelet differentiation pathways. a**, Left, experimental design (partly created with Biorender.com) for single-cell RNA sequencing of HSPCs generated by single *Vwf*-tdTomato[+] P-HSCs ($n = 7$) or *Vwf*-tdTomato[−] multi-HSCs ($n = 8$). Right, mean (dots indicate individual mice) contribution to blood lineages. **b**, UMAP of LIN[−]cKIT[+] cells replenished by single *Vwf*-tdTomato[+] P-HSCs (blue; $n = 7$ mice, 2,290 cells) or *Vwf*-tdTomato[−] multi-HSCs (red; $n = 8$ mice, 2,478 cells). HSC, GMP, MkP and CFU-E cells were classified based on molecular signatures (Extended Data Fig. 3b). **c–e**, AUC heatmaps for lineage signatures in single MkPs (**c**; $n = 133$ cells) and other HSPCs (**d**; $n = 2,157$ cells) replenished by single *Vwf*-tdTomato[+] P-HSCs (seven mice) and in preMegE progenitors with an MkP and/or erythroid AUC score of >0.1 (**e**; $n = 212$ cells) replenished by single *Vwf*-tdTomato[−] multi-HSCs (eight mice). Red rectangle, preMegE progenitors derived from *Vwf*-tdTomato[−] multi-HSCs with combined MkP−erythroid signatures without myeloid and lymphoid signatures. **f**, UMAP after removing erythroid- and myeloid-restricted progenitors, visualized by donor type, molecular HSC (MolO > 0.22), molecular MkP (AUC > 0.25) and pseudotime order. **g**, DEGs (red; adjusted $P < 0.05$, absolute log₂(fold change) > 0.5) when comparing MolO HSCs replenished by *Vwf*-tdTomato[+] P-HSCs ($n = 1,047$ cells) and *Vwf*-tdTomato[−] multi-HSCs ($n = 97$ cells). **h**, Gene-set enrichment normalized enrichment scores (NES; false discovery rate (FDR) *q*

value < 0.1) of HALLMARK pathways based on DEGs detected in **g**. **i**, Expression (log₂) of DEGs (adjusted $P < 0.05$, combined Wilcoxon/Fisher's exact test) related to mTORC1 signaling when comparing MolO HSCs derived from *Vwf*-tdTomato[+] P-HSCs ($n = 1,047$ cells, seven mice) and *Vwf*-tdTomato[−] multi-HSCs (97 cells, eight mice). Boxes, first and third quartiles; line, median; whiskers, ±1.5× interquartile range; dots, outlier cells. The percentages of cells with detected gene expression (Methods) are shown. **j**, Fold-change (log₂) tradeSeq fitted expression values of the top 70 DEGs (adjusted $P < 0.01$, log₂(fold change) > 1, patternTest tradeSeq function) along pseudotime when comparing cells replenished by *Vwf*-tdTomato[+] P-HSCs and *Vwf*-tdTomato[−] multi-HSCs. **k**, Pearson correlation (center line) along pseudotime comparing the expression of the top 70 DEGs and 70 randomly selected non-DEGs between cells replenished by *Vwf*-tdTomato[+] P-HSCs and *Vwf*-tdTomato[−] multi-HSCs. Shading indicates the 95% confidence interval (CI). **l**, Normalized gene expression along pseudotime for cells shown in **f**. Lines show the mean expression count from the generalized additive model fit using tradeSeq. **m**, FLT3 expression in LSKCD45.2[+] cells generated by single *Vwf*-tdTomato[−] multi-HSCs ($n = 8$) and *Vwf*-tdTomato[+] P-HSCs ($n = 9$). Representative profiles with mean ± s.e.m. percentages of parent gates and representative histograms are shown.

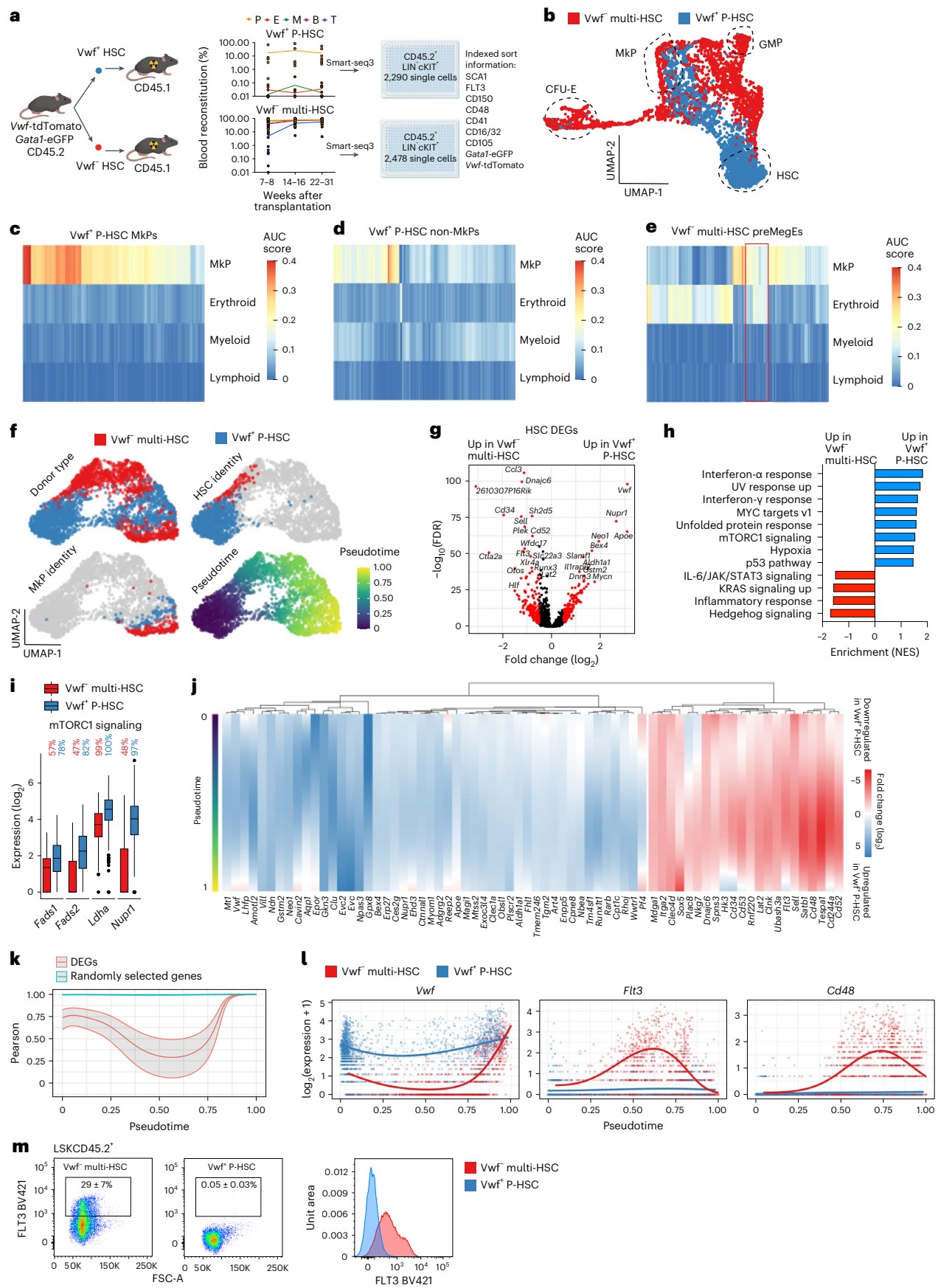

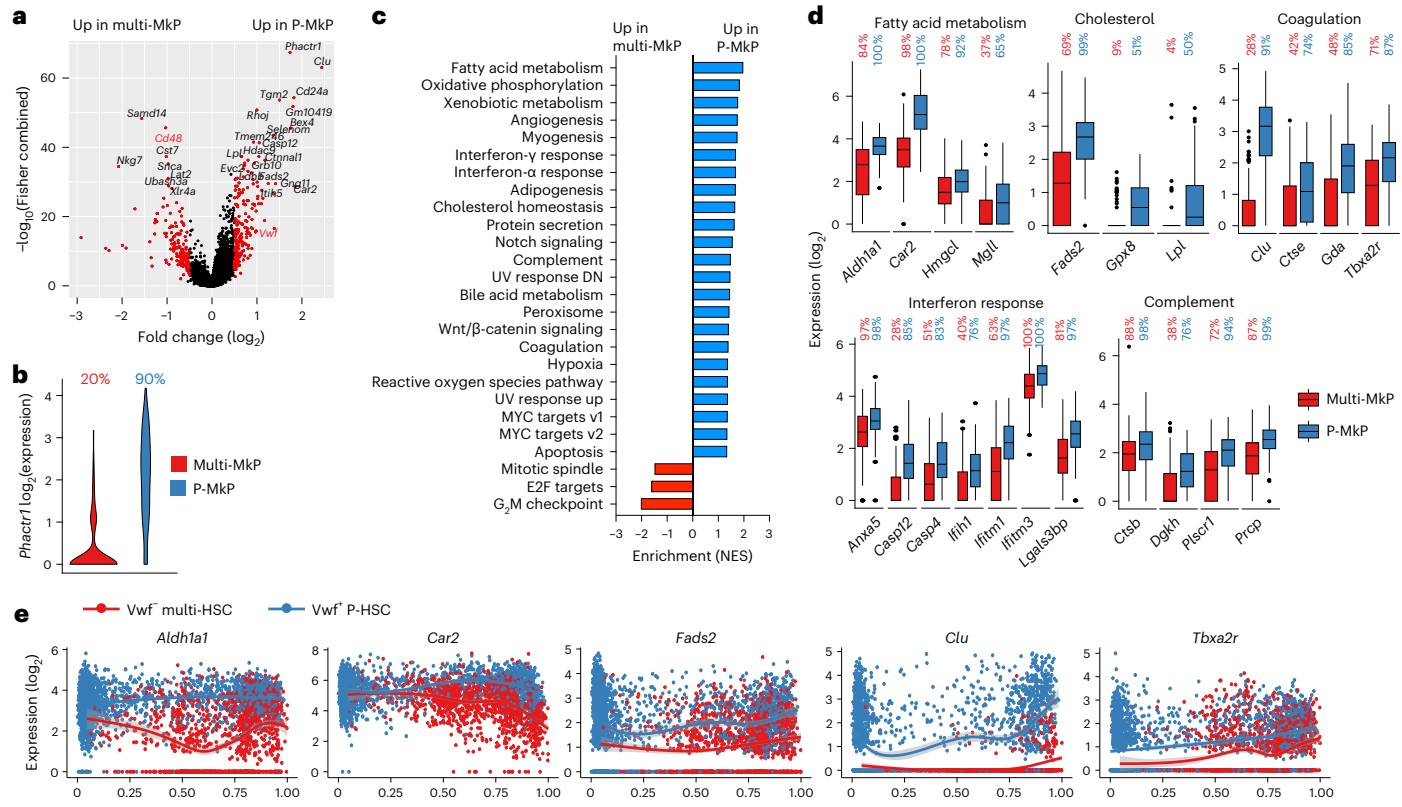

**Fig. 4 | Transcriptional characterization of MkPs replenished by single *Vwf*-tdTomato⁻ multi-HSCs and *Vwf*-tdTomato⁺ P-HSCs. a**, DEGs (red; adjusted *P* < 0.05 and absolute log₂(fold change) > 0.5, combined Wilcoxon/Fisher's exact test) when comparing molecular MkPs replenished by single *Vwf*-tdTomato⁻ multi-HSCs (multi-MkPs; *n* = 177) or *Vwf*-tdTomato⁺ P-HSCs (P-MkPs; *n* = 119). *Cd48* and *Vwf* are highlighted in red. **b**, Expression (log₂) of *Phactr1*, the top DEG (*P* < 0.05, log₂(fold change) > 0.5, combined Wilcoxon/Fisher's exact test) when comparing multi-MkPs and P-MkPs. The percentages of cells with detected expression (Methods) are indicated above the violin plots. **c**, Normalized gene-set enrichment score for HALLMARK pathways of DEGs enriched (FDR *q* value < 0.1) in multi-MkPs (red) and P-MkPs (blue). **d**, Expression (log₂) of DEGs (adjusted *P* < 0.05, combined Wilcoxon/Fisher's exact test) associated with fatty acid metabolism, cholesterol homeostasis, coagulation, interferon response and complement when comparing multi-MkPs (red; *n* = 177 cells, eight mice) and P-MkPs (blue; *n* = 119 cells, seven mice). Boxes, first and third quartiles; line, median; whiskers, the largest values within the ±1.5× interquartile range; dots, outliers. The percentages of cells with detected gene expression (Methods) are shown above the boxes. **e**, Expression (log₂) of selected genes in all LIN⁻cKIT⁺ single cells generated by single *Vwf*-tdTomato⁻ multi-HSCs or *Vwf*-tdTomato⁺ P-HSCs along pseudotime. Dots represent individual cells, and lines represent LOESS (locally estimated scatterplot smoothing) curves of the expression for the HSC subtype (gray shading indicates the 95% CI).

expression signature defining MkPs (Extended Data Fig. 5d). One of the most differentially expressed MkP genes, *Phactr1* (Fig. 4b), has been linked to increased platelet numbers[38] and risk for early-onset cardio-vascular thrombosis[39]. A significant enrichment of genes involved in fatty acid metabolism and cholesterol homeostasis was observed in P-MkPs (Fig. 4c,d and Supplementary Table 4), of relevance for the reported association between hypercholesterolemia and platelet homeostasis[40], as well as for the disruption of cholesterol efflux activating MkPs[41]. P-MkPs were also enriched for coagulation genes, including *Tbxa2r* (encoding thromboxane 2, promoting platelet activation, and targeted by acetylsalicylic acid to prevent platelet overactivation[42]), and genes associated with complement (Fig. 4c,d and Supplementary Table 4), including prolycarboxypeptidase (*Prcp*; promoting enhanced coagulation through plasma prekallikrein[43]). P-MkPs also showed upregulated expression of interferon-α and -γ response genes (Fig. 4c,d and Supplementary Table 4), including interferon-induced transmembrane protein genes (*Ifitm1* and *Ifitm3*) promoting immune-mediated platelet activation[44], and inflammation-induced genes linked to the regulation of platelet activity (*Lgals3bp* (ref. 45) and *Ifih1* (ref. 46)). As for HSCs (Fig. 3h), genes associated with hypoxia and MYC targets were enriched in P-MkPs (Fig. 4c). Multiple genes assigned to fatty acid metabolism (*Aldh1a1*, *Car2*), cholesterol homeostasis (*Clu*, *Fads2*) and coagulation (*Vwf*, *Clu*, *Tbxa2r*) were more highly expressed along

pseudotime in the *Vwf*-tdTomato⁺ P-HSC trajectory before MkP generation (Fig. 4e), suggesting that some of the differences observed in MkPs replenished from the two pathways are already programmed at the HSC stage. Notably, the expression of the coagulation-related gene *Clu* was exclusive to the *Vwf*-tdTomato⁺ P-HSC pathway (Fig. 4e).

To identify DEGs facilitating future identification and enrichment of MkPs distinct for the two differentiation pathways, genes encoding cell-surface antigens were further explored (Fig. 4a and Supplementary Table 4). A significant upregulation of *Cd24a* and *Vwf* (driving the expression of *Vwf*-tdTomato) was observed in molecularly defined P-MkPs, whereas *Cd48* and *Itga2* (encoding CD49b) were upregulated in multi-MkPs and negative in almost all P-MkPs (Fig. 5a). Flow cytometric index information confirmed the differential expression of the CD48 protein and *Vwf*-tdTomato in MkPs (Extended Data Fig. 6b), further validated in separate experiments together with CD24 and CD49b expression. In agreement with the transcriptional data, the expression of *Vwf*-tdTomato and cell-surface CD24 was distinctly upregulated in P-MkPs, whereas CD48 and CD49b were expressed in most multi-MkPs but virtually absent from P-MkPs (Fig. 5b,c).

Our findings agree with previous studies suggesting that CD48 expression might define a distinct subset of MkPs[12,13]. We found P-MkPs to be uniformly CD48ⁿᵉᵍ⁻ˡᵒ at the transcriptional and protein levels, but a fraction of multi-MkPs were also negative for *Cd48* and CD48 (Fig. 5

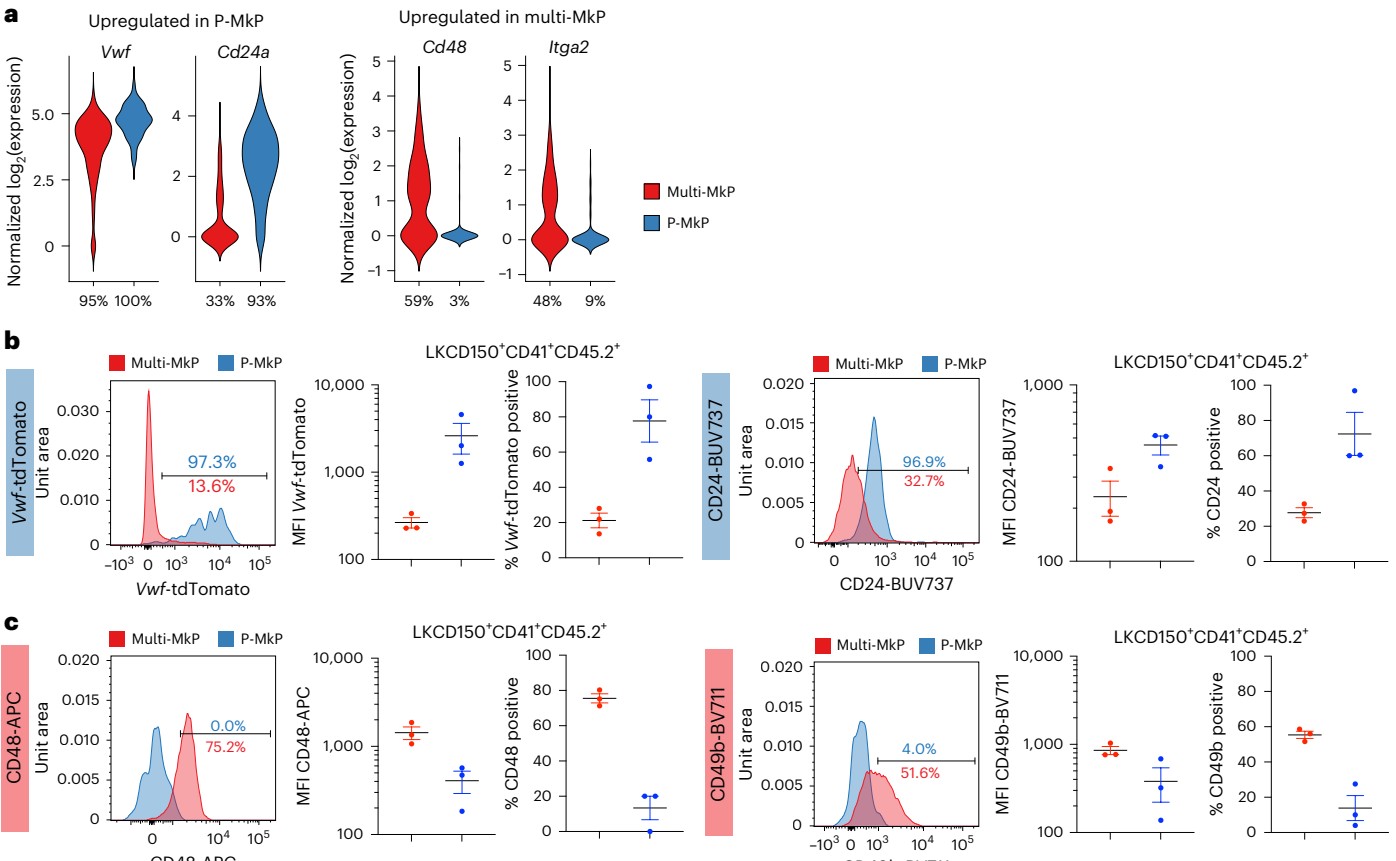

**Fig. 5 | Phenotypic characterization of MkPs replenished by single transplanted *Vwf*-tdTomato⁻ multi-HSCs and *Vwf*-tdTomato⁺ P-HSCs. a,** Normalized expression (log₂) of the indicated DEGs (adjusted *P* < 0.05, combined Wilcoxon/Fisher's exact test) encoding *Vwf* and specified cell-surface proteins (*Itga2* encodes CD49b) in single molecularly defined multi-MkPs (*n* = 177) or P-MkPs (*n* = 119). The percentage of cells expressing each gene (Methods) is indicated below each violin plot. **b,c,** Histograms (left), mean fluorescence intensity (MFI; middle) and percentage positive cells (right) based on flow

cytometry analysis of the expression of *Vwf*-tdTomato reporter and CD24 (**b**; corresponding gene expression upregulated in P-MkPs) and of CD48 and CD49b (**c**; corresponding gene expression upregulated in multi-MkPs) in LKCD150⁺CD41⁺CD45.2⁺ MkPs replenished by single *Vwf*-tdTomato⁻ multi-HSCs (*n* = 3) or *Vwf*-tdTomato⁺ P-HSCs (*n* = 3). Histograms (percentage positive cells) show the expression on gated MkPs replenished by a platelet-restricted *Vwf*-tdTomato⁺ P-HSC and a *Vwf*-tdTomato⁻ multi-HSC. Dots represent individual mice, and lines represent mean ± s.e.m.

and Extended Data Fig. 6). While compatible with multi-HSCs partly replenishing CD48⁻ MkPs overlapping with CD48⁻ P-MkPs, this was not the case, as CD48⁺ and CD48⁻ multi-MkPs showed highly overlapping DEGs when individually compared to CD48⁻ P-MkPs, including for *Cd24a*, *Itga2* and *Vwf* (Fig. 6a,b and Supplementary Table 4). The same pattern of DEGs was observed when the comparison was based on *Cd48* mRNA expression (Fig. 6c and Supplementary Table 4), whereas very few DEGs were detected when comparing multi-MkPs negative or positive for *Cd48* (Fig. 6d). Thus, while *Cd48*/CD48 expression specifically identifies multi-MkPs, *Cd48*/CD48⁻ multi-MkPs are also molecularly distinct from P-MkPs.

Taken together, single-cell RNA-sequencing analyses of HSPCs replenished by single transplanted *Vwf*-tdTomato⁺ P-HSCs and *Vwf*-tdTomato⁻ multi-HSCs unravel molecularly distinct progenitor differentiation trajectories for platelet replenishment, including transcriptionally and phenotypically distinct MkPs.

## Usage of alternative platelet replenishment pathways

One of the most striking differences revealed by single-cell RNA sequencing was the virtual absence of *Flt3* RNA expression in the entire pathway initiated by *Vwf*-tdTomato⁺ P-HSCs, contrasting with the high *Flt3* expression in the *Vwf*-tdTomato⁻ multi-HSC pathway from the earliest stages of differentiation (Fig. 3l,m). Previous *Flt-3*Cre fate-mapping studies demonstrated that replenishment of all

blood lineages, including platelets, occurs through *Flt3*-expressing stages in steady-state hematopoiesis[47]. Although we confirmed erythrocytes, granulocytes, and B and T lymphocytes to be almost 100% *Flt3*Cre-tdTomato⁺ in steady-state *Flt3*Cre^tg/+^ *R26*^Tom/+^ mice, a fraction (10%) of platelets were consistently *Flt3*Cre-tdTomato⁻ (Fig. 7a and Extended Data Fig. 7a,b), compatible with steady-state platelets, unlike other blood lineages, being partly produced through an *Flt3*⁻ pathway, initiated by P-HSCs. To test this possibility directly, we investigated to what degree single *Vwf*-eGFP⁻ multi-HSCs and platelet-restricted *Vwf*-eGFP⁺ P-HSCs from *Flt3*Cre^tg/+^ *R26*^Tom/+^ *Vwf*-eGFP^tg/+^ *Gata1*-eGFP^tg/+^ mice replenish *Flt3*Cre-tdTomato⁺ and *Flt3*Cre-tdTomato⁻ platelets after transplantation. In agreement with usage of an *Flt3*⁺ pathway, platelets and other blood cell lineages long-term replenished by single *Vwf*-eGFP⁻ multi-HSCs were almost entirely *Flt3*Cre-tdTomato⁺, whereas *Vwf*-eGFP⁺ platelet-restricted P-HSCs replenished almost exclusively *Flt3*Cre-tdTomato⁻ platelets (Fig. 7b and Extended Data Fig. 7c–e). This provided further support for a nonhierarchical relationship between *Vwf*-eGFP⁻ multi-HSCs and *Vwf*-eGFP⁺ P-HSCs and a strict separation between *Vwf*-eGFP⁻ multi-HSC and *Vwf*-eGFP⁺ P-HSC platelet replenishment pathways, as also supported by *Vwf*-eGFP⁻ multi-HSCs but not *Vwf*-eGFP⁺ P-HSCs replenishing FLT3⁺ *Flt3*Cre-tdTomato⁺ BM MPPs (Extended Data Fig. 7f).

Our present and previous findings[9] are compatible with *Vwf*-tdTomato⁺ P-HSCs replenishing MkPs through fewer progenitor

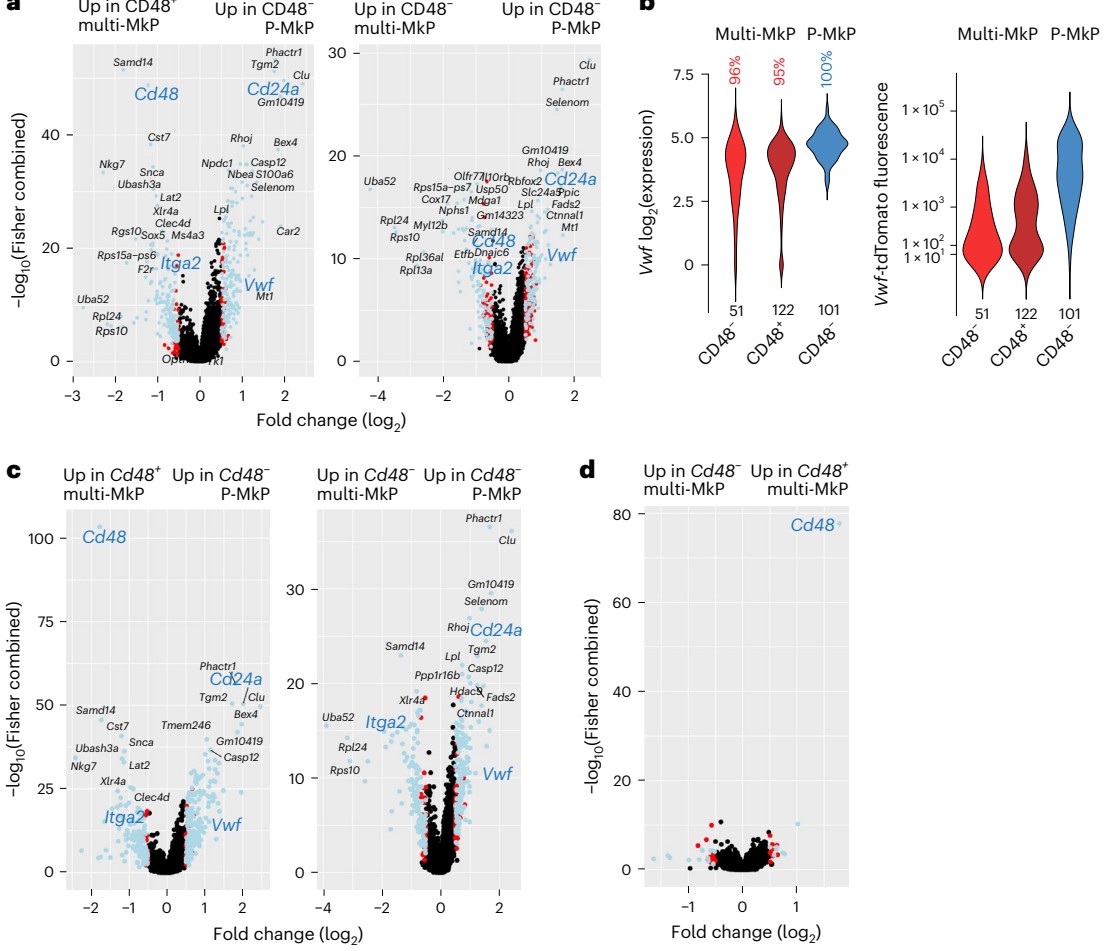

**Fig. 6 | Replenishment of molecularly distinct MkPs by *Vwf*-tdTomato⁻ multi-HSCs and *Vwf*-tdTomato⁺ P-HSCs. a**, DEGs (adjusted *P* < 0.05, combined Wilcoxon/Fisher's exact test, absolute log₂(fold change) > 0.5; blue, genes overlapping with the DEGs comparing total P-MkPs and total multi-MkPs in Fig. 4a; red, genes not overlapping with the DEGs in Fig. 4a) when comparing CD48⁺ (left; *n* = 122 cells) or CD48⁻ (right; *n* = 51 cells) molecularly defined MkPs replenished by single *Vwf*-tdTomato⁻ multi-HSCs (multi-MkPs) to CD48⁻ molecularly defined MkPs replenished by single *Vwf*-tdTomato⁺ P-HSCs (P-MkPs; *n* = 101 cells). *Cd48*, *Vwf*, *Cd24a* and *Itga2* are highlighted in blue. **b**, Distribution of *Vwf* log₂(mRNA expression) and percentage of *Vwf* transcript-positive cells (left) and *Vwf*-tdTomato reporter fluorescence distribution (right) in CD48⁻ and CD48⁺ molecularly defined multi-MkPs and CD48⁻ P-MkPs. The numbers of analyzed single cells are indicated below the violin plots. **c**, DEGs

(adjusted *P* < 0.05, combined Wilcoxon/Fisher's exact test, absolute log₂(fold change) > 0.5; blue, genes overlapping with the DEGs comparing total P-MkPs and total multi-MkPs in Fig. 4a; red, genes not overlapping with the DEGs in Fig. 4a) when comparing *Cd48* transcript-positive (left) or *Cd48* transcript-negative (right) multi-MkPs (*n* = 102 and 71 cells, respectively) to *Cd48* transcript-negative P-MkPs (*n* = 114 cells). *Cd48*, *Vwf*, *Cd24a* and *Itga2* are highlighted in blue. **d**, DEGs (adjusted *P* < 0.05, combined Wilcoxon/Fisher's exact test, absolute log₂(fold change) > 0.5; blue, genes overlapping with the DEGs comparing total P-MkPs and total multi-MkPs in Fig. 4a; red, genes not overlapping with the DEGs in Fig. 4a) when comparing *Cd48* transcript-positive (*n* = 102 cells) to *Cd48* transcript-negative (*n* = 71 cells) multi-MkPs. A detailed list of detected DEGs is provided in Supplementary Table 4.

intermediates than *Vwf*-tdTomato⁻ multi-HSCs. In agreement with this, *Gata1*⁺ progenitors produced from transplanted *Vwf*⁺ P-HSCs replenish platelets with faster kinetics than *Gata1*⁺ progenitors from *Vwf*⁺ multi-HSCs[48]. To investigate whether this might also translate into faster steady-state kinetics of MkP replenishment through the P-HSC than the multi-HSC progenitor pathway, we explored published single-cell RNA-sequencing data, in which the kinetics of progenitor replenishment were assessed after recombination induction in *Hoxb-5*Cre^ERT2/+ *R26*^Tom/+ reporter mice, specifically labeling the HSC compartment[49]. Interestingly, a subset of MkPs were the first lineage-restricted progenitors replenished by labeled HSCs[49]. Compared to MkPs replenished later, this early wave of MkPs showed an upregulation of genes also upregulated in P-MkPs and a downregulation of genes upregulated in multi-MkPs (Supplementary Fig. 3). Collectively, these findings raise the possibility that, upon insults to the hematopoietic system resulting in loss of MkPs, usage of the P-HSC pathway might more rapidly

replenish platelets than multi-HSCs. Thus, we treated *Flt3*Cre^tg/+ *R26*^Tom/+ mice with cyclophosphamide (CP), a cytotoxic agent that efficiently reduces megakaryocytes and MkPs[4,50]. We observed a small reduction in platelets and a more striking reduction in BM MkPs after CP treatment, accompanied by a clear decrease in the fraction of *Flt3*Cre-tdTomato⁺ platelets (from 95% before treatment to 82% at 7 days and 80% at 18 days after CP), followed by a return toward steady-state levels at 45 days; in contrast, *Flt3*Cre-tdTomato⁺ labeling of other lineages was unaffected at any time point (Fig. 8a and Extended Data Fig. 8a–c). This suggests that the *Flt3*⁻ P-HSC pathway might have an important and lineage-specific role in accelerated platelet replenishment after CP treatment. LIN⁻cKIT⁺FLT3⁺ BM progenitors remained close to 100% *Flt3*Cre-tdTomato⁺ after CP treatment (Extended Data Fig. 8d), suggesting that the decreased labeling of blood platelets is not a consequence of decreased *Flt3*Cre recombination in FLT3⁺ BM progenitors. To exclude unspecific effects of CP on Cre recombination, we treated

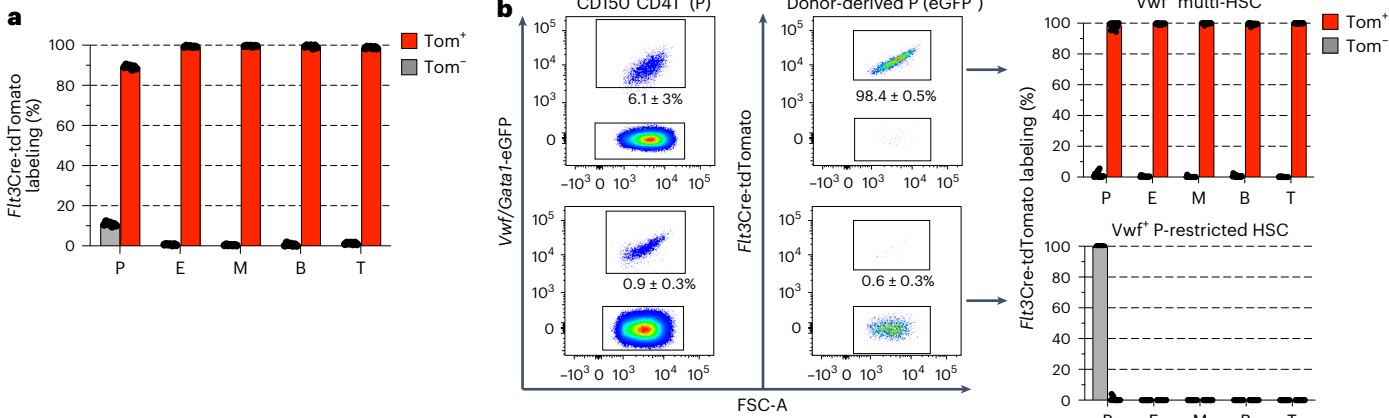

**Fig. 7 | Fate mapping of alternative platelet replenishment pathways.**
**a**, *Flt3*Cre-tdTomato labeling (mean ± s.e.m.) of blood lineages in steady-state
*Flt3*Cre[tg/+] *R26*[Tom/+] *Vwf*-eGFP[tg/+] *Gata1*-eGFP[tg/+] mice (n = 7; 8–13 weeks old):
platelets (CD150[+]CD41[+]TER119[−]), erythrocytes (TER119[+]CD150[−]CD41[−]), myeloid
(granulocyte) cells (CD11b[+]GR1[+]CD41[−]NK1.1[−]CD19[−]CD4/CD8a[−]), B lymphocytes
(CD19[+]CD41[−]NK1.1[−]CD4/CD8a[−]CD11b[−]GR1[−]) and T lymphocytes (CD4/CD8a[+]C

D41[−]NK1.1[−]CD11b[−]GR1[−]CD19[−]). Dots represent individual mice. Tom, tdTomato.
**b**, Representative profiles of *Flt3*Cre-tdTomato labeling in platelets (left) and
mean ± s.e.m. labeling of all lineages (right) replenished by a single CD45.2 *Vwf*-
eGFP[−] multi-HSC (top; n = 14) and *Vwf*-eGFP[+] platelet-restricted P-HSC (bottom;
n = 14) 18–21 weeks after transplantation.

*Vav*Cre[tg/+] *R26*[Tom/+] mice, in which Cre expression is under the control of
the pan-hematopoietic *Vav* promoter[51], resulting in all hematopoietic
cells in the BM and blood being completely labeled. In these mice, plate-
lets (and all other lineages) remained almost 100% *Vav*Cre-tdTomato[+]
following CP treatment (Extended Data Fig. 9). The nucleic acid-binding
fluorescent dye thiazole orange (TO) labels enriched RNA content of
newly generated reticulated platelets[52]. In a steady state, the majority
(>90%) of TO[+] platelets were *Flt3*Cre-tdTomato[+] and, as previously
reported[53], only a minority of platelets were newly generated TO[+]
platelets. Following the CP challenge, the *Flt3*Cre-tdTomato[−] fraction
of TO[+] platelets increased. Moreover, on day 4 after CP injection, TO[+]
reticulated platelets represented a significantly larger fraction of
*Flt3*Cre-tdTomato[−] platelets than *Flt3*Cre-tdTomato[+] platelets (Fig. 8b,c
and Extended Data Fig. 10a–c).

We next administered 5-fluorouracil (5FU), another myeloa-
blative agent shown to reduce MkPs rapidly[54]. In line with this, we
observed a transient decrease in platelets and a significant reduction
in *Flt3*Cre-tdTomato[+] platelets (from 95% to 58%) on day 10 after 5FU
treatment (Fig. 8d), whereas FLT3[+] progenitors in the BM were 100%
*Flt3*Cre-tdTomato[+] (Extended Data Fig. 10d,e). This was followed by
a significant rebound to 78% tdTomato[+] platelets on day 24 follow-
ing 5FU treatment (Fig. 8d). Notably, we also observed a smaller yet
significant decrease in *Flt3*Cre-tdTomato[+] fractions of erythrocytes
and myeloid cells (but not lymphocytes; Fig. 8d), probably reflect-
ing that P-HSCs can also replenish lower levels of erythrocytes and
myeloid cells.

Finally, we tested a challenge that specifically depletes platelets
rather than progenitors, as, following such a challenge, rapid plate-
let replenishment would probably be primarily accomplished from
existing progenitors (from both pathways) rather than HSCs; conse-
quently, the contribution by the two pathways could be expected to
be largely unaltered. We induced acute platelet depletion by admin-
istering an anti-CD42b antibody to *Flt3*Cre[tg/+] *R26*[Tom/+] mice. As previ-
ously reported[8], acute thrombocytopenia was observed 3 days after
anti-CD42b treatment (Fig. 8e), with no impact on other lineages
(Extended Data Fig. 10f) or the balance between *Flt3*Cre-tdTomato[+]
and *Flt3*Cre-tdTomato[−] platelets (Fig. 8f). Unlike the loss of MkPs in
response to CP treatment, an expansion of MkPs was observed 3 days
after platelet depletion (Extended Data Fig. 10g), suggesting that
a rapid expansion of MkPs might underlie the subsequent platelet
recovery.

Together, these findings suggest that a rapid and transient increase
in platelet replenishment can be achieved through the P-HSC pathway
in response to challenges that reduce progenitors of the megakaryo-
cyte lineage.

## Discussion

While previous studies provided evidence only for hierarchical kinships
between HSCs with different lineage biases[8,15,16], we here establish a
nonhierarchical relationship between *Vwf*-tdTomato[−] HSCs stably
replenishing all myeloid and lymphoid blood cell lineages without a
platelet bias and *Vwf*-tdTomato[+] P-HSCs replenishing only, or almost
exclusively, platelets. Moreover, in contrast to established hierarchical
models of hematopoiesis implicating only one mandatory differentia-
tion pathway from HSCs for each lineage, we uncovered two distinct
pathways for platelet replenishment. Rather than representing alterna-
tive differentiation pathways from the same HSC, these two pathways
are initiated by distinct HSCs. Previously established signature scores
for multi-HSCs and high-output HSCs[26] increased along the pseudotime
differentiation trajectory from HSCs toward MkPs, suggesting that
these signatures also capture progenitor cells, in line with the reported
deficient blood contribution upon secondary transplantation of HSCs
marked by these signatures[26]. While the exact roles of *Vwf*-tdTomato[−]
multi-HSCs and *Vwf*-tdTomato[+] P-HSCs in sustaining platelet homeo-
stasis remain unclear, phenotypic and single-cell RNA-sequencing
analyses demonstrated that *Vwf*-tdTomato[+] P-HSCs bypass several
stages of MPPs used by *Vwf*-tdTomato[−] multi-HSCs. Moreover, sin-
gle HSC transplantations showed that *Vwf*-eGFP[+] P-HSCs, unlike
*Vwf*-eGFP[−] multi-HSCs, generate platelets without passing through
*Flt3*-expressing progenitor stages, further corroborating the existence
of alternative platelet replenishment pathways from HSCs with distinct
lineage biases. Although, as previously shown[47], all other blood line-
ages were almost exclusively produced through an *Flt3*[+] pathway in the
steady state, a significant fraction of platelets had not passed through
*Flt3*-expressing progenitors. Upon suppression of megakaryopoiesis
in *Flt3*Cre[tg/+] *R26*[Tom/+] mice in response to CP treatment, we observed
a rapid and transient increase in platelets not having passed through
the *Flt3*[+] pathway, whereas all cells of other blood lineages remained
fully labeled, suggesting enhanced replenishment of platelets through
the *Flt3*[−] pathway from P-HSCs. A similar enhanced usage of the *Flt3*[−]
pathway was observed with 5FU, but with a smaller contribution also
to myeloid and erythroid cells. This aligns with most P-HSCs, although

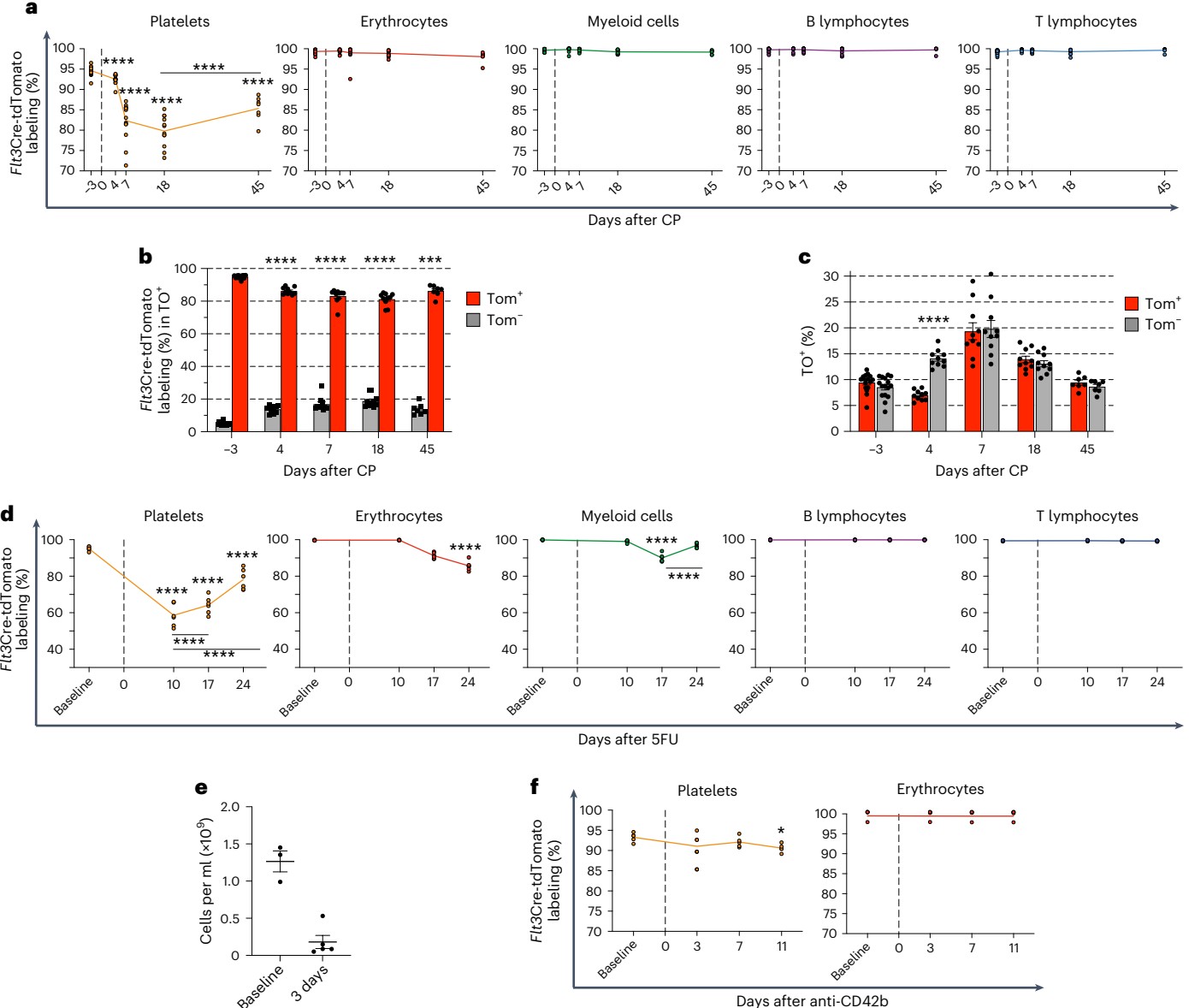

**Fig. 8 | Fate mapping of alternative platelet replenishment pathways upon hematopoietic challenges. a**, *Flt3*Cre-tdTomato labeling of blood lineages (as in Fig. 7a) in 7- to 11-week-old *Flt3*Cre[tg/+] *R26*[Tom/+] mice after CP treatment on day 0. Analysis at baseline (day −3; $n = 22$) and on day 4 ($n = 14$), day 7 ($n = 14$), day 18 ($n = 12$) and day 45 ($n = 7$). Lines connect the mean of each time point. In platelets, ****$P = 3.23 \times 10^{-6}$ for day 4, ****$P = 7.67 \times 10^{-86}$ for day 7, ****$P = 7.19 \times 10^{-99}$ for day 18 and ****$P = 6.88 \times 10^{-45}$ for day 45 compared to baseline; ****$P = 6.18 \times 10^{-17}$ between days 18 and 45. Linear mixed-model two-sided analysis with $P$-value adjustment by the Benjamini−Hochberg procedure. **b**, *Flt3*Cre-tdTomato labeling (mean ± s.e.m.) of TO[+] cells in 7- to 11-week-old *Flt3*Cre[tg/+] *R26*[Tom/+] mice after CP treatment. Baseline (day −3), $n = 16$; day 4, $n = 10$; day 7, $n = 10$; day 18, $n = 10$; day 45, $n = 7$. Compared to baseline, ****$P = 3.28 \times 10^{-8}$ for day 4, ****$P = 1.87 \times 10^{-6}$ for day 7, ****$P = 2.63 \times 10^{-7}$ for day 18 and ***$P = 1.23 \times 10^{-4}$ for day 45. Two-way analysis of variance (ANOVA) with Bonferroni correction. Dots represent individual mice. **c**, TO labeling (mean ± s.e.m.) in the same mice as in **b**. Data represent the percentages of TO[+] cells. ****$P = 3.18 \times 10^{-8}$ for day 4. Two-way

ANOVA with Bonferroni correction. Dots represent individual mice. **d**, *Flt3*Cre-tdTomato labeling (as in **c**) upon 5FU treatment (day 0). Analysis at baseline (day −7 or −2) and on days 10, 17 and 24 after 5FU ($n = 6$). In platelets, ****$P = 1.19 \times 10^{-47}$ for day 10, ****$P = 5.68 \times 10^{-42}$ for day 17 and ****$P = 3.95 \times 10^{-23}$ for day 24 compared to baseline; ****$P = 3.60 \times 10^{-5}$ for day 17 and ****$P = 4.10 \times 10^{-27}$ for day 24, both compared to day 10. Compared to baseline, ****$P = 4.05 \times 10^{-18}$ for day 24 in erythrocytes and ****$P = 6.30 \times 10^{-11}$ for day 17 in myeloid cells. For myeloid cells, ****$P = 9.60 \times 10^{-7}$ for day 24 when compared to day 17. Linear mixed-model two-sided analysis with $P$-value adjustment by the Benjamini−Hochberg procedure. **e**, Platelet counts in *Flt3*Cre[tg/+] *R26*[Tom/+] mice at baseline (day −10 or −3; $n = 3$) and on day 3 after anti-CD42b antibody treatment ($n = 5$). Dots represent individual mice, and lines represent mean ± s.e.m. **f**, *Flt3*Cre-tdTomato labeling (as in **c**) after anti-CD42b treatment (day 0). Analysis at baseline (day −10 or −3) and on days 3, 7 and 11 after anti-CD42b treatment ($n = 5$). A marginal significance was observed on day 11 (*$P = 0.0466$). Linear mixed-model two-sided analysis with $P$-value adjustment by the Benjamini−Hochberg procedure.

being heavily platelet biased, also contributing to a lesser degree to granulocytes/monocytes and erythrocytes (more prominent upon challenge through secondary transplantation)[9]. In contrast, upon specific depletion of platelets, the relative contributions of the *Flt3*[−] and *Flt3*[+] pathways were not significantly affected, compatible with

preexisting MkPs, rather than HSCs, being responsible for the rapid platelet replenishment. Collectively, these findings suggest that the slower but more potent platelet replenishment by multi-HSCs through an *Flt3*[+] pathway, including multiple stages of progenitor amplification before megakaryocyte commitment, is the default pathway in steady

state; in contrast, a shorter and faster *Flt3⁻* progenitor pathway initiated by P-HSCs becomes more prominent shortly after challenges that reduce relevant MkPs in the BM.

Single-cell RNA-sequencing analysis demonstrated that even fully Mk-restricted progenitors (MkPs)[19] are molecularly distinct in the two pathways. This agrees with the recent identification of MkP and megakaryocyte heterogeneity[13,14,55], raising the possibility that the two pathways might also replenish platelets with distinct properties[56].

Single-cell RNA sequencing provided insights into DEGs encoding cell-surface antigens that should facilitate the identification and further characterization of P-MkPs and multi-MkPs also in wild-type mice. *Cd24a* was highly upregulated in P-MkPs, also at the protein level, overlapping with differential expression of *Vwf*-tdTomato. In contrast, CD49b and CD48 expression was almost exclusive to multi-MkPs. CD48 has been reported to be expressed on an MkP subset[14] and a distinct megakaryocyte subset with proposed immunoregulatory functions, including pathogen recognition and phagocytosis[55].

To what degree the two pathways result in the replenishment of functionally distinct megakaryocytes or platelets remains to be investigated. It would also be important to exclude that neither of the two pathways produces dysregulated platelets. Being the first evidence of alternative differentiation pathways from distinct and nonhierarchically related HSCs, for any short-lived blood cell lineage, it raises the possibility that other short-lived myelo-erythroid blood cell lineages might also be replenished through more than one pathway. While no evidence exists for alternative platelet replenishment pathways in human hematopoiesis, this possibility is supported by findings compatible with the existence of human P-HSCs[57,58]. The identification of a fast-track platelet replenishment pathway initiated by a distinct class of HSCs could provide a platform for combatting transplantation- and drug-induced thrombocytopenia through means to stimulate this pathway or by expanding P-HSCs.

## Online content

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

¹Haematopoietic Stem Cell Biology Laboratory, MRC Weatherall Institute of Molecular Medicine, University of Oxford, Oxford, UK. ²MRC Molecular Haematology Unit, MRC Weatherall Institute of Molecular Medicine, University of Oxford, Oxford, UK. ³Centre for Inflammatory Disease, Department of Immunology and Inflammation, Imperial College London, London, UK. ⁴Department of Medicine Huddinge, Center for Hematology and Regenerative Medicine, Karolinska Institutet, Stockholm, Sweden. ⁵Department of Cell and Molecular Biology, Karolinska Institutet, Stockholm, Sweden. ⁶Division of Medical Systems Bioengineering, Department of Medical Biochemistry and Biophysics, Karolinska Institutet, Stockholm, Sweden. ⁷Interdepartmental Centre for Stem Cells and Regenerative Medicine (CIDSTEM), Department of Biomedical, Metabolic and Neural Sciences, University of Modena and Reggio Emilia, Modena, Italy. ⁸Department of Pediatric Oncology, Karolinska University Hospital, Stockholm, Sweden. ⁹Division of Molecular Neurobiology, Department of Medical Biochemistry and Biophysics, Karolinska Institutet, Stockholm, Sweden. ¹⁰Department of Translational Neuroscience, University Medical Center Utrecht, Utrecht, the Netherlands. ¹¹Centre for Computational Biology, MRC Weatherall Institute of Molecular Medicine, University of Oxford, Oxford, UK. ¹²Cancer and Haematology Centre, Churchill Hospital, Oxford University Hospitals NHS Foundation Trust, Oxford, UK. ¹³Department of Hematology, Karolinska University Hospital, Stockholm, Sweden. ¹⁴These authors contributed equally: Joana Carrelha, Stefania Mazzi, Axel Winroth. ✉e-mail: j.carrelha@imperial.ac.uk; sten.eirik.jacobsen@ki.se

## Methods

### Animals

Animal experiments performed at the University of Oxford were approved by the Oxford Clinical Medicine Ethical Review Committee, and those performed at the Karolinska Institutet were approved by the regional review committee for animal ethics (Stockholms djurförsöksetiska nämnd). All experimental and mouse breeding procedures were performed in accordance with the UK Home Office and Swedish Jordbruksverket regulations.

Young adult (7–14 weeks old) *Vwf*-tdTomato/*Gata1*-eGFP mice[9,17] (*Vwf*-tdTomato[tg/+] *Gata1*-eGFP[tg/+]) and *Flt3*Cre/*Rosa26*tdTomato(Ai 9)/*Vwf*-eGFP/*Gata1*-eGFP mice[8,17,59] (*Flt3*Cre[tg/+] *R26*[Tom/+] *Vwf*-eGFP[tg/+] *Gata1*-eGFP[tg/+]) on a C57BL/6OlaHsd (University of Oxford) or C57BL/6JrJ (Karolinska Institutet) background were used as BM donors in single HSC transplantations. Seven- to 18-week-old wild-type CD45.1 B6.SJL-*Ptprc*[a]*Pepc*[b]/BoyJ (University of Oxford) and B6.SJL-*Ptprc*[a]*Pepc*[b]/BoyCrl (Karolinska Institutet) mice were used as recipients in primary and secondary transplantations, as donors of unfractionated BM competitor cells, and for BM analysis after anti-CD42b antibody treatment. Recipient mice that did not survive or had to be killed before 16–18 weeks after the primary or secondary transplantation were excluded from analyses. *Flt3*Cre/*Rosa26*tdTomato(Ai14) mice (*Flt3*Cre[tg/+] *R26*[Tom/+]; 7–11 weeks old) and *Vav*Cre/*Rosa26*tdTomato(Ai14) mice[51] (*Vav*Cre[tg/+] *R26*[Tom/+]; 8–23 weeks old) on a C57BL/6JrJ background were used for fate-mapping and CP and 5FU treatment experiments. *Flt3*Cre/*Rosa26*tdTomato(Ai9)/*Vwf*-eGFP/*Gata1*-eGFP (*Flt3*Cre[tg/+] *R26*[Tom/+] *Vwf*-eGFP[tg/+] *Gata1*-eGFP[tg/+]; 8–13 weeks old) mice on a C57BL/6OlaHsd background were also used for steady-state fate-mapping and anti-CD42b treatment experiments. Mice were housed in individually ventilated cages at the Oxford JR facility (12/12 h light/dark cycle, 19–24 °C and 45–65% humidity) and the Karolinska Institutet KM facility (12/12 h light/dark cycle, 22 ± 1 °C and 50% humidity).

### Single-cell transplantations

Single-cell sorting of adult BM HSCs was performed using a FACSAria II or FACSAria Fusion cell sorter (BD Biosciences), prepared by crushing pelvic and leg bones (and optionally also sternum and spine bones) into PBS with 5% fetal calf serum (FCS; Sigma-Aldrich) and 2 mM EDTA (Sigma-Aldrich). Single phenotypically defined HSCs (Extended Data Fig. 1a) were sorted from *Vwf*-tdTomato[tg/+] *Gata1*-eGFP[tg/+] mice (*Vwf*-tdTomato[+] and *Vwf*-tdTomato[−] fractions of LSK*Gata1*-eGFP[−]CD34[−]CD150[+]CD48[−] or LSK*Gata1*-eGFP[−]CD34[−]CD150[+]CD48[−]CD201[+] cells) and *Flt3*Cre[tg/+] *R26*[Tom/+] *Vwf*-eGFP[tg/+] *Gata1*-eGFP[tg/+] mice (*Vwf*/*Gata1*-eGFP[+] and *Vwf*/*Gata1*-eGFP[−] fractions of LSKCD34[−]CD150[+]CD48[−] or LSKCD34[−]CD150[+]CD48[−]CD201[+] cells).

Comparison of single-cell expression and coexpression of *Vwf*-eGFP and *Gata1*-eGFP in BM LSKCD34[−]CD150[+]CD48[−] cells showed that <10% express *Gata1*-eGFP (Extended Data Fig. 7c). Moreover, *Gata1*-eGFP and CD201 are mutually exclusive in LSKCD34[−]CD150[+]CD48[−] cells, indicating that eGFP expression in LSKCD34[−]CD150[+]CD48[−]CD201[+] cells mainly reflects *Vwf*-eGFP[+] HSCs (Extended Data Fig. 7c). Regardless of their cell-surface phenotype, P-HSCs and multi-HSCs were defined based on their long-term lineage replenishment pattern as established by blood lineage analysis at multiple time points (see the next sections).

In experiments with mice that coexpress *Vwf*-eGFP and *Gata1*-eGFP, eGFP was used for sorting of single eGFP[+] and eGFP[−] HSCs for transplantation and to identify donor-derived blood platelets (which express both *Vwf* and *Gata1*) and erythrocytes (which express *Gata1*) in the transplantation recipients.

Single HSCs were sorted by an automated cell deposition unit, refrigerated at 4 °C, into 96-well round-bottom plates (Corning) with 100 μl per well of Iscove's modified Dulbecco's medium (IMDM, Gibco) with 20% BIT-9500 serum substitute (Stem Cell Technologies), 100 U ml[−1] penicillin and 0.1 mg ml[−1] streptomycin (100× Pen/Strep,

Hyclone), 2 mM L-glutamine (Gibco) and 0.1 mM 2-mercaptoethanol (Sigma-Aldrich). Single index-sorted HSCs were mixed with 2–3 × 10[5] wild-type CD45.1 unfractionated BM competitor cells (100 μl per well) and transplanted by intravenous lateral tail-vein injection into lethally irradiated CD45.1 mice (10–10.5 Gy, cesium-137 or X-ray). BM cell counts were measured manually with a hemacytometer and/or an automated cell counter (Sysmex XP-300 or ABX Pentra ES 60).

### Blood reconstitution analysis

Peripheral blood was collected from a lateral tail vein into lithium–heparin or K3 EDTA microvettes (Sarstedt). The platelet supernatant was collected after centrifugation of blood samples at 100*g* for 10 min at room temperature. Then, it was mixed with a small fraction (0.5–1 μl) of red precipitate for combined analysis of platelets and erythrocytes. The remaining precipitate was incubated 1:1 with dextran (Sigma-Aldrich, *M*ᵣ 450,000–650,000) 2% w/v in PBS for 20–30 min at 37 °C. Erythrocytes were lysed by incubation in ammonium chloride solution (Stem Cell Technologies) for 2 min at room temperature. Leukocyte samples were incubated with purified CD16/32 (Fc-block) for 10–15 min at 4 °C. Then, they were stained with anti-mouse antibodies for 15–20 min at 4 °C in PBS with 1–5% FCS and 2 mM EDTA. Samples were analyzed using LSRII and Fortessa cytometers (BD Biosciences). See Supplementary Table 5 for antibody details.

Donor-derived platelets were defined as follows: CD150[+]CD41[+]TER119[−]*Vwf*-tdTomato[+]*Gata1*-eGFP[+] for *Vwf*-tdTomato[tg/+] *Gata1*-eGFP[tg/+] donors and CD150[+]CD41[+]TER119[−]*Vwf*/*Gata1*-eGFP[+] for *Flt3*Cre[tg/+] *R26*[Tom/+] *Vwf*-eGFP[tg/+] *Gata1*-eGFP[tg/+] donors. Donor-derived erythrocytes: TER119[+]CD150[−]CD41[−]*Vwf*-tdTomato[−]*Gata1*-eGFP[+] for *Vwf*-tdTomato[tg/+] *Gata1*-eGFP[tg/+] donors and TER119[+]CD150[−]CD41[−]*Vwf*/*Gata1*-eGFP[+] for *Flt3*Cre[tg/+] *R26*[Tom/+] *Vwf*-eGFP[tg/+] *Gata1*-eGFP[tg/+] donors. Donor-derived myeloid (granulocyte/monocyte) cells: CD11b[+]NK1.1[−]CD19[−]CD4/CD8a[−]CD45.1[−]CD45.2[+]. Donor-derived B cells: CD19[+]NK1.1[−]CD4/CD8a[−]CD11b[−]CD45.1[−]CD45.2[+]. Donor-derived T cells: CD4/CD8a[+]NK1.1[−]CD11b[−]CD19[−]CD45.1[−]CD45.2[+]. The granulocyte/monocyte identity of CD11b[+]NK1.1[−]CD19[−]CD4/CD8a[−]CD45.1[−]CD45.2[+] cells from reconstituted recipient mice was confirmed by cytospins stained with eosin Y/azure A/methylene blue (Richard-Allan Scientific Three-Step Stain Set, Thermo Fisher Scientific) (Supplementary Fig. 4).

### Categorization of reconstitution patterns

All five mature blood cell lineages (platelets, erythrocytes, myeloid cells, B cells and T cells) were considered. We considered primary and secondary recipients to be reconstituted by HSCs if the donor contribution to platelets was ≥0.1% at ≥16–18 weeks after transplantation[9]. Mice reconstituted by single *Vwf*[+] P-HSCs and *Vwf*[−] multi-HSCs were defined at ≥16–18 weeks after transplantation as follows. *Vwf*[+] P-HSC: donor platelets ≥0.1%; donor platelet percentage ≥50-fold higher than donor erythrocytes and myeloid cells; and donor B and T cells undetectable (≤0.01%). Where indicated (Fig. 7b and Extended Data Fig. 7e,f), the stricter group of platelet-restricted HSCs was considered, in which donor erythrocyte, myeloid cell, B cell and T cell lineages were all below the detection level (≤0.01%). *Vwf*[−] multi-HSC: donor platelets ≥0.1%; donor erythrocytes, myeloid cells, B cells and T cells all >0.01%; and donor platelets, erythrocytes and myeloid cells all ≤2-fold higher than B and T cells. Mice reconstituted with multilineage patterns with ≥2-fold platelet, platelet–erythroid and platelet–erythroid–myeloid bias were excluded from the *Vwf*[−] multi-HSC group, as such biases are typical of *Vwf*[+] multi-HSCs[9].

### Reconstitution analysis of HSPCs

BM HSPC reconstitution analysis was performed using FACSAria Fusion, LSRII and LSR Fortessa flow cytometers (BD Biosciences) after crushing pelvic and leg bones (and optionally also sternum and spine bones) into PBS with 5% FCS and 2 mM EDTA, followed by cKIT enrichment according to the manufacturer's instructions (CD117 MicroBeads

and magnetic activated cell sorting (MACS) LS columns, Miltenyi Biotec). cKIT-enriched BM cells were incubated with purified CD16/32 (Fc-block) for 15–20 min at 4 °C, followed by anti-mouse antibody staining for 15–20 min at 4 °C. For the myeloid progenitor panel, cells were incubated with fluorophore-conjugated CD16/32 before further staining. See Supplementary Table 5 for antibody details.

Phenotypic BM populations were defined as follows: LSK, $LIN^-SCA1^+cKIT^+$; LK, $LIN^-SCA1^-cKIT^+$; LT-HSC, $LSKFLT3^-CD150^+CD48^-$; ST-HSC, $LSKFLT3^-CD150^-CD48^-$; MPP2, $LSKFLT3^-CD150^+CD48^+$; MPP3, $LSKFLT3^-CD150^-CD48^+$; MPP4, $LSKFLT3^+$; MkP, $LKCD150^+CD41^+$; pre-MegE progenitor, $LKCD41^-CD16/32^-CD150^+CD105^-$; CFU-E, $LKCD41^-CD16/32^-CD150^-CD105^+$; preGM progenitor, $LKCD41^-CD16/32^-CD150^-CD105^-$; GMP, $LKCD41^-CD16/32^+CD150^-CD105^-$. Cells within each population were considered donor-derived cells when $CD45.2^+CD45.1^-$ or when $Gata1$-$eGFP^+$ in the case of CFU-E cells due to their low CD45 expression.

### In vitro lineage potentials

Donor-derived ($CD45.2^+CD45.1^-$) phenotypic HSC and MPP populations, as defined above, were bulk sorted into Eppendorf tubes with X-VIVO 15 medium containing gentamycin and L-glutamine (Lonza) and supplemented with 10% FCS, 0.1 mM 2-mercaptoethanol, 25 ng ml$^{-1}$ mouse stem cell factor (PeproTech), 25 ng ml$^{-1}$ human thrombopoietin (PeproTech), 10 ng ml$^{-1}$ human FLT3 ligand (Immunex) and 5 ng ml$^{-1}$ mouse interleukin-3 (PeproTech). The volume of sorted cells was diluted as needed and manually distributed at an average of one cell per well into Terasaki microplates (Thermo Fisher Scientific) at 20 µl per well. The growth of granulocytes/macrophages and megakaryocytes was scored under an inverted microscope after 8 days of culture at 37 °C and 5% $CO_2$ in a humidified incubator. Granulocyte/macrophage and megakaryocyte scores were also confirmed in a representative subset of wells through the analysis of cytospins stained with eosin Y/azure A/ methylene blue (Richard-Allan Scientific Three-Step Stain Set, Thermo Fisher Scientific).

### Secondary hierarchical transplantations

Secondary transplantations were performed 16–38 weeks after the primary single-HSC transplantation, and a final blood analysis of primary recipients was performed a maximum of 2 weeks before sorting to confirm the reconstitution pattern. BM cell suspensions from primary recipients of a single P-HSC or multi-HSC were prepared by crushing pelvic and leg bones (and optionally also sternum and spine bones) into PBS with 5% FCS and 2 mM EDTA, followed by cKIT enrichment according to the manufacturer's instructions (CD117 MicroBeads and MACS LS columns, Miltenyi Biotec). Phenotypically defined donor-derived ($CD45.2^+CD45.1^-$) HSCs were sorted into supplemented IMDM as described above, mixed with $2$–$3 × 10^5$ wild-type CD45.1 unfractionated BM competitor cells and transplanted by intravenous lateral tail-vein injection into lethally irradiated CD45.1 secondary recipient mice (10–10.5 Gy, cesium-137 or X-ray).

In some experiments (Supplementary Table 1), all sortable donor-derived ($CD45.2^+CD45.1^-$) LSK $Vwf$-tdTomato$^+$ cells were transplanted while the $Vwf$-tdTomato$^-$ cells were transplanted, keeping the same $Vwf$-tdTomato$^+$/$Vwf$-tdTomato$^-$ ratio observed within the donor-derived phenotypic HSC compartment ($LSKCD150^+CD48^-CD45.2^+$). In other experiments (Supplementary Tables 2 and 3), all sortable donor-derived ($CD45.2^+CD45.1^-$) HSCs ($LSKCD150^+CD48^-$ split into $Vwf$-tdTomato$^+$ and $Vwf$-tdTomato$^-$ fractions or $Gata1$-$eGFP^-LSKCD150^+CD48^-$ split into $Vwf$-tdTomato$^+$/CD201$^+$, $Vwf$-tdTomato$^+$/CD201$^-$, $Vwf$-tdTomato$^-$/CD201$^+$ and $Vwf$-tdTomato$^-$/CD201$^-$ fractions) were transplanted from each primary donor.

### Smart-seq3 single-cell library preparation and sequencing

Single $CD45.2^+CD45.1^-LIN^-cKIT^+$ HSPCs generated in vivo from $Vwf$-tdTomato$^+$ P-HSCs and $Vwf$-tdTomato$^-$ multi-HSCs were sorted from single-HSC-transplanted mice 24–31 weeks after transplantation (Fig. 3a). In all mice, the lineage reconstitution pattern in the blood was established from at least two time points, the last one performed ≤2 weeks before the isolation of cells for Smart-seq3 processing. From mice reconstituted by $Vwf$-tdTomato$^+$ P-HSCs, an unbiased isolation of all $CD45.2^+CD45.1^-$ donor-derived $LIN^-cKIT^+$ cells was performed in combination with collection of index-sorting information. From mice reconstituted by $Vwf$-tdTomato$^-$ multi-HSCs, in addition to collection of index-sorting information, a targeted selection of distinct donor-derived HSPCs was performed to ensure that all $LIN^-cKIT^+$ compartments were sufficiently represented in the Smart-seq3 analysis.

To detect potential batch variations between individual plates and individual experiments, we combined the BM samples from three 12-week-old wild-type CD45.1 mice and stored them as frozen aliquots. One BM vial was thawed for each experiment, in which each 384-well sorted plate contained 24 GMP and 24 LSK cells from this internal batch control.

As previously described[23], single donor-derived $LIN^-cKIT^+$ cells and internal control GMP and LSK cells were sorted into 384-well plates with 3 µl of Smart-seq3 lysis buffer containing 0.5 U µl$^{-1}$ RNase inhibitor (Takara), 0.1% Triton X-100 (Sigma-Aldrich), 0.5 mM dNTPs each (Thermo Fisher Scientific), 0.5 µM Smart-seq3 oligonucleotide-dT primer (5′-biotin-ACGAGCATCAGCAGCATACGAT$_{30}$VN-3′, IDT) and 5% polyethylene glycol (Sigma-Aldrich). The concentrations of dNTPs, oligonucleotide-dT primer and polyethylene glycol were calculated based on a 4-µl total volume after the addition of 1 µl of reverse transcription mix. Immediately after the sort, each plate was spun down and stored at −80 °C. Generation of the cDNA library was done as previously described[23], according to detailed protocols published on protocols.io (https://doi.org/10.17504/protocols.io.bcq4ivyw), with 22 cycles of PCR amplification and using 100 pg of amplified cDNA for tagmentation. The amplified tagmented libraries were pooled, bead purified and analyzed using a high-sensitivity DNA chip (Agilent Bioanalyzer) and quantified using a Qubit fluorometer (Thermo Fisher Scientific). Pooled libraries were sequenced on a NovaSeq S-prime (batch 3) or S4 (batches 1 and 2) 150 × 2-bp paired-end flow cell. FASTQ files from batch 1 were downsampled to 25% to match the sequencing depth from batches 2 and 3 to achieve similar sensitivity in gene detection across batches.

### Processing and quality control of RNA-sequencing data

FASTQ files were demultiplexed and mapped to the mouse genome (Genome Reference Consortium Mouse Build 38, Ensembl release 99) using zUMIs (versions 2.9.4cq, 2.9.6 and 2.9.7)[23,60], and unique molecular identifier (UMI) reads for each gene were quantified. In line with previous studies[61], the exclusion of cells from analysis was based on UMI count distribution; only cells with >10,000 UMIs and <5% mitochondrial reads were included in the analysis (Supplementary Fig. 1a–c). In total, 5,014 cells derived from single transplanted $Vwf$-tdTomato$^+$ P-HSCs or $Vwf$-tdTomato$^-$ multi-HSCs were sequenced. After filtering, 4,768 cells were included in the downstream analyses. Counts were normalized using the logNormCounts function from the scran package[62]. After normalization, a pseudocount of 1 was added, and the data were $log_2$ transformed.

To identify HVGs, we used the modelGeneVar function from the scran package (Supplementary Fig. 1d). Batches were then integrated using a mutual nearest-neighbor approach (batchelor::fastMNN)[24] by first performing a multisample principal component analysis (PCA) and then finding the mutual nearest neighbors in this PCA space. To visualize any potential technical batch effects from each sequencing run, we plotted the internal control LSK and GMP cells, as well as the cells isolated from transplanted mice, before and after batch correction on the PCA plots (Supplementary Fig. 1e,f) used to generate the tSNE and UMAP visualization plots. Batch integration performance was evaluated using the iLISI and cLISI metrics[63] (Supplementary Fig. 1g).

UMAP plots were generated using a range of HVGs (500–5,000 genes; Extended Data Fig. 4c) demonstrating a preserved pattern. The 2,000 most HVGs were selected for both UMAP and tSNE plots (Supplementary Fig. 2a–c and Extended Data Fig. 4c,d).

Cell-surface marker expression was extracted from FlowJo workspaces using flowWorkspace[64] to parse the gating hierarchies. Cells lacking index-sorting information were assigned as nonindexed, and cells falling outside the defined gates were classified as nongated.

### Gene signature analysis
Molecular signatures were computed using the rank-based AUC score implemented in the AUCell package[65]. HSCs and MkPs derived from *Vwf*-tdTomato[+] P-HSCs and *Vwf*-tdTomato[−] multi-HSCs were compared based on their AUC scores for previously published signatures of lineage-restricted progenitors[19] and cell-cycle status[66] and for multiple HSC-associated signatures, including low-output HSC[26], high-output HSC[26], megakaryocyte-biased HSC[26], multi-HSC[26], HSC1 cluster[26], serial-engrafter HSC[26], LT-HSC[21], stem score[28], surface marker overlap[25], MolO[25], RA-CFP-dim HSC[29] and dormant HSC versus active HSC[27].

### Differential gene expression analysis
Differential gene expression analysis was performed with a combination of the Wilcoxon test for differences in expression levels and Fisher's exact test for expression frequencies, as previously described[67]. HSCs were defined as single cells with an AUCell score of >0.22 for the MolO signature[25], and MkPs were defined as single cells with an AUCell score of >0.25 for the MkP signature[19]. Gene expression differences with an adjusted $P$ value (combined Fisher) of <0.05 and an absolute value of $\log_2$(fold change) of >0.5 were considered significantly differentially expressed. Genes with one or more reads were classified as expressed within single cells. HALLMARK (v2023.2) gene-set enrichment analysis on the DEGs was performed by the preranked test in GSEA software version 4.3.3, using $\log_2$(fold changes) as the rank based on genes detected in more than ten cells.

### Pseudotime analysis
Total counts were normalized, and committed erythroid and myeloid progenitor cells were excluded based on >2 $\log_2$(expression) values of *Car1* and *Mpo*, shown to define committed erythroid and myeloid lineage progenitors[19], respectively (Extended Data Fig. 4a,b). The cell-cycle phase effect in the progenitor compartment was removed while retaining the difference from quiescent stem cells by regressing the difference between the S-phase score and the $G_2$M-phase score, as described in the Seurat package[68]. Variable features were selected, and batches were integrated as described above before dimensionality reduction with UMAP (Supplementary Fig. 2).

Pseudotime was computed using diffusion pseudotime[69] with default parameters, implemented in the SCANPY[70] package through the reticulate package in R. One cell with high expression of the MolO score (cell-id: AAGCCGTTGTCCATTG) was used as the starting cell. Differential gene expression analysis on the trajectories, with diffusion pseudotime as the independent variable, was performed using the tradeSeq package[36]. The filtered non-normalized count matrix was used as the input. Cells were assigned to either the P-HSC or multi-HSC trajectory with weight 1. Generalized additive models were fit with the default setting of six knots. Each gene was tested for differential expression as a function of pseudotime between P-HSC and multi-HSC using the patternTest function[36]. The genes were ordered according to their Wald statistic for downstream analysis and visualization. DEGs were identified within the top 70 genes when comparing cells replenished by *Vwf*-tdTomato[+] P-HSCs and *Vwf*-tdTomato[−] multi-HSCs, sorted based on the Wald statistic with an adjusted $P$ value of <0.01 and mean fold change of >1, and used to calculate the Pearson correlation between the two patterns. Correlation estimates and 95% CIs were calculated at 100 points along pseudotime. The same procedure was applied to the 70 most variable genes, representing randomly selected genes, to create a background correlation trajectory.

### Reanalysis of MkPs produced by *Hoxb5*Cre[ERT2]-labeled HSCs in steady-state mice
A recent publication[49] combined genetic fate mapping in *Hoxb5*Cre[ERT2/+] *R26*[Tom/+] reporter mice (a model for specific labeling of HSCs upon tamoxifen treatment) with single-cell RNA sequencing, in which the authors sequenced (Smart-seq2 or 10× single-cell RNA sequencing) HSPCs from steady-state *Hoxb5*Cre[ERT2/+] *R26*[Tom/+] mice at different time points following recombination induced by tamoxifen. We obtained the normalized gene expression (kindly provided by the authors) and compared molecularly defined MkPs replenished at early (days 3, 7 and 12) and late (days 112, 161 and 269) time points after tamoxifen for the expression of the most highly DEGs identified between MkPs derived from *Vwf*-tdTomato[+] P-HSCs (P-MkPs) and from *Vwf*-tdTomato[−] multi-HSCs (multi-MkPs), focusing on genes with an absolute $\log_2$(fold change) value of >1 and a $P$ value of <$10^{-15}$ in the comparison between P-MkPs and multi-MkPs (Fig. 4a and Supplementary Table 4). Of the 30 genes meeting these criteria, genes not detected in the *Hoxb5*Cre[ERT2] *R26*[Tom/+] dataset were excluded, resulting in five and six genes upregulated and downregulated in P-MkPs, respectively, when compared to multi-MkPs. A one-tailed Wilcoxon test was used for the analysis, and multiple testing was adjusted using the Benjamini–Hochberg method.

### In vivo *Flt3*Cre and *Vav*Cre fate mapping
Blood and BM samples were processed as described above and analyzed using LSRII and Fortessa cytometers (BD Biosciences). See Supplementary Table 5 for antibody details.

In 8- to 13-week-old *Flt3*Cre[tg/+] *R26*[Tom/+] *Vwf*-eGFP[tg/+] *Gata1*-eGFP[tg/+] steady-state mice and in recipients of single HSCs, platelets and erythrocytes were gated based on the eGFP[+] signal to exclude possible debris and to identify donor-derived cells, respectively. Leukocytes were gated CD41[−] in steady-state and CP- and 5FU-treated mice and *Vwf/Gata1*-eGFP[−] in single-HSC-transplanted mice (even if *Gata1* is expressed in some myeloid cells[17]) to exclude transfer of the *Flt3*Cre-tdTomato signal from adhering platelets.

To exclude cases with inefficient *Flt3*Cre recombination, we checked the coexpression of cell-surface FLT3 and *Flt3*Cre-tdTomato in the BM of single-HSC transplantation donors. Steady-state *Flt3*Cre and *Vav*Cre mice were only used if erythroid, myeloid and B and T blood cells were ≥98% *Flt3*Cre-tdTomato[+].

### In vivo treatments
For CP treatment, 7- to 11-week-old *Flt3*Cre[tg/+] *R26*[Tom/+] mice and 8- to 23-week-old *Vav*Cre[tg/+] *R26*[Tom/+] mice received a single intraperitoneal injection of 200 mg kg[−1] CP (European Pharmacopoeia, C3250000) or PBS control on day 0 (ref. 71). Blood was analyzed on days −3 (baseline), 4, 7, 18 and 45.

For 5FU treatment, 8- to 10-week-old *Flt3*Cre[tg/+] *R26*[Tom/+] mice received a single intraperitoneal injection of 150 mg kg[−1] 5FU (Accord Healthcare) on day 0 (ref. 72). Blood was analyzed on days −7 or −2 (baseline), 5 (only blood cell counts), 10, 17 and 24. For both treatments, blood cell counts were measured with a Sysmex XP-300 automated cell counter, and peripheral blood and BM were processed as described above for transplantation experiments. CP and 5FU were obtained from the pharmacy at the Karolinska University Hospital.

For platelet depletion (anti-CD42b) treatment, 12- to 13-week-old *Flt3*Cre[tg/+] *R26*[Tom/+] mice received a single intravenous injection of 2 mg kg[−1] of an anti-CD42b antibody (R300, Emfret Analytics)[73]. Platelets and erythrocytes were processed as described above. For BM analysis, 11- to 12-week-old CD45.1 mice were compared on day 3 after anti-CD42b treatment to untreated CD45.1 mice. BM cell suspensions were prepared by crushing femurs, tibiae and pelvic bones from both legs of each treated and untreated mouse into PBS with 5% FCS and

2 mM EDTA. BM cells were incubated with fluorophore-conjugated CD16/32 for 15–20 min at 4 °C, followed by anti-mouse antibody staining for 15–20 min at 4 °C. BM cellularity and blood cell counts were measured with a Sysmex XP-300 automated cell counter. See Supplementary Table 5 for antibody details.

## Analysis of reticulated platelets with TO

Platelet/erythrocyte cell suspensions, processed and antibody stained as described above, were incubated with 1 ml of BD Retic-COUNT reagent[74] (BD Biosciences) or with 1 ml PBS with 5% FCS and 2 mM EDTA—samples and negative controls, respectively—for 1 h at room temperature in the dark (according to the manufacturer's instructions, excluding fixation) and directly analyzed using Fortessa cytometers (BD Biosciences).

## Data analysis

Statistical comparisons were performed as specified in the figure legends, using R scripts, GraphPad Prism 9 software and QuickCalcs online tools (https://www.graphpad.com/quickcalcs).

Kinetics, in which the blood of the same mouse was sampled at multiple time points (replenishment kinetics after transplantation, blood cell counts after CP treatment and Cre-driven tdTomato labeling after CP treatment), were analyzed with mixed-effects models using the lme4 R package. Mouse identity was assigned a random effect to account for intermouse variation. All combinations of time points and experimental conditions were assigned a fixed effect. Significance testing of all fixed-effects contrasts was done using the emmeans R package, and $P$ values were adjusted using the Benjamini–Hochberg procedure.

Mixed-effects model analysis was also performed with a similar setup in the built-in statistical package of GraphPad Prism 9 for Cre-driven tdTomato labeling of reticulated platelets.

Flow cytometry data were acquired using BD FACSDiva version 9.0 software and analyzed using BD FlowJo version 10.8.1 software.

## Reporting summary

Further information on research design is available in the Nature Portfolio Reporting Summary linked to this article.

## Data availability

RNA-sequencing data have been deposited to the public repository ArrayExpress under accession number E-MTAB-13935. Additional relevant information and material will be available from the corresponding authors upon request (j.carrelha@imperial.ac.uk and sten.eirik. jacobsen@ki.se). Source data are provided with this paper.

## Code availability

Code for RNA-sequencing analysis and statistical analysis of fate-mapping data is available via Zenodo at https://doi.org/10.5281/zenodo.10925564 (ref. 75).

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

## Acknowledgements

We appreciate the support of animal work from the University of Oxford Biomedical Services and Comparative Medicine at Karolinska Institutet, as well as the technical support from the WIMM Flow Cytometry Core Facility. We thank previous laboratory members A. Lord, L. Kettyle and E. Wojtowicz for scientific discussion and input during the initial phase of the study. We thank the Luc laboratory (Karolinska Institutet) for supplying the anti-mouse CD49b antibody and the Göttgens laboratory (University of Cambridge) for facilitating our reanalysis of their published single-cell RNA-sequencing data. The work was supported by grants from the following institutions: Knut and Alice Wallenberg Foundation grant KAW 2016.0105 (S.E.W.J.); Tobias Foundation grant 4-1122/2014 (S.E.W.J.); the Torsten Söderberg Foundation (S.E.W.J.); Strategic Research Area Stem Cells and Regenerative Medicine at Karolinska Institutet (S.E.W.J.), The Swedish Research Council, 538-2013-8995 (S.E.W.J.); UK Medical Research Council, MC_UU_12009/5 (S.E.W.J.); Knut and Alice Wallenberg Foundation, KAW 2015.0195 (P.S.W.); Swedish Research Council, 2015-03561 (P.S.W.); Cancerfonden, 22-2178-Pj (P.S.W.); and Radiumhemmets Forskningsfonder, 224132 (P.S.W.).

## Author contributions

S.E.W.J. conceptualized and supervised the studies with input from J.C., S.M., A.W., C.N. and P.S.W. S.E.W.J., J.C., S.M., A.W., T.Y. and P.S.W. designed the experiments and analyzed the data. J.C., S.M., A.W., M.H.-J., C.Z. and K.H. performed the experiments with assistance from M.S., M.S.B., M.L., B.W., Y.M., E.M., R.N., H.I., K.B.S., F.G., C.S.K. and A.A. B.W., A.H. and E.C. assisted with mouse husbandry and procedures.

Single-cell RNA-sequencing data were analyzed by A.W. and P.S.W. with input from M.H.-J., C.Z., T.Y., K.S., S.T., A.J.M., S.L., R.S. and S.E.W.J. S.E.W.J., J.C., S.M., A.W., T.Y. and P.S.W. wrote the paper, which was subsequently reviewed and approved by all authors.

## Competing interests

M.H.-J., C.Z. and R.S. report holding shares in Xpress Genomics AB, and M.H.-J. and R.S. are inventors on the patent relating to Smart-seq3 that is licensed to Takara Bio USA. The other authors declare no competing interests.

## Additional information

**Extended data** is available for this paper at https://doi.org/10.1038/s41590-024-01845-6.

**Correspondence and requests for materials** should be addressed to Joana Carrelha or Sten Eirik W. Jacobsen.

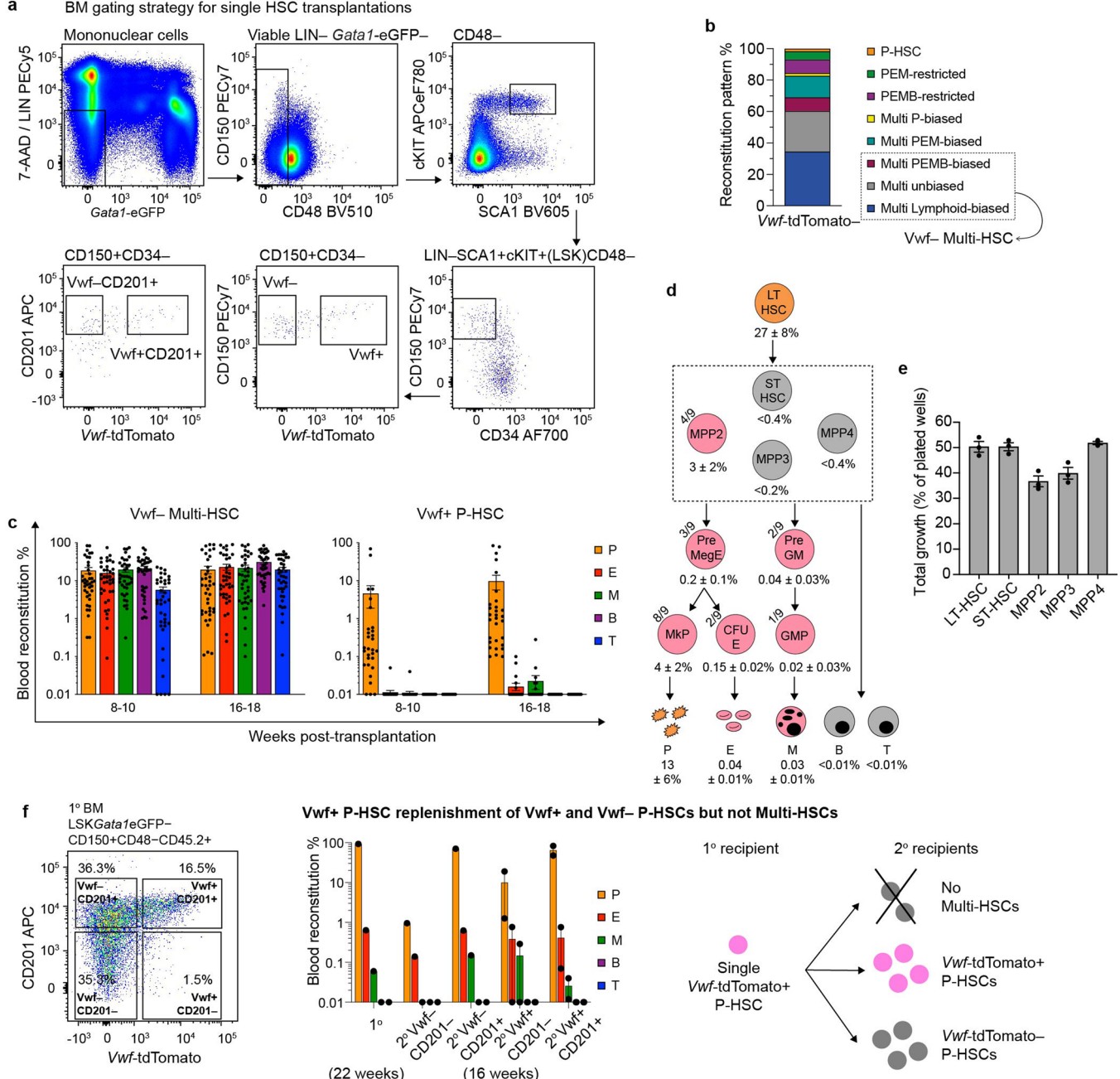

**Extended Data Fig. 1 | In vitro and in vivo potentials of *Vwf*-tdTomato⁻ Multi-HSCs and *Vwf*-tdTomato⁺ P-HSCs. a**, Representative flow cytometry profiles and gating strategy for sorting of BM *Vwf*-tdTomato⁻ and *Vwf*-tdTomato⁺ LSK*Gata1*-eGFP⁻CD34⁻CD150⁺CD48⁻ cells for single cell transplantations. **b**, Frequency (%) of long-term (16–18 weeks post-transplantation) blood replenishment patterns in recipients reconstituted by a single LSK*Gata1*-eGFP⁻CD34⁻CD150⁺CD48⁻ (also CD201⁺ in some experiments) *Vwf*-tdTomato⁻ HSC (n = 58; 13 experiments). The 3 blood patterns included as *Vwf*⁻ Multi-HSCs in the present studies are highlighted. **c**, Peripheral blood replenishment (mean % ± s.e.m.) in recipients reconstituted by a single LSK*Gata1*-eGFP⁻CD34⁻CD150⁺CD48⁻ *Vwf*-tdTomato⁻ Multi-HSC (n = 39) or *Vwf*-tdTomato⁺ P-HSC (n = 30); 13 experiments. Phenotypic populations defined in Methods and Fig. 1a. **d**, Complete HSPC hierarchy by single *Vwf*-tdTomato⁺ P-HSCs (n = 9). Same mice as in Fig. 1a. Mean ± s.e.m. % contribution to each population. Orange: reconstituted in all mice. Grey: below detection level in all mice; mean of detection thresholds. Pink: reconstitution in some but not all mice (frequency of reconstituted mice is indicated in the upper left of each circle); mean of positive mice. Phenotypic populations defined in Methods and Fig. 1a. **e**, Frequency of wells with cell growth (mean ± s.e.m.) of the in vitro assay for granulocyte/macrophage (GM) and Mk lineage potentials shown in Fig. 1c. LT-HSC: 580 plated wells; ST-HSC: 660 wells; MPP2: 652 wells; MPP3: 720 wells; MPP4: 720 wells. Phenotypic populations defined in Methods and Fig. 1a. **f**, Unique case of a single *Vwf*-tdTomato⁺ P-HSC replenishing *Vwf*-tdTomato⁺ as well as *Vwf*-tdTomato⁻ P-HSCs but no Multi-HSCs. Left: *Vwf*-tdTomato and CD201 expression in LSK*Gata1*-eGFP⁻CD150⁺CD48⁻CD45.2⁺ cells in the primary CD45.1 recipient of a single CD45.2 LSK*Gata1*eGFP⁻CD34⁻CD150⁺CD48⁻CD201⁺ *Vwf*-tdTomato⁺ P-HSC. Middle: Blood reconstitution in the primary (1⁰) recipient at 22 weeks post-transplantation and in the secondary (2⁰) recipients (mean ± s.e.m.) 16 weeks after transplantation. CD201⁻ *Vwf*-tdTomato⁻ n = 1; CD201⁺ *Vwf*-tdTomato⁻ n = 1; CD201⁻ *Vwf*-tdTomato⁺ n = 2; CD201⁺ *Vwf*-tdTomato⁺ n = 2. Right: Interpretation of results regarding (non-) hierarchical replenishment of Multi-HSCs and P-HSCs. See Fig.1a for cell phenotypes and Supplementary Table 3.

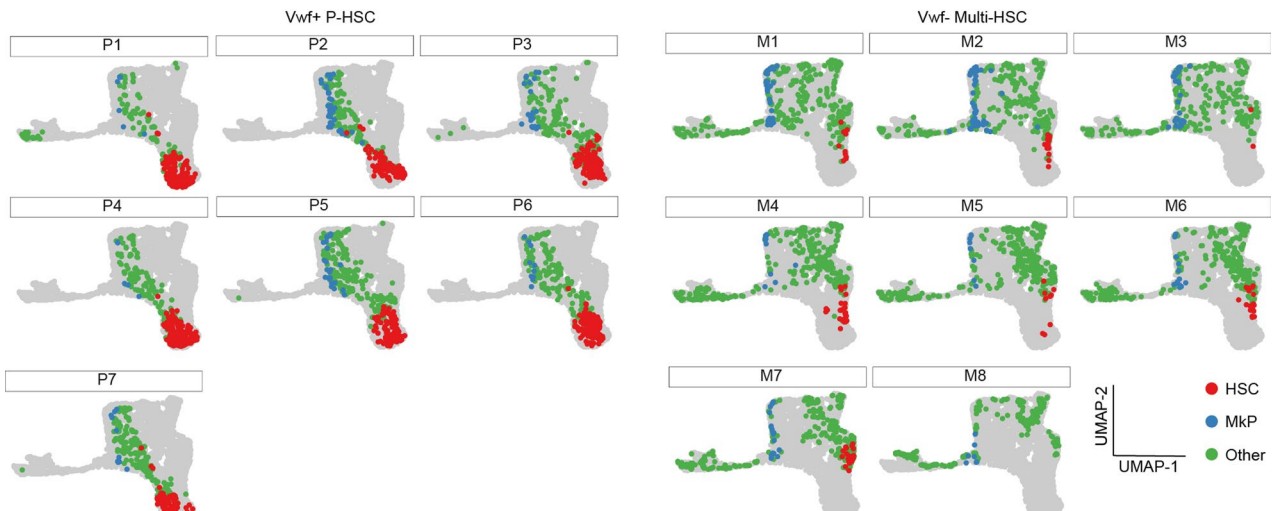

**Extended Data Fig. 2 | Single cell RNA sequencing UMAP plots of cells from replicate recipients of single *Vwf*-tdTomato⁺ P-HSC or *Vwf*-tdTomato⁻ Multi-HSC.** Batch corrected UMAP plots of LIN⁻cKIT + cells replenished by a single transplanted *Vwf*-tdTomato⁺ P-HSC (7 recipient mice; P1-7) or *Vwf*-tdTomato⁻ Multi-HSC (8 recipient mice; M1-8). Colors indicate molecularly defined HSCs (red), MkPs (blue), and other cells (green). Mice used for RNA sequencing do not overlap with mice in Supplementary Tables 1–3.

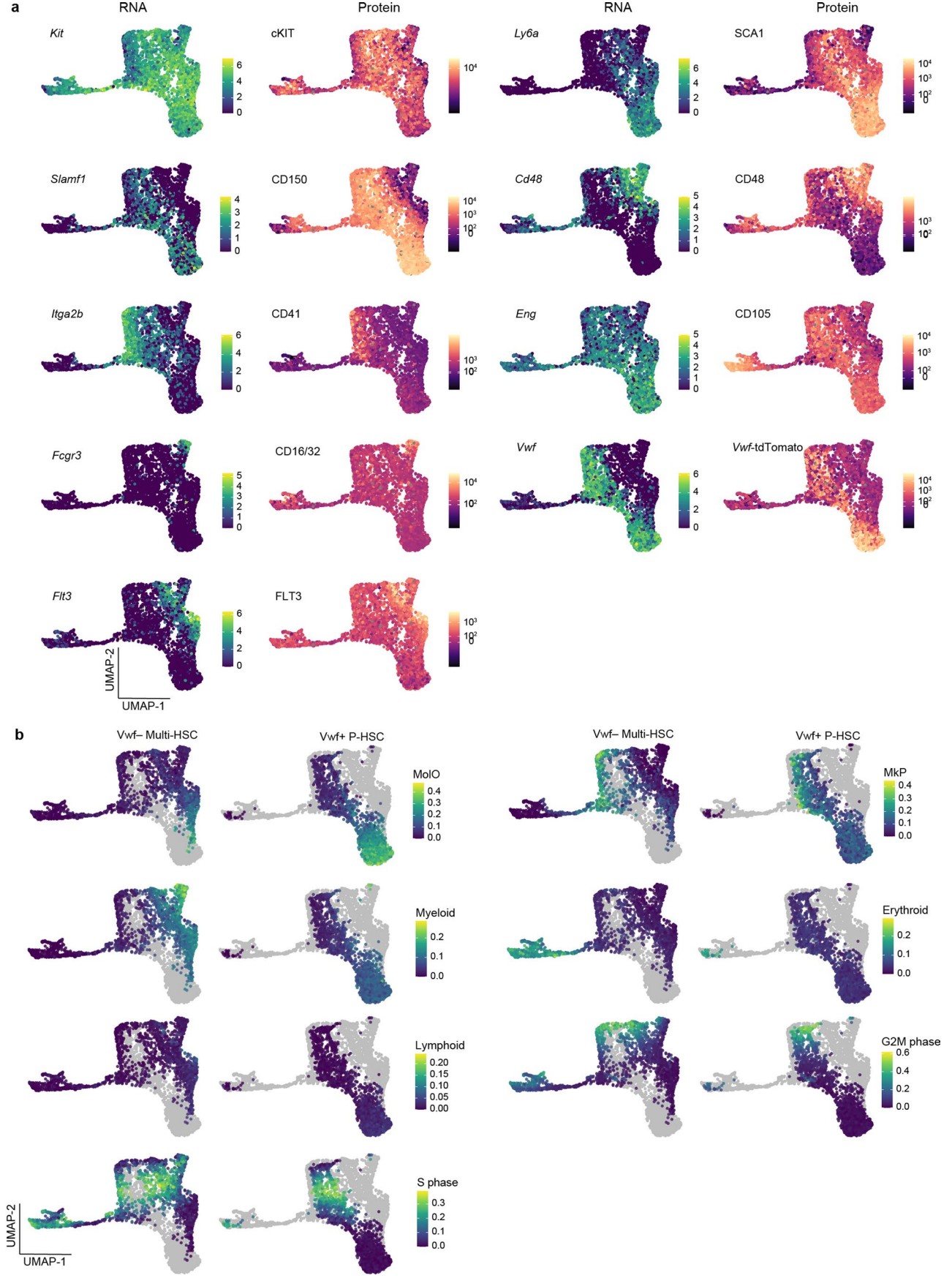

**Extended Data Fig. 3 | See next page for caption.**

**Extended Data Fig. 3 | Molecular signatures of HSPCs replenished from single *Vwf*-tdTomato⁻ Multi-HSCs or *Vwf*-tdTomato⁺ P-HSCs. a**, UMAP plots (as in Fig. 3b) of HSPC cell surface antigens; mRNA expression (left) and protein/reporter expression based on index sorting information from the same cells (right). **b**, UMAP plots of single LIN⁻cKIT⁺ HSPCs from mice reconstituted by single *Vwf*-tdTomato⁻ Multi-HSCs or *Vwf*-tdTomato⁺ P-HSCs. Color scales indicate expression of molecular signatures (AUCell scores) for HSCs (MolO), lineage restriction (MkP, erythroid, myeloid, and lymphoid), and cell cycle (G2M and S phases).

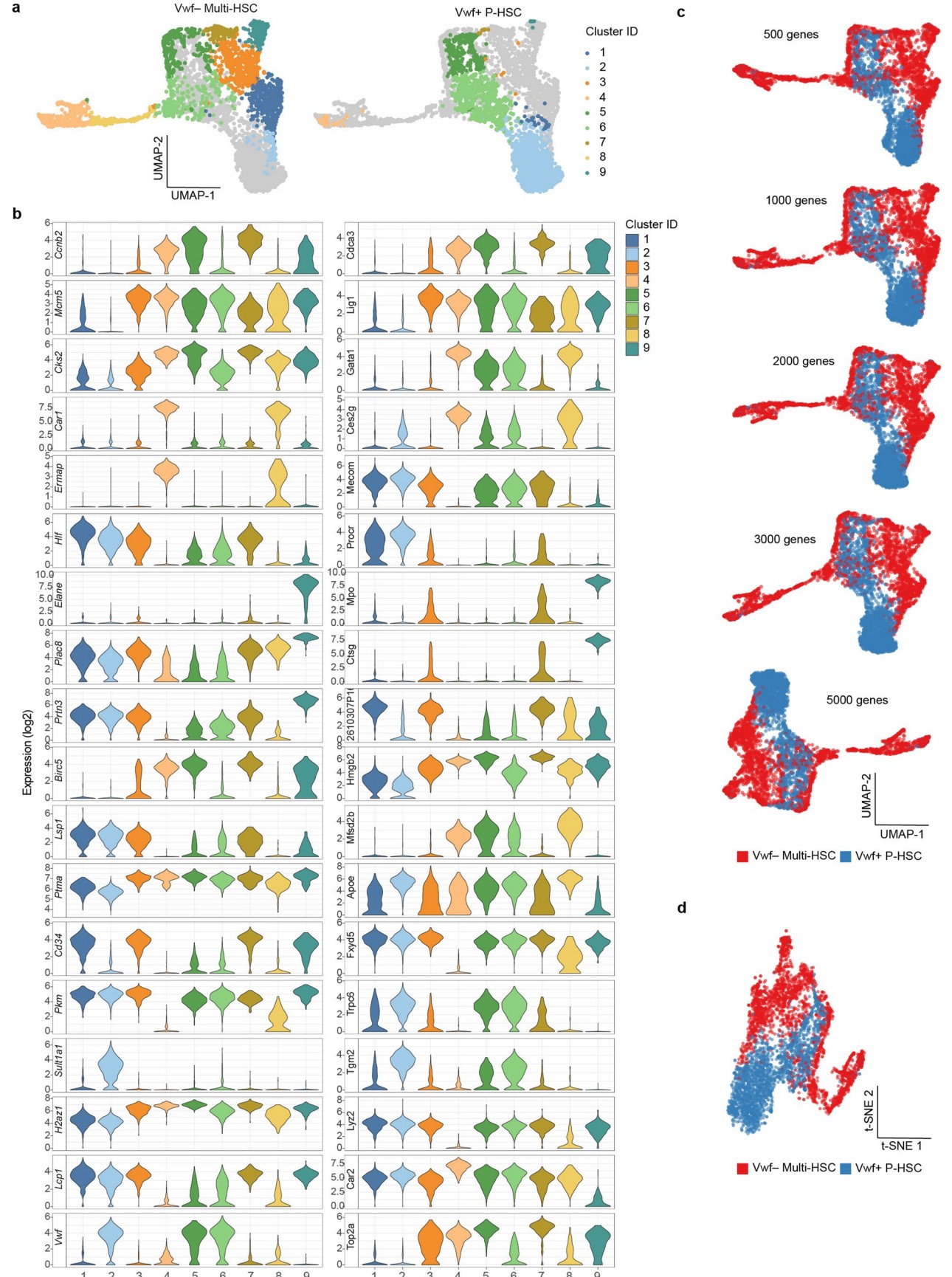

**Extended Data Fig. 4 | Nearest neighbor clustering and cluster markers.**
**a**, UMAP plot (as in Fig. 3b) with identified molecular clusters among the cells derived from *Vwf*-tdTomato⁻ Multi-HSC or *Vwf*-tdTomato⁺ P-HSC. Colors indicate individual clusters. **b**, Expression patterns of top genes defining clusters in **a**.

**c**, UMAP reduction computed based on indicated numbers of variable genes generated independently of the analysis pipeline used to generate UMAPs shown in Fig. 3b. **d**, t distributed stochastic neighborhood embedding (t-SNE) of the cells in Fig. 3b.

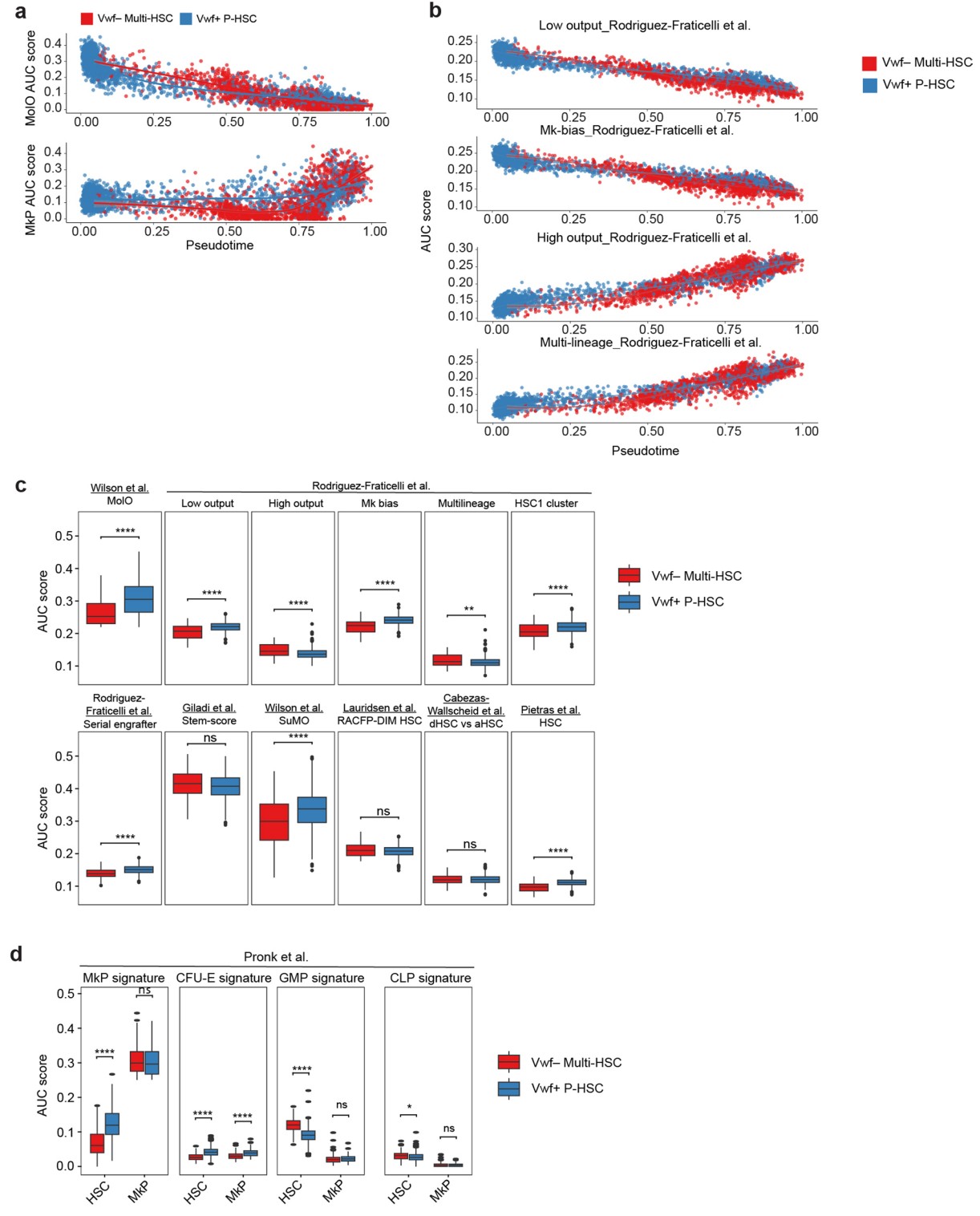

**Extended Data Fig. 5 | See next page for caption.**

**Extended Data Fig. 5 | HSC and Mk signatures of HSPCs replenished by single transplanted *Vwf*-tdTomato⁻ Multi-HSCs or *Vwf*-tdTomato⁺ P-HSCs. a**, MolO HSC (top) and MkP (bottom) AUC signature scores for HSPCs replenished by *Vwf*-tdTomato⁺ P-HSCs or *Vwf*-tdTomato⁻ Multi-HSCs along pseudotime. Dots represent individual cells and lines represent LOESS curves (local regression to fit a smooth curve through scatterplot) of AUC signature for the HSC subtype with 95% confidence interval (CI). **b**, Low output, high output, Mk-bias and multilineage AUC signature scores for HSPCs replenished by *Vwf*-tdTomato⁺ P-HSCs or *Vwf*-tdTomato⁻ Multi-HSCs along pseudotime. Dots represent individual cells and lines represent LOESS curves (local regression to fit a smooth curve through scatterplot) of AUC signature for the HSC subtype with 95% CI. **c, d**, AUC scores for stem cell associated signatures in HSCs (**c**) and for lineage-restricted progenitor signatures in HSCs and MkPs (**d**) replenished by *Vwf*-tdTomato⁺ P-HSCs (n = 1047 HSC and 119 MkPs from 7 mice) or *Vwf*-tdTomato⁻ Multi-HSCs (n = 97 HSCs and 177 MkPs from 8 mice), based on published gene signatures (see Methods). Boxes show first and third quartiles of the normalized expression values, where the line within each box indicates the median, whiskers indicate the largest value within the $\pm1.5*$IQR, and dots indicate outlier cells. * $P < 0.05$, ** $P < 0.01$, **** $P < 0.0001$, ns $P > 0.05$ (non-significant) by two-sided Wilcoxon test. $P$ values: MolO = $3.29 \times 10^{-13}$, Low output = $9.84 \times 10^{-11}$, High output = $4.88 \times 10^{-6}$, Mk bias = $8.58 \times 10^{-13}$, Multilineage = 0.0022, HSC1 cluster = $4.88 \times 10^{-6}$, Serial engrafter = $1.37 \times 10^{-11}$, Stem-score = 0.15, SuMO = $4.48 \times 10^{-5}$, RACFP-DIM = 0.15, dHSCvsaHSC = 0.97, HSC = $2.54 \times 10^{-13}$, HSC-MkP = $6.06 \times 10^{-21}$, MkP-MkP = 0.72, HSC-CFUE = $4.08 \times 10^{-24}$, MkP-CFUE = $2.65 \times 10^{-11}$, HSC-GMP = $1.90 \times 10^{-24}$, MkP-GMP = 0.72, HSC-CLP = 0.02, MkP-CLP = 0.72.

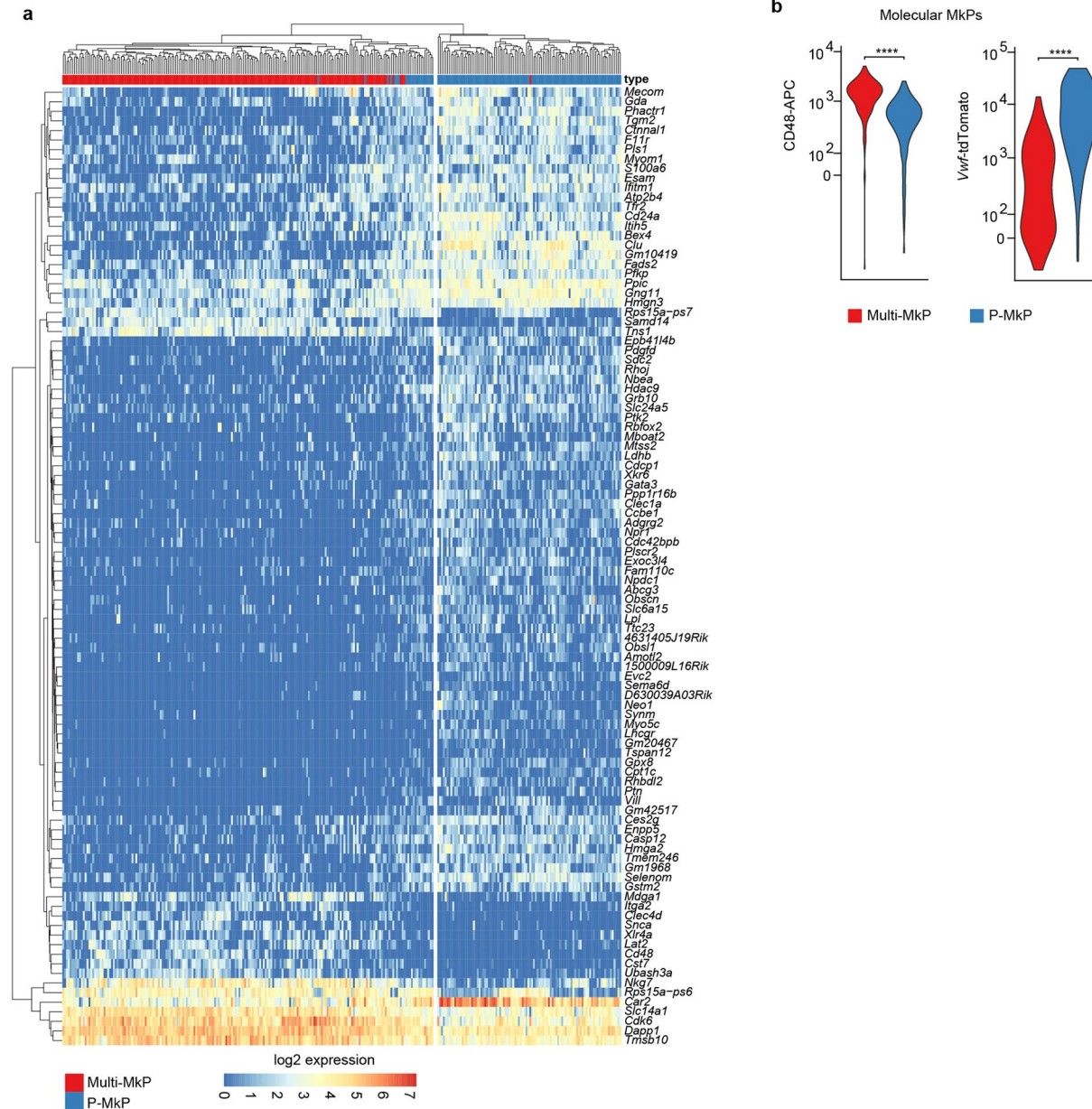

**Extended Data Fig. 6 | Molecularly distinct MkPs generated by *Vwf*-tdTomato⁺ P-HSCs and *Vwf*-tdTomato⁻ Multi-HSCs. a**, Heatmap of molecularly defined single MkPs replenished by single *Vwf*-tdTomato⁻ Multi-HSCs (Multi-MkPs; red) and *Vwf*-tdTomato⁺ P-HSCs (P-MkPs; blue) clustered based on expression of the 100 most differentially expressed genes (DEGs). **b**, Distribution of CD48 and *Vwf*-tdTomato protein/reporter level based on index data from FACS within molecularly defined Multi-MkPs and P-MkPs. **** $P < 0.0001$ by two-sided Wilcoxon test. $P$ values: CD48-APC = $1.69 \times 10^{-22}$, *Vwf*-tdTomato = $1.89 \times 10^{-26}$.

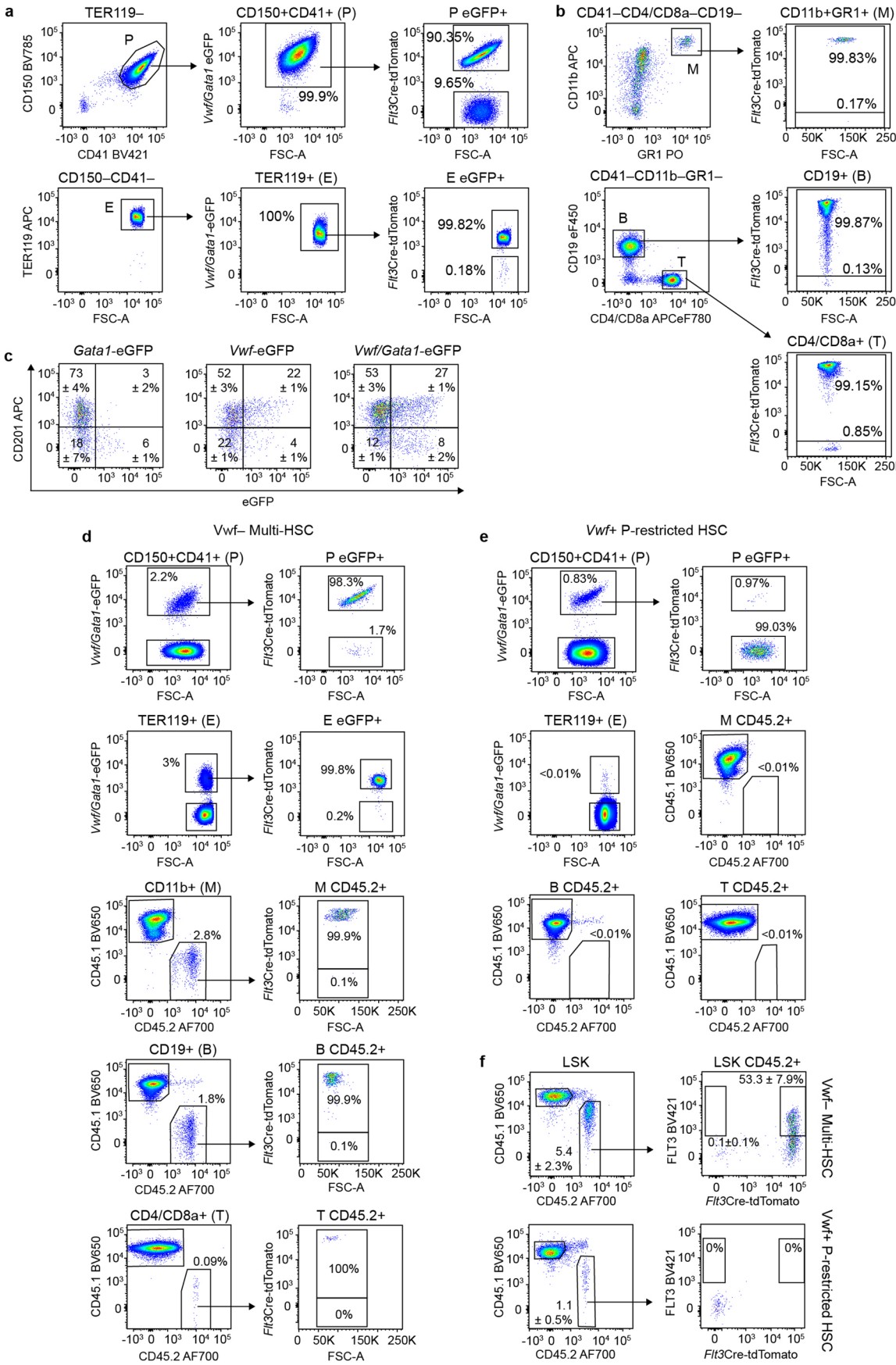

**Extended Data Fig. 7 | See next page for caption.**

**Extended Data Fig. 7 | *Flt3*Cre fate mapping of single HSCs. a-b**, Representative flow cytometry profiles (% of parent gates) of platelets and erythrocytes (**a**) and leukocyte lineages (**b**) of a 12 week old steady-state *Flt3*Cre$^{tg/+}$ *R26*$^{Tom/+}$ *Vwf*-eGFP$^{tg/+}$ *Gata1*-eGFP$^{tg/+}$ mouse. P: platelets, E: erythrocytes, M: myeloid cells, B: B lymphoid cells, T: T lymphoid cells. **c**, CD201, *Vwf*-eGFP, and *Gata1*-eGFP expression (mean ± s.e.m. % of parent gate) in BM LSKCD34$^-$CD150$^+$CD48$^-$ HSCs of 9–13 week old steady-state *Flt3*Cre$^{tg/+}$ *R26*$^{Tom/+}$ *Gata1*-eGFP$^{tg/+}$ (n = 2), *Flt3*Cre$^{tg/+}$ *R26*$^{Tom/+}$ *Vwf*-eGFP$^{tg/+}$ (n = 2), and *Flt3*Cre$^{tg/+}$ *R26*$^{Tom/+}$ *Vwf*-eGFP$^{tg/+}$ *Gata1*-eGFP$^{tg/+}$ mice (n = 4). **d, e**, Representative flow cytometry profiles (% of parent gates) of blood lineages of a mouse reconstituted by a single *Vwf*-tdTomato$^-$ Multi-HSC (**d**, n = 14) or a single *Vwf*-tdTomato$^+$ P-restricted HSC (**e**, n = 14) sorted from a *Flt3*Cre$^{tg/+}$ R26$^{Tom/+}$ *Vwf*-eGFP$^{tg/+}$ *Gata1*-eGFP$^{tg/+}$ donor. P: platelets, E: erythrocytes, M: myeloid cells, B: B lymphoid cells, T: T lymphoid cells. **f**, Cell-surface FLT3 and tdTomato (Tom) labelling (mean ± s.e.m. % of parent gate) in BM LSK cells of mice reconstituted by single *Vwf*-eGFP$^-$ Multi-HSCs (n = 14) or *Vwf*-eGFP$^+$ P-restricted HSCs (n = 14) sorted from *Flt3*Cre$^{tg/+}$ R26$^{Tom/+}$ *Vwf*-eGFP$^{tg/+}$ *Gata1*-eGFP$^{tg/+}$ donors. Note that all LSKFLT3$^+$ cells are Tom$^+$ whereas *Vwf*-eGFP$^+$ P-restricted HSCs do not generate LSKFLT3$^+$ or LSKTom$^+$ cells.

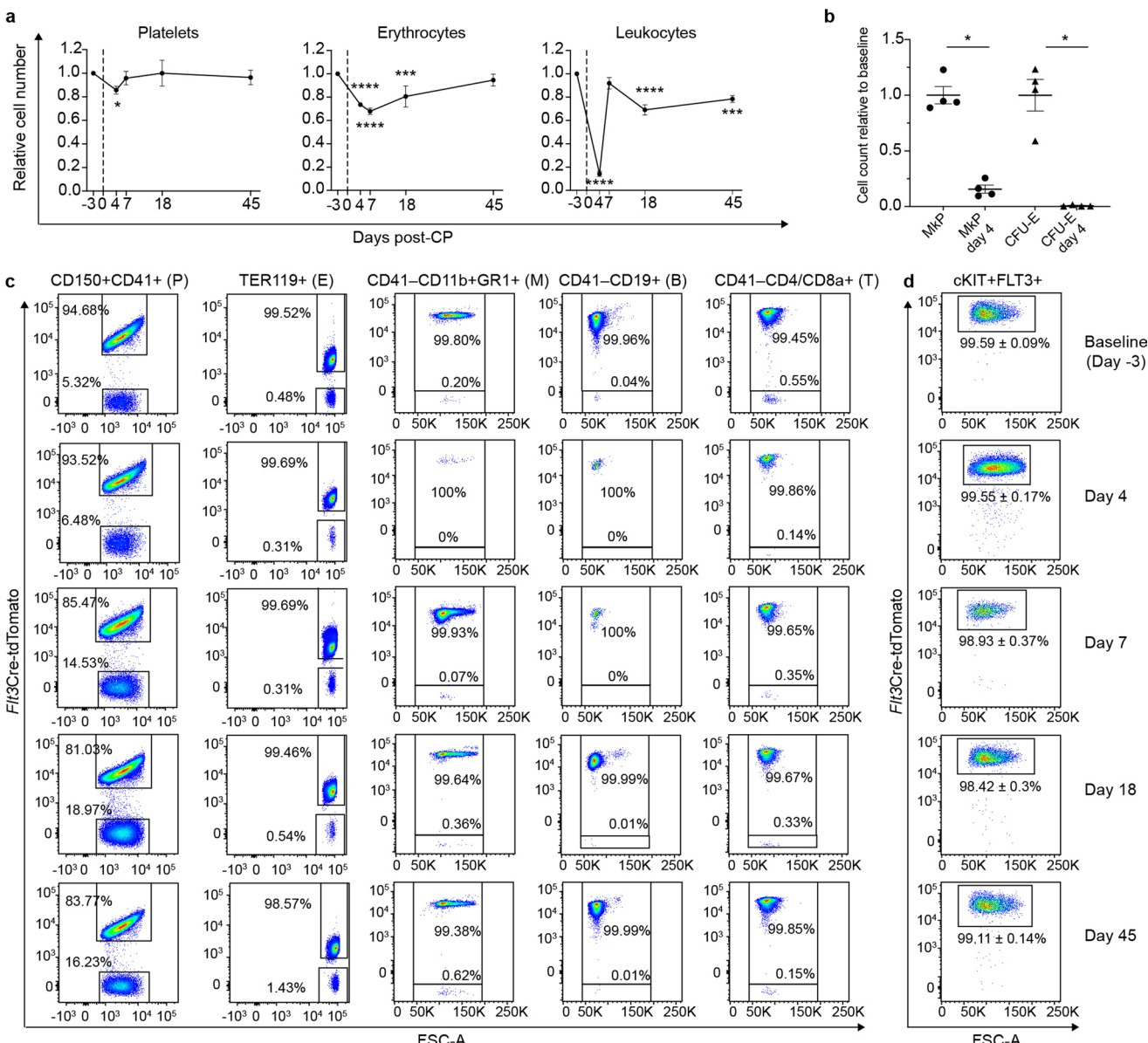

**Extended Data Fig. 8 | Fate mapping of *Flt3*Cre^tg/+ *R26*^Tom/+ mice after Cyclophosphamide treatment. a**, Blood cell counts (mean ± s.e.m.) of 7–11 week old *Flt3*Cre^tg/+ *R26*^Tom/+ mice before and after Cyclophosphamide (CP) treatment (day 0), relative to untreated baseline. Analysis on day -3 (baseline, n = 19), 4 (n = 13), 7 (n = 13), 18 (n = 10), and 45 (n = 7). All 19 mice with baseline measurement were bled on at least one additional time point; 7 of them were bled at all time points. Compared to baseline, *P = 2.27 × 10^−2 for d4 in platelets; in erythrocytes ****P = 8.54 × 10^−6 for d4, ****P = 4.81 × 10^−8 for d7 and ***P = 1.88 × 10^−3 for d18; in leukocytes ****P = 4.09 × 10^−36 for d4, ****P = 2.56 × 10^−7 for d18 and ***P = 1.72 × 10^−3 for d45. Linear mixed-model two-sided analysis with P-value

adjustment by Benjamini-Hochberg procedure. **b**, BM MkP and CFU-E progenitor cell counts (mean ± s.e.m.) in *Flt3*Cre^tg/+ *R26*^Tom/+ mice on day 4 following CP treatment (n = 4), relative to untreated mice (n = 4). Both progenitor counts were significantly decreased (*P = 0.03); two-tailed Mann-Whitney test. **c**, Representative flow cytometry profiles (% of parent gates) of blood lineages of an 8 week old *Flt3*Cre^tg/+ *R26*^Tom/+ mouse before and after CP treatment. **d**, *Flt3*Cre-tdTomato expression (mean ± s.e.m. % of parent gate) in BM LIN^−cKIT^+FLT3^+ progenitor cells of *Flt3*Cre^tg/+ *R26*^Tom/+ mice before and after CP treatment. Untreated (n = 4), and day 4 (n = 4), 7 (n = 4), 18 (n = 6), 45 (n = 7) after CP.

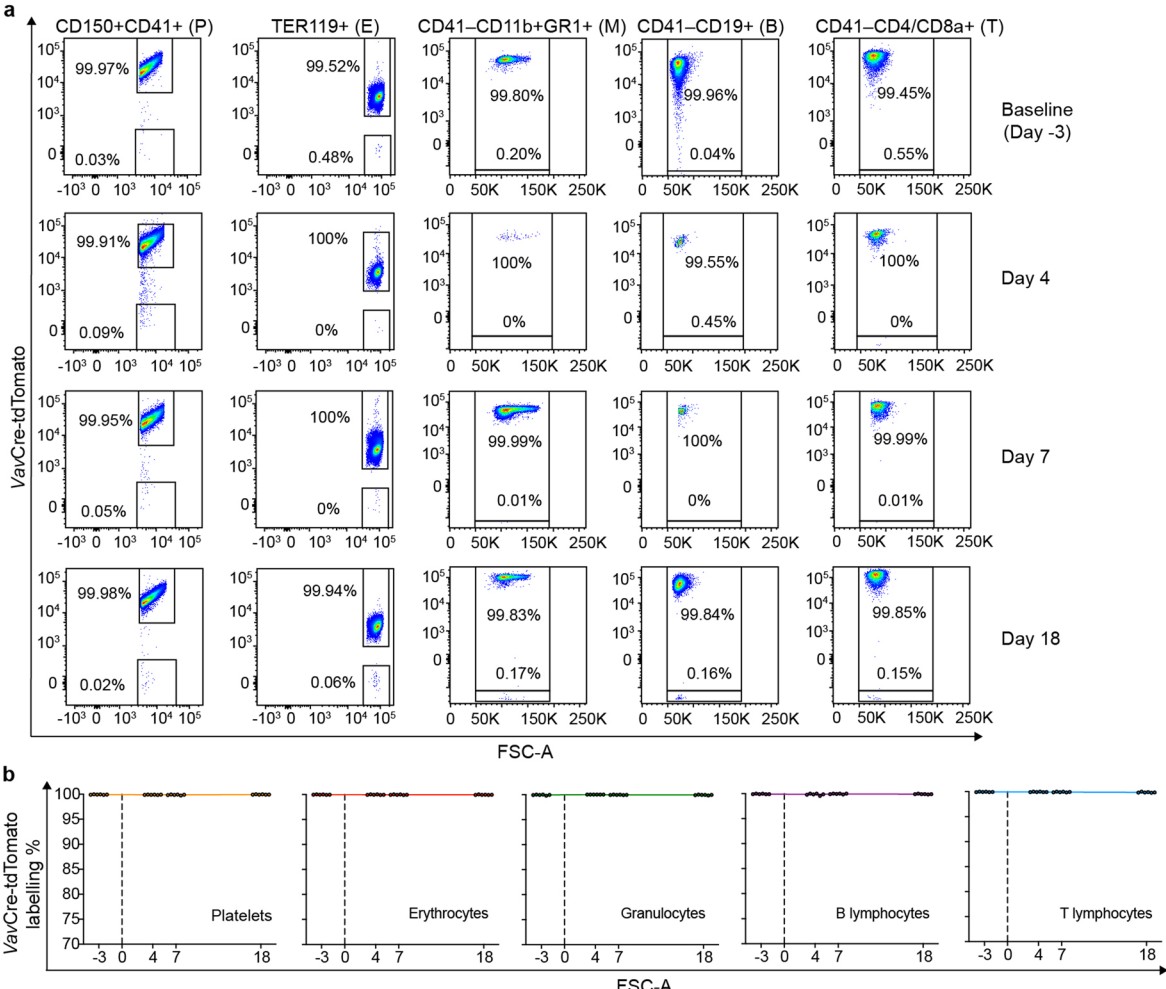

**Extended Data Fig. 9 | Blood lineage analysis of *Vav*Cre^tg/+ *R26*^Tom/+ mice before and after Cyclophosphamide treatment. a**, Representative flow cytometry profiles (% of parent gates) of blood lineages of a 23 week old *Vav*Cre^tg/+ *R26*^Tom/+ mouse before CP and on days 4 and 45 after CP. P: platelets, E: erythrocytes,

M: myeloid (granulocytic) cells, B: B lymphoid cells, T: T lymphoid cells.
**b**, *Vav*Cre-tdTomato labelling of blood lineages in 8–23 week old *Vav*Cre^tg/+ *R26*^Tom/+ mice before and after CP (n = 6, same mice for all time points). Baseline on day -3 and CP injection on day 0. Lines connect the mean of each time point.

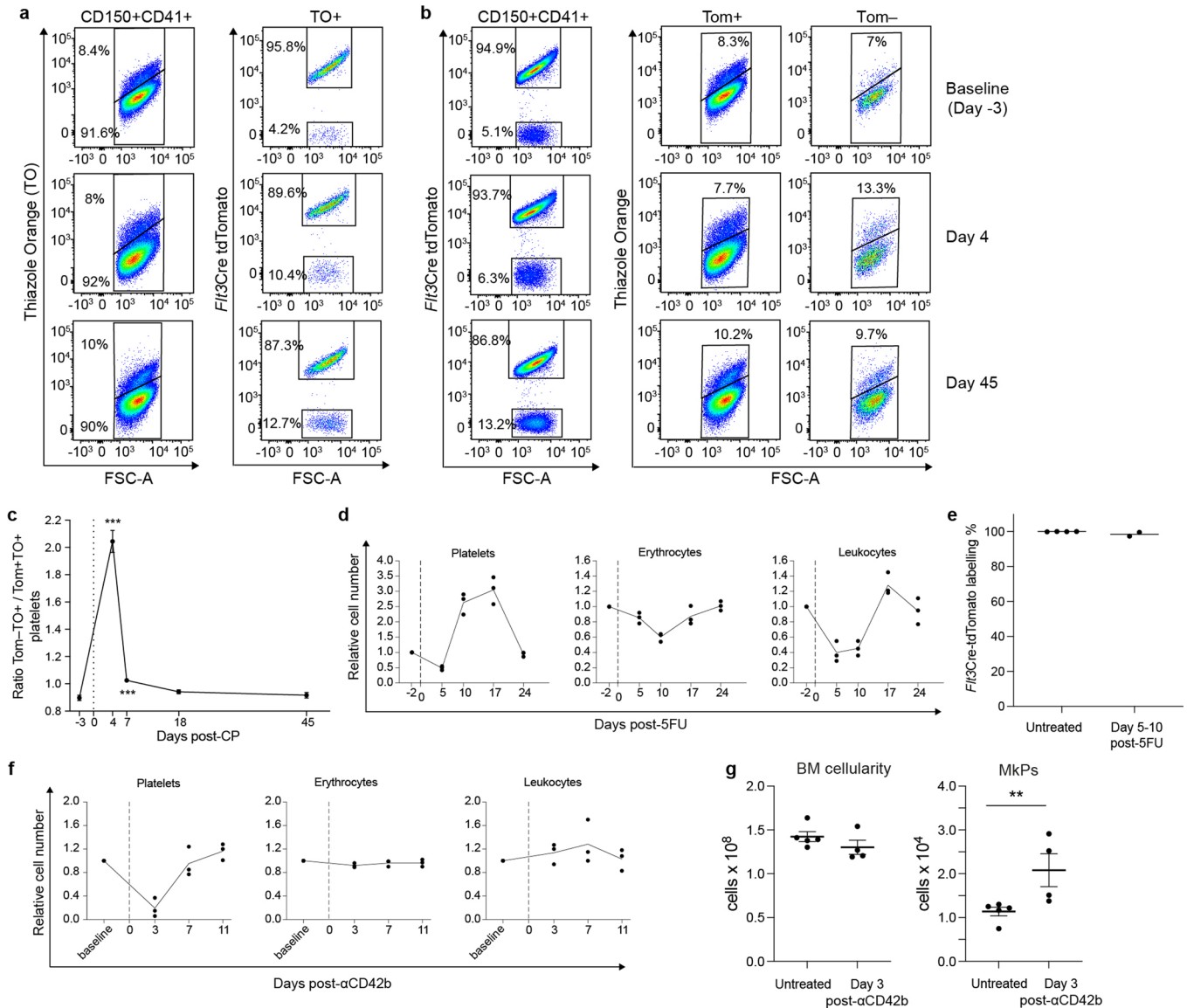

**Extended Data Fig. 10 | Staining of blood platelets with Thiazole Orange after Cyclophosphamide treatment, and fate mapping of *Flt3*Cre^tg/+ *R26*^Tom/+ mice after 5FU and antibody-induced platelet depletion. a**, Representative flow cytometry profiles (% of parent gates) of Tomato labelling in Thiazole Orange positive (TO⁺) platelets of a *Flt3*Cre^tg/+ *R26*^Tom/+ mouse before CP (upper panels), and on day 4 (middle panels) and 45 (lower panels) after CP. **b**, Representative flow cytometry profile (% of parent gates) of TO labelling in Tom⁺ and Tom⁻ platelets of a *Flt3*Cre^tg/+ *R26*^Tom/+ mouse before and after CP. **c**, Ratio (mean ± s.e.m.) between TO⁺ Tom⁻ and TO⁺ Tom⁺ platelets in *Flt3*Cre^tg/+ *R26*^Tom/+ mice before and after CP (day 0). Analysis at baseline (day -3, n = 16), and on day 4 (n = 10), 7 (n = 10), 18 (n = 10), and 45 (n = 7). Compared to baseline, ****$P = 3.77 \times 10^{-7}$ for d4 and ****$P = 3.90 \times 10^{-5}$ for d7. Two-tailed Mann-Whitney test. **d**, Blood cell counts, relative to baseline (d-2), of 8–10 week old *Flt3*Cre^tg/+ *R26*^Tom/+ mice before and after 5-Fluorouracil (5FU, n = 3) treatment. Lines

connect the means for each time point. **e**, *Flt3*Cre-tdTomato labelling of BM LIN⁻FLT3⁺ progenitor cells in *Flt3*Cre^tg/+ *R26*^Tom/+ mice, untreated (n = 4) and 5–10 days after 5FU treatment (n = 2). Dots represent individual mice and lines represent means. **f**, Blood cell counts, relative to baseline of 12–13 week old *Flt3*Cre^tg/+ *R26*^Tom/+ mice (n = 3), each analyzed before treatment (baseline; d-10 or -3) and on day 3, 7 and 11 after treatment with anti-CD42b antibody (αCD42b). Lines connect the means for each time point. **g**, BM cellularity (left) and BM LK CD150⁺CD41⁺ MkP (right) cell number recovered after crushing both femurs, tibiaes, and pelvic bones of 12–13 week old untreated (n = 5) and αCD42b-treated (n = 4) mice. Dots represent individual mice and lines represent mean ± s.e.m. BM cellularity was not significantly different, whereas BM MkP cell number was significantly increased after αCD42b treatment (**$P = 1.59 \times 10^{-2}$); two-tailed Mann-Whitney test.

# Reporting Summary

## Statistics

For all statistical analyses, confirm that the following items are present in the figure legend, table legend, main text, or Methods section.

| n/a | Confirmed | |
|---|---|---|
| ☐ | ☒ | The exact sample size (*n*) for each experimental group/condition, given as a discrete number and unit of measurement |
| ☐ | ☒ | A statement on whether measurements were taken from distinct samples or whether the same sample was measured repeatedly |
| ☐ | ☒ | The statistical test(s) used AND whether they are one- or two-sided *Only common tests should be described solely by name; describe more complex techniques in the Methods section.* |
| ☐ | ☒ | A description of all covariates tested |
| ☐ | ☒ | A description of any assumptions or corrections, such as tests of normality and adjustment for multiple comparisons |
| ☐ | ☒ | A full description of the statistical parameters including central tendency (e.g. means) or other basic estimates (e.g. regression coefficient) AND variation (e.g. standard deviation) or associated estimates of uncertainty (e.g. confidence intervals) |
| ☐ | ☒ | For null hypothesis testing, the test statistic (e.g. *F*, *t*, *r*) with confidence intervals, effect sizes, degrees of freedom and *P* value noted *Give P values as exact values whenever suitable.* |
| ☒ | ☐ | For Bayesian analysis, information on the choice of priors and Markov chain Monte Carlo settings |
| ☒ | ☐ | For hierarchical and complex designs, identification of the appropriate level for tests and full reporting of outcomes |
| ☐ | ☒ | Estimates of effect sizes (e.g. Cohen's *d*, Pearson's *r*), indicating how they were calculated |

*Our web collection on statistics for biologists contains articles on many of the points above.*

## Software and code

Policy information about availability of computer code

| Data collection | BD FACSDiva version 9.0 software was used to acquire flow cytometry data and to perform cell sorting. |
|---|---|
| Data analysis | BD FlowJo version 10.8.1 software was used to gate and analyse flow cytometry data. |

Microsoft Excel for Mac version 16.65 was used for data compilation and simple calculations (percentages, means, etc).

GraphPad Prism version 9.4.1 software was used to generate graphs and to perform statistical analysis, along with the online tool https://www.graphpad.com/quickcalcs/). Additional statistical analysis was also performed in R (version 4.1.1) using the following packages: lme4 version 1.1-30 and emmeans version 1.8.1-1.

Analysis of RNA sequencing data and integration with index-sort FACS data was performed in R (version 4.1.1) and GSEA software version 4.3.3. The code has been deposited to a public repository and is available on Zenondo under digital object identifier (DOI) 10.5281/zenodo.10925564 and include the following packages: abind (1.4-5), annotate (1.80.0), AnnotationDbi (1.64.1), askpass (1.2.0), assertthat (0.2.1), AUCell (1.24.0), backports (1.4.1), base64enc (0.1-3), batchelor (1.18.1), beachmat (2.18.0), beeswarm (0.4.0), BH (1.84.0-0), biglm (0.9-2.1), Biobase (2.62.0), BiocGenerics (0.48.1), BiocManager (1.30.22), BiocNeighbors (1.20.2), BiocParallel (1.36.0), BiocSingular (1.18.0), BiocVersion (3.18.1), Biostrings (2.70.1), bit (4.0.5), bit64 (4.0.5), bitops (1.0-7), blob (1.2.4), bluster (1.12.0), brio (1.1.4), broom (1.0.5), bslib (0.6.1), cachem (1.0.8), Cairo (1.6-2), callr (3.7.3), car (3.1-2), carData (3.0-5), caTools (1.18.2), cellranger (1.1.0), classInt (0.4-10), cli (3.6.2), clipr (0.8.0), colorspace (2.1-0), commonmark (1.9.0), conflicted (1.2.0), corrplot (0.92), cowplot (1.1.3), cpp11 (0.4.7), crayon (1.5.2), crosstalk (1.2.1), curl (5.2.0), cytolib (2.14.1), CytoML (2.14.0), data.table (1.14.10), DBI (1.1.3), dbplyr (2.4.0), DelayedArray (0.28.0), DelayedMatrixStats (1.24.0), deldir (2.0-2), desc (1.4.3), diffobj (0.3.5), digest (0.6.33), dotCall64 (1.1-1), dplyr (1.1.4), dqrng (0.3.2), dtplyr (1.3.1), e1071 (1.7-14), edgeR (4.0.6), ellipsis (0.3.2), evaluate (0.23), fansi (1.0.6), farver (2.1.1), fastDummies (1.7.3), fastmap (1.1.1),

fitdistrplus (1.1-11), flowClust (3.40.0), flowCore (2.14.1), flowViz (1.66.0), flowWorkspace (4.14.2), FNN (1.1.3.2), fontawesome (0.5.2), forcats (1.0.0), formatR (1.14), fs (1.6.3), furrr (0.3.1), futile.logger (1.4.3), futile.options (1.0.1), future (1.33.1), future.apply (1.11.1), gargle (1.5.2), generics (0.1.3), GENIE3 (1.24.0), GenomeInfoDb (1.38.5), GenomeInfoDbData (1.2.11), GenomicRanges (1.54.1), ggbeeswarm (0.7.2), ggcyto (1.30.0), ggplot2 (3.4.4), ggpointdensity (0.1.0), ggprism (1.0.4), ggrastr (1.0.2), ggrepel (0.9.5), ggridges (0.5.6), ggthemes (5.0.0), globals (0.16.2), glue (1.6.2), goftest (1.2-3), googledrive (2.1.1), googlesheets4 (1.1.1), gplots (3.1.3.1), graph (1.80.0), gridExtra (2.3), grr (0.9.5), GSEABase (1.64.0), gtable (0.3.4), gtools (3.9.5), haven (2.5.4), HDF5Array (1.30.0), here (1.0.1), hexbin (1.28.3), highr (0.10), hms (1.1.3), htmltools (0.5.7), htmlwidgets (1.6.4), httpuv (1.6.13), httr (1.4.7), ica (1.0-3), IDPmisc (1.1.21), ids (1.0.1), igraph (1.6.0), interp (1.1-6), IRanges (2.36.0), irlba (2.3.5.1), isoband (0.2.7), jpeg (0.1-10), jquerylib (0.1.4), jsonlite (1.8.8), KEGGREST (1.42.0), kernlab (0.9-32), knitr (1.45), labeling (0.4.3), lambda.r (1.2.4), later (1.3.2), latticeExtra (0.6-30), lazyeval (0.2.2), leiden (0.4.3.1), leidenbase (0.1.27), lifecycle (1.0.4), limma (3.58.1), listenv (0.9.0), lme4 (1.1-35.1), lmtest (0.9-40), locfit (1.5-9.8), lubridate (1.9.3), magrittr (2.0.3), Matrix (1.6-5), MatrixGenerics (1.14.0), MatrixModels (0.5-3), matrixStats (1.2.0), memoise (2.0.1), metapod (1.10.1), mime (0.12), miniUI (0.1.1.1), minqa (1.2.6), mixtools (2.0.0), modelr (0.1.11), munsell (0.5.0), ncdfFlow (2.48.0), nloptr (2.0.3), numDeriv (2016.8-1.1), openCyto (2.14.0), openssl (2.1.1), openxlsx (4.2.5.2), parallelly (1.36.0), patchwork (1.2.0), pbapply (1.7-2), pbkrtest (0.5.2), pbmcapply (1.5.1), pheatmap (1.0.12), pillar (1.9.0), pkgbuild (1.4.3), pkgconfig (2.0.3), pkgload (1.3.3), plogr (0.2.0), plotly (4.10.3), plyr (1.8.9), png (0.1-8), polyclip (1.10-6), praise (1.0.0), prettyunits (1.2.0), princurve (2.1.6), processx (3.8.3), progress (1.2.3), progressr (0.14.0), promises (1.2.1), proxy (0.4-27), ps (1.7.5), pscl (1.5.5.1), purrr (1.0.2), quantreg (5.97), R.methodsS3 (1.8.2), R.oo (1.25.0), R.utils (2.12.3), R6 (2.5.1), ragg (1.2.7), randomForest (4.7-1.1), RANN (2.6.1), rappdirs (0.3.3), RBGL (1.78.0), RColorBrewer (1.1-3), Rcpp (1.0.12), RcppAnnoy (0.0.21), RcppArmadillo (0.12.8.0.0), RcppEigen (0.3.3.9.4), RcppHNSW (0.5.0), RcppML (0.3.7), RcppProgress (0.4.2), RcppTOML (0.2.2), RCurl (1.98-1.14), readr (2.1.4), readxl (1.4.3), rematch (2.0.0), rematch2 (2.1.2), remotes (2.4.2.1), reprex (2.0.2), reshape2 (1.4.4), ResidualMatrix (1.12.0), reticulate (1.34.0), Rgraphviz (2.46.0), rhdf5 (2.46.1), rhdf5filters (1.14.1), Rhdf5lib (1.24.1), RhpcBLASctl (0.23-42), rlang (1.1.2), rmarkdown (2.25), ROCR (1.0-11), rprojroot (2.0.4), RProtoBufLib (2.14.0), rsample (1.2.0), RSpectra (0.16-1), RSQLite (2.3.4), rstatix (0.7.2), rstudioapi (0.15.0), rsvd (1.0.5), Rtsne (0.17), rvest (1.0.3), s2 (1.1.6), S4Arrays (1.2.0), S4Vectors (0.40.2), sass (0.4.8), ScaledMatrix (1.10.0), scales (1.3.0), scater (1.30.1), scattermore (1.2), scran (1.30.0), sctransform (0.4.1), scuttle (1.12.0), segmented (2.0-1), selectr (0.4-2), Seurat (5.0.1), SeuratObject (5.0.1), sf (1.0-15), shiny (1.8.0), SingleCellExperiment (1.24.0), sitmo (2.0.2), slam (0.1-50), slider (0.3.1), slingshot (2.10.0), snow (0.4-4), sourcetools (0.1.7-1), sp (2.1-2), spam (2.10-0), SparseArray (1.2.3), SparseM (1.81), sparseMatrixStats (1.14.0), spatstat.data (3.0-4), spatstat.explore (3.2-6), spatstat.geom (3.2-8), spatstat.random (3.2-2), spatstat.sparse (3.0-3), spatstat.utils (3.0-4), spData (2.3.0), spdep (1.3-1), speedglm (0.3-5), statmod (1.5.0), stringi (1.8.3), stringr (1.5.1), SummarizedExperiment (1.32.0), sys (3.4.2), systemfonts (1.0.5), tensor (1.5), terra (1.7-65), testthat (3.2.1), textshaping (0.3.7), tibble (3.2.1), tidyr (1.3.0), tidyselect (1.2.0), tidyverse (2.0.0), timechange (0.2.0), tinytex (0.49), tradeSeq (1.16.0), TrajectoryUtils (1.10.0), tzdb (0.4.0), units (0.8-5), utf8 (1.2.4), uuid (1.1-1), uwot (0.1.16), vctrs (0.6.5), vipor (0.4.7), viridis (0.6.4), viridisLite (0.4.2), vroom (1.6.5), waldo (0.5.2), warp (0.2.1), withr (2.5.2), wk (0.9.1), xfun (0.41), XML (3.99-0.16), xml2 (1.3.6), xtable (1.8-4), XVector (0.42.0), yaml (2.3.8), zip (2.3.0), zlibbioc (1.48.0), zoo (1.8-12), base (4.3.2), boot (1.3-28.1), class (7.3-22), cluster (2.1.4), codetools (0.2-19), compiler (4.3.2), datasets (4.3.2), foreign (0.8-85), graphics (4.3.2), grDevices (4.3.2), grid (4.3.2), KernSmooth (2.23-22), lattice (0.21-9), MASS (7.3-60), Matrix (1.6-1.1), methods (4.3.2), mgcv (1.9-0), nlme (3.1-163), nnet (7.3-19), parallel (4.3.2), rpart (4.1.21), spatial (7.3-17), splines (4.3.2), stats (4.3.2), stats4 (4.3.2), survival (3.5-7), tcltk (4.3.2), tools (4.3.2), utils (4.3.2).

For manuscripts utilizing custom algorithms or software that are central to the research but not yet described in published literature, software must be made available to editors and reviewers. We strongly encourage code deposition in a community repository (e.g. GitHub). See the Nature Portfolio guidelines for submitting code & software for further information.

# Data

Policy information about availability of data

All manuscripts must include a data availability statement. This statement should provide the following information, where applicable:
- Accession codes, unique identifiers, or web links for publicly available datasets
- A description of any restrictions on data availability
- For clinical datasets or third party data, please ensure that the statement adheres to our policy

Data availability
Source data for all figures related to FACS analysis and in vitro lineage potentials is available in the online version of the paper. Additional relevant information and material will be available from the corresponding authors upon request (j.carrelha@imperial.ac.uk / sten.eirik.jacobsen@ki.se). RNA sequencing data have been deposited to the public repository ArrayExpress under accession number E-MTAB-13935.

Code availability
Code for RNA sequencing analysis and statistical analysis of fate mapping data has been deposited to the public repository Zenodo with digital object identifier (DOI): 10.5281/zenodo.10925564.

# Research involving human participants, their data, or biological material

Policy information about studies with human participants or human data. See also policy information about sex, gender (identity/presentation), and sexual orientation and race, ethnicity and racism.

| Reporting on sex and gender | N/A |
| --- | --- |
| Reporting on race, ethnicity, or other socially relevant groupings | N/A |
| Population characteristics | N/A |
| Recruitment | N/A |
| Ethics oversight | N/A |

# Field-specific reporting

Please select the one below that is the best fit for your research. If you are not sure, read the appropriate sections before making your selection.

☒ Life sciences        ☐ Behavioural & social sciences        ☐ Ecological, evolutionary & environmental sciences

For a reference copy of the document with all sections, see nature.com/documents/nr-reporting-summary-flat.pdf

# Life sciences study design

All studies must disclose on these points even when the disclosure is negative.

| | |
|---|---|
| Sample size | Based on our previous knowledge with the experimental setup, the required number of single HSC transplanted mice was estimated with the goal of generating enough reconstituted mice with each of the reconstitution categories of interest to be used in multiple downstream experiments. Being P-HSC ~5-10% of Vwf-tdTomato+ (Carrelha et al. 2018) and Multi-HSC ~ 70% Vwf-tdTomato- (see ED Fig.1), we single cell transplanted equal numbers of Vwf-dTom+ and Vwf-dTom- HSCs in each cohort of 40-60 mice to obtain beetween 2-10 mice per experimental cohort. RNA sequencing: sample sizes were not predetermined using statistical analysis, but cell type comparisons or methods comparisons typically contained hundreds of cells per group. One 384 well plate was collected per mouse. At least 7 mice were included per reconstitution category/group for experiments, and at least 3 mice when validating results from RNA sequencing. Considering the preserved patterns of gene expression for cells within individual biological replicates and between biological replicates with the same reconstitution pattern, this sample size was considered sufficient for the study. As sample size was not predetermined for other experiments, a similar strategy as above was applied when considering a sufficient sample size. |
| Data exclusions | Blood lineage reconstitution patterns upon single HSC transplantation were defined as described in detail in Results and Methods. We included for further analysis all mice reconstituted with single cells fulfilling the described definition of being Vwf+ P-restricted, Vwf+ P-HSCs, or Vwf- Multi-HSCs without P, PE, or PEM bias.<br>In secondary transplantations, primary donors that failed to reconstitute secondary recipients with a positive control population were excluded from further analysis (i.e., lack of output from a population of interest was not considered robust unless accompanied by positive output from the positive control population).<br>In order not to include mice with incomplete recombination in fate mapping experiments, Flt3Cre and VavCre mice with <98% of reporter labelling in erythroid cells, myeloid cells, B cells and T cells and/or for Flt3Cre mice if <98% of reporter labelling inFLT3 positive bone marrow progenitor cells were excluded from analysis (even if possessing the genotype of interest).<br>Single-cell RNA-seq data were filtered according to established quality control criteria for removing technically failed cells. Cutoffs are listed where appropriate.<br>For anti-CD42b experiments mice, Sysmex analysis at 3 days post injection were used to validate the platelet depletion. When Sysmex analysis could not validate the platelet depletion due to, for example, clot formation during blood collection or suboptimal intravenous injections, mice were excluded from downstream analysis. |
| Replication | Each type of experiment was repeated at least twice (platelet depletion, CP treatment in VavCre mice) or more (34 replicates for single cell transplantations), by multiple investigators, across multiple dates, and using mice from multiple litters. All experimental replicates showed very little inter-experiment variability. All RNA sequencing experiments were performed across hundreds of individual cells and using several mice transplanted both at Karolinska Institute and at University of Oxford. |
| Randomization | For in vivo single cell transplantation experiment randomization was not required as we are unable to predict the reconstitution pattern from a single HSC prior to transplantation. For single cell sorting for transplantation and for RNA sequencing, random individual cells of each population were FACS sorted into wells of microplates according to a predefined sort layout. For treatment with CP, 5FU and CD42b antibody the mice were randomly allocated to PBS and treatment with agents. |
| Blinding | For in vivo single cell transplantation experiment blinding was not required as we are unable to predict the reconstitution pattern from a single HSC prior to transplantation. With the exception chemotherapy treatment (CP and 5FU) where the mice had to be allocated to different cages during treatment in order to avoid unspecific effects in untreated cage-mates, the investigators were blinded to condition or treatment by the use of generic mouse ID numbers which did not reveal the experimental group. |

# Reporting for specific materials, systems and methods

We require information from authors about some types of materials, experimental systems and methods used in many studies. Here, indicate whether each material, system or method listed is relevant to your study. If you are not sure if a list item applies to your research, read the appropriate section before selecting a response.

## Materials & experimental systems

| n/a | Involved in the study |
|---|---|
| ☐ | ☒ Antibodies |
| ☒ | ☐ Eukaryotic cell lines |
| ☒ | ☐ Palaeontology and archaeology |
| ☐ | ☒ Animals and other organisms |
| ☒ | ☐ Clinical data |
| ☒ | ☐ Dual use research of concern |
| ☒ | ☐ Plants |

## Methods

| n/a | Involved in the study |
|---|---|
| ☒ | ☐ ChIP-seq |
| ☐ | ☒ Flow cytometry |
| ☒ | ☐ MRI-based neuroimaging |

## Antibodies

| Antibodies used | Antibody details in Supplementary Table 6 where dilutions, catalog number and lot number are listed for each antibody used in the study. |
|---|---|
| Validation | The antibodies used have been previously validated in the mouse haematopoietic system for the same applications as in this study. In addition, we validated all antibodies (and each batch of each antibody) by titration with relevant cells, using staining panels that included antibodies for negative and positive control antigens. |

## Animals and other research organisms

Policy information about studies involving animals; ARRIVE guidelines recommended for reporting animal research, and Sex and Gender in Research

| Laboratory animals | Vwf-tdTomatotg/+ Gata1-eGFPtg/+ mice, C57BL/6OlaHsd and C57BL/6JrJ background, 7-14 weeks old.<br>Flt3Cretg/+ R26Tom/+(Ai9) Vwf-eGFPtg/+ Gata1-eGFPtg/+ mice, C57BL/6OlaHsd and C57BL/6JrJ background, 7-14 weeks old.<br>Flt3Cretg/+ R26Tom/+(Ai14) mice, C57BL/6JrJ background, 7-11 weeks old.<br>VavCretg/+ R26Tom/+(Ai14) mice, C57BL/6JrJ background, 8-23 weeks.<br>Wildtype CD45.1 B6.SJL-Ptprca Pepcb/BoyJ and B6.SJL-Ptprca Pepcb/BoyCrl, 7-16 weeks old.<br>Mice were housed in individually ventilated cages at the Oxford JR facility, with 12/12h light/dark cycle, at 19-24 °C, and humidity 45-65% and at the Karolinska Institute KM facilities, with 12/12h light/dark cycle, 22 ± 1°C and 50% humidity. |
|---|---|
| Wild animals | No wild animal were used in the study |
| Reporting on sex | In the Flt3Cre mouse model, the Cre+ genotype is restricted to males and, therefore, only males were used. Donor mice, recipient mice and competitor cells were sex-matched in transplantation experiments (i.e. all females or all males). |
| Field-collected samples | No field-collection samples were used in the study. |
| Ethics oversight | Animal experiments performed at University of Oxford were approved by the Oxford Clinical Medicine Ethical Review Committee, and at the Karolinska Institutet by the regional review committee for animal ethics; Stockholms djurförsöksetiska nämnd. All experimental procedures and mouse breeding were performed in accordance with UK Home Office regulations and Swedish Jordbruksverket regulations. |

Note that full information on the approval of the study protocol must also be provided in the manuscript.

## Plants

| Seed stocks | N/A |
|---|---|
| Novel plant genotypes | N/A |
| Authentication | N/A |

# Flow Cytometry

## Plots

Confirm that:

☒ The axis labels state the marker and fluorochrome used (e.g. CD4-FITC).

☒ The axis scales are clearly visible. Include numbers along axes only for bottom left plot of group (a 'group' is an analysis of identical markers).

☒ All plots are contour plots with outliers or pseudocolor plots.

☒ A numerical value for number of cells or percentage (with statistics) is provided.

## Methodology

| | |
|---|---|
| Sample preparation | Leg, pelvis, sternum, and spine bones were collected immediately after culling mice and crushed with pestle and mortar. Blood samples were collected from live mice by tail vein bleeding into Lithium-Heparin tubes, or by cardiac puncture immediately after culling mice. Bone marrow and peripheral blood were prepared into single cell suspension in PBS supplemented with 1-5% fetal calf serum and 2 mM EDTA. All samples were incubated with purified CD16/32 (Fc-Block) prior to staining with monoclonal antibodies, unless the analysis used conjugated CD16/32 for analysis of myeloid progenitor populations. Details in Methods. |
| Instrument | BD FACSAriaII, BD FACSAriaIII, BD FACSAria Fusion, BD LSRII, BD LSR Fortessa, BD LSR Fortessa X-20. |
| Software | BD FACSDiva version 9.0 software was used to acquire flow cytometry data and to perform cell sorting. BD FlowJo version 10.8.1 software was used to gate and analyse flow cytometry data. |
| Cell population abundance | At the beginning and end of sorting sessions, a test sort and immediate purity analysis in the same sorter was performed for the population of interest (or from a parent gate for rare populations). Typically 100-300 cells were sorted and purity upon reanalysis was ≥95% percent when factoring in all impurities in all hierarchical gates. Additional considerations in single cell sort experiments: index sorting recorded the cell surface expression of markers in each single cell sorted; accurate single cell deposition into plates was validated using 488 nm fluorescent beads before and after sorting sessions. |
| Gating strategy | FSC-A/SSC-A was used for gating mononuclear cells. SSC-H/SSC-W and/or FSC-A/FSC-H were used to exclude doublets and select singlets. DAPI-positive or 7AAD-positive cells were gated out to exclude non-viable cells. cKIT/LIN was used for gating out Lineage-positive cells in bone marrow, in order to focus analysis on early progenitors. Whenever possible, the phenotypic definition of each population included both positive and negative markers. Fluorescence-minus-one controls (FMOs) were recorded but, as much as possible, the boundaries of negative and positive populations were set based on internal negative control populations within each sample. Specific gating strategies are outlined in Methods and exemplified in Figures and Extended Data Figures. |

☒ Tick this box to confirm that a figure exemplifying the gating strategy is provided in the Supplementary Information.

