## [Peer Review File · Nature Immunology]

Peer Review Information

Journal: Nature Immunology

Manuscript Title: Alternative platelet differentiation pathways initiated by non-hierarchically related hematopoietic stem cells

Corresponding author name(s): Professor Sten Eirik Jacobsen, Dr Joana Carrelha

Reviewer Comments & Decisions:

Decision Letter, initial version:
--

16th Jun 2023

Dear Professor Jacobsen,

Thank you for sending your proposed response to reviewers comments on your article "Alternative platelet differentiation pathways initiated by distinct hematopoietic stem cells". We would be interested in considering a revised version as you outlined.

We hope you will find the referees' comments useful as you decide how to proceed. If you wish to submit a substantially revised manuscript, please bear in mind that we will be reluctant to approach the referees again in the absence of major revisions.

If you choose to revise your manuscript taking into account all reviewer and editor comments, please highlight all changes in the manuscript text file [OPTIONAL: in Microsoft Word format].

* If you have not done so already please begin to revise your manuscript so that it conforms to our Article format instructions at <http://www.nature.com/ni/authors/index.html>. Refer also to any guidelines provided in this letter.

The Reporting Summary can be found here:

When submitting the revised version of your manuscript, please pay close attention to our [href="https://www.nature.com/nature-portfolio/editorial-policies/image-integrity">Digital Image Integrity Guidelines](https://www.nature.com/nature-portfolio/editorial-policies/image-integrity). and to the following points below:

[REDACTED]

If you wish to submit a suitably revised manuscript we would hope to receive it within 6 months. If you cannot send it within this time, please let us know. We will be happy to consider your revision so long as nothing similar has been accepted for publication at Nature Immunology or published elsewhere.

Nature Immunology is committed to improving transparency in authorship. As part of our efforts in this direction, we are now requesting that all authors identified as 'corresponding author' on published papers create and link their Open Researcher and Contributor Identifier (ORCID) with their account on the Manuscript Tracking System (MTS), prior to acceptance. ORCID helps the scientific community achieve unambiguous attribution of all scholarly contributions. You can create and link your ORCID from the home page of the MTS by clicking on 'Modify my Springer Nature account'. For more information please visit www.springernature.com/orcid.

Thank you for the opportunity to review your work.

Sincerely,

Stephanie Houston
Editor

Nature Immunology

Reviewers' Comments:

Reviewer #1:

Remarks to the Author:

Alternative platelet differentiation pathways initiated by distinct hematopoietic stem cells

The manuscript submitted by Carrelha et al., is a follow up study of the previously proposed Vwf-expressing platelet biased HSCs (Nature, 2013). In the current manuscript, they further elucidated the non-hierarchical relationship between two types of stem cells: those responsible for replenishing all blood cell lineages and those primarily involved in platelet production at single cell resolution. They further showed that these distinct type of stem cells employ different pathways for platelet replenishment: a slower multipotent pathway during steady-state conditions; and a fast-track pathway specifically activated during emergencies in a platelet specific manner. These findings offer insights for improving platelet replenishment in clinical situations with slow recovery.

Overall, it's an interesting follow-up study. One concern is that the observations in this manuscript heavily rely on previous studies by the same group, thereby continuing the discussion and exhibiting limited novel impact. Another concern is the rationale for emergence haematopoiesis and modes of stressors in this study.

Specific comments are listed below.

Major concerns

1. Observations in Figure 2e is confusing, can the authors explain how could single case observation lead to convincing conclusion. More such cases need to be observed in order to draw conclusion. It should be noted that single case may be a result of random impurity in the cells isolation/sorting procedure.
2. Since the authors propose distinct MkP subsets were initiated based on the absence of CD48 expression, they should isolate CD48 positive and negative fractions of the MkPs and perform phenotypic and functional assays to verify.
3. The transition from steady state haematopoiesis to stress insult is a bit abrupt, simply under the assumption that platelet replenishment via shorter pathway is associated with responses to stress. Can the authors show preliminary observation of this association or show reference?
4. Ideally, in order to make the conclusion of emergency haematopoiesis more convincing, the authors should perform single cell analysis of the reconstituted HSPCs under stress and compare with steady state haematopoiesis results.
5. The authors investigated only one mode of stress condition. Other than CTX treatment, stress such as bleeding can also influence platelet production and replenishment.
6. Authors indicated that mTORC1 signalling was the among the pathways most enriched in MoIO-defined HSCs replenished by Vwf-tdTomato positive P-HSCs. Hence mTORC1 inhibition in vivo might be interesting to validate the observations of Vwf-tdTomato positive P-HSCs differentiation and their functions.

Minor concerns

1. The manuscript utilizes multiple terms such as P, PE, or PEM-bias, alongside the predominantly used category "P-HSCs." This mixture of terms can create confusion. It would be helpful for the author

- to provide clarification regarding the need for differentiating these terms within the manuscript. How different are their characteristics and functions?
2. Figure 3d and 4d volcano plots needs more proper labelling on the graph, to show the details of up- and down-regulation.
 3. In the discussion session, it is advisable to incorporate the figure number when summarizing the main points of the manuscript.
 4. Genes selected to represent Figure 4g are noisy, particularly *Aldh1a1*, *Car2* and *Tbxa2r*.

Reviewer #2:

Remarks to the Author:

In this study, the authors focus on a subset of HSC's which express *Vwf*, isolated using a *Vwf*-tdTomato reporter (P-HSC). These cells were found to be biased to reconstitute platelet, erythroid, and myeloid lineages relative to the *Vwf*- HSCs. *Vwf*- cells could not reconstitute *Vwf*+ (tdTomato+) cells, indicating that there is a hierarchical restriction in lineage specification to the platelet lineage. Next, scRNA-seq with index sorting for critical surface markers was performed on transplanted *Vwf*-tdTomato+ and *Vwf*-tdTomato- cells. The *Vwf*+ cells almost exclusively led to megakaryocyte production, while *Vwf*- cells made GMP, MkP, and CFU-E populations. The authors then focused on the MkP trajectory, excluding GMP and Erythroid cells and found dynamic genes along a pseudo-time trajectory. This is an exciting analysis that pointed to specifically differentially expressed genes between cells in the *Vwf*+ trajectory vs. the *Vwf*- trajectory. *Flt3* and *CD48* in particular are markers that a specific between the transitions, and could immediately followed with functional validations. The authors then elucidated some of the gene expression differences between MkPs derived from *Vwf*+ and *Vwf*- trajectories. Cyclophosphamide was used to perturb hematopoiesis and look for changes in platelet production, revealing that *Vwf*+ P-HSC play an important role in maintaining platelet numbers. Although platelet biased HSC populations have previously been identified in the literature, the elucidation of the gene expression dynamics between the *Vwf*+ and *Vwf*- differentiation trajectories is a valuable step in our understanding of hematopoiesis. This paper would be significantly improved if the below concerns are answered:

Major Concerns

- The authors utilize a *Gata1*-eGFP reporter to look at *Vwf*-tdTomato expression but not explicitly state for what reason the *Gata1* signal is cut (Figure 2). This information should be stated in the main text. The authors should also explicitly state the importance of *CD201* for the identification of stem cell subsets and explain why it is used in this context. There should be a lot more detail in the text about our current understanding of *Gata1* as a marker gene and its relevance in the studied populations.
- The authors use Myeloid (M) cells for a label of cells characterized by *NK1.1*-*CD4*/*CD8a*-*CD19*-*CD11b*+*CD45.1*-*CD45.2*+. However, this label is confusing. Traditionally, myeloid refers to the branch of hematopoietic cells that are not lymphoid and erythroid and megakaryocytes are thought to come from the common myeloid progenitor. What cell types are in this Myeloid label? Can the authors validate this label with further profiling using additional flow cytometry markers or cytopspins with microscopy?
- In some experiments within the paper, the P-HSC seems to give rise to platelet cells uniquely (Figure 1A, 5A/B), while in other experiments (Ext. Figure 2B, Figure 2D/E), these cells have some erythroid

and myeloid potential. The authors should carry out a high cell number experiment that firmly establishes which cell populations the Vwf+ HSC cells can generate and use more of the main text to explain these discrepancies. Of particular interest, the authors should try to find out if the MkP cells from the Vwf+ HSC population go through a bipotent Erythroid/Megakaryocyte state, or if they are truly restricted to platelets. There has been evidence of alternative platelet trajectories directly from HSC populations (See PMID PMC6443046), so the authors should further clarify the state of our understanding of platelet biased stem cells from the literature in the manuscript.

Minor Concerns

- In the first paragraph of the results, the authors state that they had previously shown a population of HSCs expresses Vwf. The authors should explicitly state what markers were used to classify these cells as HSCs.
- Figure 1B, the gating scheme (LIN-SCA1+cKIT+(LSK)CD45.1-CD45.2+ gate) prior to the HSC/MPP gates presented here should be explicitly stated in the Figure itself, not just in the figure legend to improve clarity
- Figure 2B-E, the timepoint for the capture of these cells should be explicitly stated in the Figure itself to improve readability.
- The authors should state how they are identifying P, E, M, B, and T cell types in Figure 2 legend or figure
- A graphical summary for which cells get transplanted, what cells are used for secondary transplantation, and what time points the flow cytometry captures are profiled would greatly improve the ability to interpret this figure
- Adding better titles to the flow cytometry plots and gates for the targeted populations would also help the reader understand the results better. For example, the authors should label the CD201 by Vwf gates in Figure 2E with specific names that they can refer to in the text.
- For the final statement in the first results section:
" Thus, Vwf-tdTomato- Multi-HSCs and Vwf-tdTomato+ P-HSCs are not hierarchically related and replenish platelets through distinct pathways."
The authors should add something like Vwf-tdTomato- cells are incapable of producing Vwf+ cells, and thus they must not be hierarchically related.
- For Figure 3, again I would suggest the authors should explain the reason for using Gata1 reporter mice
- Perhaps the authors could use a statistical test to show that "Single cells replenished by Vwf-tdTomato+ P-HSCs and Vwf-tdTomato- Multi-HSCs displayed similar quality control metrics"
- The authors should state the batch effect correction tool used in the main text results section for Figure 3.

- Can the authors comment on Figure 3H? Why do randomly selected genes have such a high correlation along pseudotime (close to 1)? Shouldn't randomly selected genes not be so highly correlated?
- In Figure 5, it would be helpful for the authors to illustrate that the tdTomato expression here is reporting the Flt3-cre activity and not the Vwf activity (which is on GFP now).
- The authors should state how each of the P, E, M, B, and T subsets are identified in Figure 5, either in the legend or figure itself.

Author Rebuttal to Initial comments

See Inserted PDF

Alternative platelet differentiation pathways initiated by non-hierarchically related hematopoietic stem cells

RESPONSES TO REVIEWERS. All revisions have been underlined in the revised manuscript.

Reviewer #1 (All revisions have been underlined in the revised manuscript)

(Remarks to the Author)

General Comment: Alternative platelet differentiation pathways initiated by distinct hematopoietic stem cells. The manuscript submitted by Carrelha et al., is a follow up study of the previously proposed Vwf-expressing platelet biased HSCs (Nature, 2013). In the current manuscript, they further elucidated the non-hierarchical relationship between two types of stem cells: those responsible for replenishing all blood cell lineages and those primarily involved in platelet production at single cell resolution. They further showed that these distinct type of stem cells employ different pathways for platelet replenishment: a slower multipotent pathway during steady-state conditions; and a fast-track pathway specifically activated during emergencies in a platelet specific manner. These findings offer insights for improving platelet replenishment in clinical situations with slow recovery. Overall, it's an interesting follow-up study. One concern is that the observations in this manuscript heavily rely on previous studies by the same group, thereby continuing the discussion and exhibiting limited novel impact.

GENERAL RESPONSE: We highly appreciate that this reviewer highlights that we in these studies show a non-hierarchical relationship between platelet-restricted/biased Vwf+ P-HSCs and Vwf- Multilineage HSCs (Multi-HSCs), and that s/he acknowledges that we “*showed that these distinct type of stem cells employ different pathways for platelet replenishment: a slower multipotent pathway during steady-state conditions; and a fast-track pathway specifically activated during emergencies in a platelet specific manner*”. These are obviously key and novel findings in the manuscript, as previous studies have only shown hierarchical relationships between different HSCs, which is most compatible with using shared rather than distinct pathways for lineage replenishment. With regard to the novelty of our findings, we have as outlined below in the revised manuscript more clearly introduced and discussed the novel findings presented in this manuscript in light of previous studies from us and other research groups. We thank the reviewer for also raising a number of relevant points/suggestions which we have addressed as outlined in our specific point-by-point responses below, and which have resulted in extensive revisions of the manuscript, including inclusion of extensive new and novel data, translating into what we think is a considerably improved manuscript.

Major concerns

1. Observations in Figure 2e is confusing, can the authors explain how could single case observation lead to convincing conclusion. More such cases need to be observed in order to draw conclusion. It should be noted that single case may be a result of random impurity in the cells isolation/sorting procedure.

RESPONSE: We understand why the original Fig. 2e appeared confusing to the reviewer since it is a single case. However, from a conceptual/functional point of view with regard to the hierarchical relationship between Vwf+ P-HSCs and Vwf- Multi-HSCs being addressed in Fig. 2, the results/conclusions from the single case in original Fig. 2e (now revised **Extended Data Fig. 1f**) are, from a functional point of view, identical to those in the original Fig. 2d (now revised **Fig. 2e**), with regard to the hierarchical question addressed, namely whether Vwf+ P-HSCs can give rise to Vwf- Multi-HSCs. As shown in the figure, neither the Vwf+ or Vwf- cells replenished in the primary recipient of the single Vwf+ P-HSC that were sorted and transplanted into secondary recipients gave rise to any multilineage reconstitution, as evidenced

by no replenishment of B or T lymphocytes. On the contrary, in this case, the Vwf⁻ cells like the Vwf⁺ cells only gave rise to highly platelet-biased reconstitution in the secondary recipients, overlapping with what was observed in the primary recipient (revised **Extended Data Fig. 1f**). Therefore, just as in the cases included in revised **Fig. 2e**, Vwf⁺ P-HSCs cannot replenish Multi-HSCs. The only distinction between this “unique” case (revised **Extended Data Fig. 1f**) and the others (revised **Fig. 2e**) was therefore that some of the P-HSCs replenished by the single Vwf⁺ P-HSC obviously had lost their expression of *Vwf*-tdTomato, perhaps reflecting a silencing of the reporter. We have revised the Results text (pages 8-9) to better describe the results and interpretations of this interesting case.

Highly relevant for this comment of this reviewer, the other reviewer requested the inclusion of a graphical summary in Fig. 2, to make the experimental strategy and steps clearer and easier to follow. We agree that such a graphical summary will aid the readers in understanding the design and experimental stages for the important hierarchical experiments in revised **Fig. 2** (and revised **Extended Data Fig. 1f**) and have therefore included this in the revised manuscript as new **Fig. 2a**. Moreover, we have also introduced new “summary panels” to illustrate how the results were interpreted regarding (non-) hierarchical replenishment of Multi-HSCs and P-HSCs in revised **Fig. 2c-e** and **Extended Data Fig. 1f**.

2. Since the authors propose distinct MkP subsets were initiated based on the absence of CD48 expression, they should isolate CD48 positive and negative fractions of the MkPs and perform phenotypic and functional assays to verify.

RESPONSE: We have pursued several lines of investigation to address the suggestion to characterize the distinct MkPs replenished by the P-HSC and Multi-HSC pathways in further detail, including but not restricted to comparison of CD48 positive and negative fractions of the MkPs. When requested by the editor (before she made a decision to invite a revised manuscript) to outline how we would plan to revise the manuscript within the given timeline in response to the different points raised, we argued for pursuing further phenotypic analysis as suggested by the reviewer as that (as outlined below) could help identify cell surface antigens that in future studies could be used to identify, track and further characterize MkPs replenished through the distinct pathways initiated by Vwf⁺ P-HSCs and Vwf⁻ Multi-HSCs, in other mouse models than the transgenic *Vwf*-tdTomato and *Flt3*Cre-tdTomato mice used in our studies. Related to that goal, but also to better characterize the molecular differences between the distinct MkPs replenished by the two pathways, we also suggested that we together with our expert collaborators on single cell RNA sequencing would aim to pursue further single cell RNA sequencing analysis comparing the MkPs replenished by the two pathways. We therefore argued, also in light of the many other revisions requested, to focus the revision experiments on these two related lines of investigations, as they were likely to be the most informative with regard to providing further insights into the distinction between the MkPs replenished by these two pathways, and more so than what a functional analysis of CD48⁺ and CD48⁻ MkPs realistically could have achieved within the 6-month deadline for resubmission of a revised manuscript. The editor responded that she would be interested in receiving a revised manuscript revised as we had outlined. Further functional studies will obviously be important in the future, but the most interesting and informative functional studies would be to genetically *in vivo* fate map the MkPs replenished by the two pathways, but currently we have no genetic mouse models that would make that possible in the given time frame. As for the limited number of MkPs replenished by a single HSC, it is unlikely these would be sufficient for meaningful *in vivo* transplantations, and the value of *in vitro* experiments would be much more limited, and in light of that we prioritized the relevant and available mice for the further molecular (including phenotypic) analysis of MkPs replenished from the two pathways.

The extended analysis of single cell RNA sequencing of MkPs replenished by the two HSC pathways has provided a number of interesting additional insights with regard to differentially regulated gene expression and pathways, resulting in a considerable expansion with regard to the description of the RNA sequencing results as described on pages 12-15 of the revised Results and displayed in new and revised **Fig. 4a-b, 4d, 5a-c, 6a-d, and Extended Data Fig. 6b**.

Of particular interest, to identify differentially expressed genes that might facilitate future identification and enrichment of MkPs distinct for the two differentiation pathways, genes encoding cell surface antigens were specifically explored in the list of DEGs (revised **Fig. 4a**; **Supplementary Table 4**). A significantly differential upregulation of transcriptional expression of *Cd24a* (similar to *Vwf*, driving the expression of *Vwf*-tdTomato) was observed in molecularly defined P-MkPs whereas *Cd48* and *Itga2* (encoding CD49b protein) were upregulated in Multi-MkPs and negative in almost all P-MkPs (new **Fig. 5a**). Flow cytometric index information confirmed the differential expression of CD48 protein and *Vwf*-tdTomato on MkPs from the two pathways (revised **Extended Data Fig. 6b**), which was further validated in separate experiments together with CD24 and CD49b protein expression. In agreement with the transcriptional data, *Vwf*-tdTomato and cell surface CD24 expression were distinctly upregulated in P-MkPs (new **Fig. 5b**) whereas CD48 and CD49b expression was detected on most Multi-MkPs but virtually absent from P-MkPs (new **Fig. 5c**). Our findings are in agreement with previous studies suggesting that CD48 expression might define a distinct subset of MkPs (Prins et al., *Science Advances*, 2020, reference #13; Morcos et al., *Nature Communications*, 2022, reference #14). We found P-MkPs to be uniformly CD48^{-low} at the transcriptional and protein level (new **Fig. 5**), but also a significant fraction of Multi-MkP was negative for *Cd48* and CD48 (new **Fig. 5**). While this was compatible with Multi-HSCs in part replenishing CD48⁻ MkPs that might overlap with CD48⁻ P-MkPs, this was not the case as CD48⁺ and CD48⁻ Multi-MkPs showed highly overlapping DEGs when individually compared with CD48⁻ P-MkPs, including for *Cd24a*, *Itga2* and *Vwf* (new **Fig. 6a-b**; new **Supplementary Table 5**). The same pattern of DEGs was also observed doing the same comparison based on *Cd48* mRNA expression (new **Fig. 6c**; new **Supplementary Table 5**), whereas very few DEGs were detected when comparing Multi-MkPs negative or positive for *Cd48* transcript (new **Fig. 6d**; new **Supplementary Table 5**). Thus, while these findings (described in revised Results page 14) demonstrate that *Cd48*/CD48 expression specifically identifies Multi-MkPs, also the fraction of *Cd48*/CD48⁻ Multi-MkPs are molecularly distinct from P-MkPs. On page 21 of the revised Discussion we highlight how the single cell RNA sequencing of P-MkPs and Multi-MkPs provided important insights into differentially expressed genes encoding cell surface antigens and how this should facilitate identification and further characterization of P-MkPs and Multi-MkPs also in wild type mice. Taken together, the extended single cell RNA sequencing analyses of HSPCs replenished by single transplanted *Vwf*-tdTomato⁺ P-HSCs and *Vwf*-tdTomato⁻ Multi-HSCs unravel molecularly distinct progenitor differentiation trajectories for platelet replenishment, including the replenishment of transcriptionally and phenotypically distinct MkPs.

3. The transition from steady state haematopoiesis to stress insult is a bit abrupt, simply under the assumption that platelet replenishment via shorter pathway is associated with responses to stress. Can the authors show preliminary observation of this association or show reference?

RESPONSE: We agree that the transition from steady state to stress insult in the manuscript could be better justified, and we therefore provide a clearer rationale for this in the revised manuscript, including some new data that we have analyzed from a recent relevant single cell RNA sequencing manuscript (Kucinski et al., *Cell Stem Cell*, 2024, reference #49). As a result, we have now in the revised manuscript included the following new paragraph (page 16), as well as a new **Supplementary Fig. 3**, as part of the rationale for and transition to the study of stress insults:

*“Our present and previous findings (Carrelha et al., Nature, 2018, reference #9) are compatible with *Vwf*-tdTomato⁺ P-HSCs replenishing MkPs through less progenitor intermediates than *Vwf*-tdTomato⁻ Multi-HSCs, and in agreement with this *Gata1*⁺ progenitors produced from transplanted *Vwf*⁺ P-HSCs replenish platelets with faster kinetics than *Gata1*⁺ progenitors from *Vwf*⁺ Multi-HSCs HSCs (Meng et al., Nature Cell Biology, 2023, reference #48). To investigate whether this also translates into faster steady-state kinetics of MkP replenishment through the P-HSC than Multi-HSC progenitor pathway, we explored recently published single cell RNA sequencing data, in which the kinetics of progenitor replenishment were*

assessed after recombination induction in *Hoxb5Cre^{ERT2/+} R26^{Tom/+}* reporter mice, specifically labelling the HSC compartment (Kucinski et al., *Cell Stem Cell*, 2024, reference #49). Interestingly, a subset of MkPs were the first lineage-restricted progenitors to be replenished by the labeled HSCs (Kucinski et al., *Cell Stem Cell*, 2024, reference #49), and when compared to MkPs replenished at later time points this early wave of MkPs showed a significant upregulation of genes up-regulated in P-MkPs and down-regulation of genes upregulated in Multi-MkPs (**Supplementary Fig. 3**). Collectively, these findings raise the possibility that upon insults to the hematopoietic system resulting in loss of MkPs, utilization of the P-HSC pathway might more rapidly replenish platelets than Multi-HSCs, and if so that the P-HSC pathways might play a more important role in response to hematopoietic insults than in steady-state. To explore this, we treated *Flt3Cre^{tg/+} R26^{Tom/+}* mice with Cyclophosphamide (CP)...”). Our analysis of the Kucinski et al. data in relation to the single cell RNA sequencing data of P-MkPs and Multi-MkPs in our manuscript is detailed on page 45 of the Methods in the revised manuscript.

4. Ideally, in order to make the conclusion of emergency haematopoiesis more convincing, the authors should perform single cell analysis of the reconstituted HSPCs under stress and compare with steady state haematopoiesis results.

RESPONSE: We are glad that the reviewer wrote that we “*ideally*” should perform such experiments (“*single cell analysis of the reconstituted HSPCs under stress*”), as these experiments (not totally unexpectedly) turned out to be difficult to perform. We tried to pursue the proposed experiments with the goal of performing additional single cell RNA sequencing analysis of progenitors reconstituted by single Vwf+ P-HSCs and Vwf- Multi-HSCs after the challenge with cyclophosphamide. However, since this involved the application (for the first time) of 2 consecutive procedures (single HSC transplantation of lethally irradiated mice followed by cyclophosphamide treatment) each resulting in significant bone marrow suppressive effects, this resulted in serious side effects so that the mice had to be killed prior to the planned analysis, and also these side effects would most likely have affected the results. We also believe that the interpretation of the RNA sequencing results anyhow would have been further complicated by the treatment since it would most likely result in extensive changes in RNA expression largely related to the general effects of cyclophosphamide rather than a specific effect on activation of stem and progenitor cells in response to the platelet depletion and stress resulting from this treatment. Therefore, while we were willing to make a new and probably alternative attempt to transcriptionally analyze and compare progenitors replenished by Vwf+ P-HSCs and Vwf- Multi-HSCs following cyclophosphamide treatment, this would require an additional extension of 5-6 months (as it takes 4 months for transplanted single HSCs to give long-term multilineage reconstitution) before we could resubmit a revised manuscript, and in light of this we consulted the editor, who responded that the editors could “*appreciate that this experiment was technically not feasible*” and in light of that she recommended that we should simply explain this to the reviewer rather than attempting additional experiments. While such mechanistic studies (as for point 6) obviously are of considerable interest and important to pursue to also further unravel the regulatory cues governing the herein identified distinct pathways for platelet replenishment, in steady-state as well as in response to challenges, such studies will be extensive in themselves, and therefore we believe also beyond the reasonable scope of this study/manuscript, also in light of the maximum 6 month timeline given for revisions. The main novelty of the findings reported in our revised manuscript, further enhanced through the new revisions in response to the points raised by the 2 reviewers (including inclusion of several new lines of data), is the first definitive evidence for alternative differentiation trajectories for the replenishment of MkPs and platelets (or any short-lived blood cell lineage) initiated by functionally distinct and non-hierarchically related HSCs; a slower multilineage HSC pathway prevailing in steady-state, and a fast-track platelet-restricted HSC pathway bypassing many of the progenitor stages utilized by the multilineage pathway and that shows enhanced activity upon elimination of megakaryocyte progenitors with chemotherapy. Therefore, as outlined in response to other points raised by this and the other reviewer, we have in the limited time frame given, focused on pursuing and including new experiments with regard to

the molecular (single cell RNA sequencing and protein) analysis of the distinct MkPs replenished by the P-HSC and Multi-HSC initiated pathways, as well as also suggested by this reviewer, new lines of data on the effect of additional platelet challenges on the replenishment of platelets through these alternative HSC initiated pathways for platelet replenishment.

5. The authors investigated only one mode of stress condition. Other than CTX treatment, stress such as bleeding can also influence platelet production and replenishment.

RESPONSE: We thank the reviewer for suggesting the investigation of additional stress conditions. In the revised manuscript we now include results from 2 additional challenges. First we administered 5-Fluorouracil (5FU), another commonly used myeloablative agent shown to rapidly reduce MkPs in the BM, and in line with this we observed a transient decrease in blood platelets (new **Extended Data Fig. 10d**) and a significant reduction in *Flt3Cre*-tdTomato⁺ platelets (from 95% to 58%) on day 10 post-5FU (New **Fig. 7f**). Notably, in response to 5FU, we also observed a smaller yet significant decrease in *Flt3Cre*-tdTomato⁺ fractions of erythrocytes and myeloid cells (but not lymphocytes; new **Fig. 7f**), probably reflecting that P-HSCs also can replenish lower levels of erythrocytes and myeloid cells.

We also wanted to test a challenge that specifically depletes platelets rather than the upstream progenitors, as we would postulate that following such a challenge the rapid platelet replenishment could primarily be accomplished by already existing progenitors (from both pathways) rather than from HSCs, and if so the contribution of the two pathways could be expected to be largely unaltered. We therefore induced acute platelet depletion by administration of a single dose of the anti-CD42b antibody (α CD42b) to *Flt3Cre*^{tg/+} *R26*^{Tom/+} mice. As previously reported, acute thrombocytopenia was observed 3 days post- α CD42b (new **Fig. 7g**), with no impact on other blood lineages (new **Extended Data Fig. 10f**), nor on the balance between *Flt3Cre*-tdTomato⁺ and *Flt3Cre*-tdTomato⁻ platelets (new **Fig. 7h**). Unlike the loss of MkPs in response to CP treatment (**Extended Data Fig. 8b**), an expansion of MkPs was observed 3 days after platelet depletion (new **Extended Data Fig. 10g**), suggesting that a rapid expansion of existing MkPs underly the subsequent platelet recovery. We believe these new data add considerably to the manuscript, as combined, these findings suggest that a rapid and transient increase in platelet replenishment can be achieved through the P-HSC pathway in response to challenges that reduce progenitors of the Mk lineage.

These new results are described on pages 17-18 of the revised Results, discussed on page 20 of the revised Discussion, and detailed in the Methods on pages 33 and 46-47.

6. Authors indicated that mTORC1 signalling was the among the pathways most enriched in MoLO-defined HSCs replenished by Vwf-tdTomato positive P-HSCs. Hence mTORC1 inhibition in vivo might be interesting to validate the observations of Vwf-tdTomato positive P-HSCs differentiation and their functions.

RESPONSE: While we certainly agree that it would be interesting to validate the functional role of mTORC1 in regulation of P-HSC function and platelet replenishment, we believe for the same arguments as those raised under point 4, that due to the extensive and challenging nature of such experiments and their interpretation (like mTORC1 affecting many other processes that could directly or indirectly influence platelet replenishment through other mechanisms), such and other mechanistic studies would be better suited for separate follow up studies. Therefore, when requested by the editor (before making a decision to invite a revised manuscript) to outline how we would plan to revise the manuscript within the given timeline in response to all the points raised by the two reviewers, we for this specific point argued for not pursuing such experiments, as we for the reasons listed above felt such experiments (although interesting) were beyond the reasonable scope of a revised manuscript, also in light of the many other lines of revisions and experiments requested by this and the other reviewer within the given 6 month deadline for resubmission of a revised manuscript. We had also noticed that the reviewer did not seem to require such experiments

since stating that it “*might be interesting*” to pursue such experiments. In light of this we were pleased that the editor responded “*We would be interested in considering a revised version as you outlined*”, and therefore we are not including data along addressing this point in the revised manuscript.

Minor concerns

1. The manuscript utilizes multiple terms such as P, PE, or PEM-bias, alongside the predominantly used category "P-HSCs." This mixture of terms can create confusion. It would be helpful for the author to provide clarification regarding the need for differentiating these terms within the manuscript. How different are their characteristics and functions?

RESPONSE: We agree with the reviewer that this can be hard to follow, and for that reason we have tried to clarify and as much as possible also simplify this in the revised manuscript. In fact, a related point was brought up by the other reviewer, who wanted us to clarify the distribution of lineage replenishment patterns by transplanted single Vwf+ HSC, which has actually been described in detail in our referenced 2018 publication (Carrelha et al., Nature, 2018, reference #9) in which >1,000 mice were transplanted with a single BM LSKCD34⁻CD150⁺CD48⁻ HSC. In response to the other reviewer, we summarized in the first paragraph of the Results of the revised manuscript (pages 5-6), the main findings from those studies with regard to the lineage reconstitution patterns observed from transplanted single Vwf+ HSCs; namely that a large fraction of these do not (at any time) contribute to B or T lymphocytes, and of these many replenish blood in a platelet (P)-biased manner, including fraction that are P-restricted, replenishing exclusively platelets in the first four months after transplantation into primary recipients. However, as we have also shown in the same publication, P-restricted HSCs can in some instances also replenish low levels of the erythroid and myeloid (granulocyte/monocyte) lineages, following long-term reconstitution in primary recipients or when transplanted into secondary recipients (Carrelha et al., Nature, 2018, reference #9). Nevertheless, they always remain highly P-biased and fail to contribute to the B and T lymphoid lineages (Carrelha et al., Nature, 2018, reference #9), suggesting that P-bias is a stable and HSC-intrinsic property, independently of whether low levels of myeloid or erythroid contribution also is observed. In contrast, most single Vwf- HSCs replenish all lympho-myeloid blood lineages upon transplantation (Multi-HSC), and typically in a lineage-balanced or lymphoid-biased rather than P-biased manner (**Extended Data Fig. 1b-c**). Therefore, for scientific reasons and also to simplify, Vwf+ P-biased HSCs (P-HSCs) were in the present studies (as described on Results page 6 and Methods page 36) defined as single HSCs which upon transplantation stably (at multiple analysis time points) contribute ≥ 50 -fold more to platelets than to both erythrocytes and myeloid cells, and with little or no ($\leq 0.01\%$) contribution to B and T lymphocytes, whereas Vwf- Multi-HSCs were defined as HSCs that replenish all blood cell lineages in a balanced or lymphoid-biased pattern. In the revised manuscript, we quantify different lineage-biased and lineage-restricted patterns, in a similar way to our previous publication (Carrelha et al., Nature, 2018, reference #9) in revised **Extended Data Fig. 1b**, as this is the first time this was explored in detail for Vwf- HSCs. That the definition of P-HSCs in the current manuscript encompasses a distinct and quite homogenous subpopulation of HSCs is highlighted by the very stable and reproducible findings with regard to replenishment of a cellular and molecular distinct progenitor pathway for replenishment of MkPs and platelets.

2. Figure 3d and 4d volcano plots needs more proper labelling on the graph, to show the details of up- and down-regulation.

RESPONSE: We thank the reviewer for this suggestion as we agree that these plots as originally presented were not so easy to follow. We have therefore in the revised manuscript revised the presentation of the two volcano plots (revised **Fig. 3g** and **Fig. 4a**) to make it simpler to comprehend what the plots show. Although not specifically requested we have also incorporated similar labelling to the new volcano plots included in revised/new **Fig. 6**.

3. In the discussion session, it is advisable to incorporate the figure number when summarizing the main points of the manuscript.

RESPONSE: While we agree that this could be helpful for the readers, the editor specifically advised us not to do this as it is against the journal formatting guidelines.

4. Genes selected to represent Figure 4g are noisy, particularly *Aldh1a1*, *Car2* and *Tbxa2r*.

RESPONSE: Single cell RNA expression analysis is associated with noise due to the number of amplification cycles required to detect the low RNA quantity from single cells, as well as dropouts resulting in false negative signal for detected genes (Vallejos et al., Nature Methods, 2017). Importantly, our data do not show more variation than similar data from other studies something we have illustrated in the figure below (in interest of space we are including this figure only for the information of the reviewer and not in the revised manuscript) by comparing the CV for *Aldh1a1*, *Clu*, *Tbxa2r* as well as the housekeeping gene *Actb* within molecularly defined HSCs and MkPs identified in our single cells and the corresponding gene expression data in molecularly defined HSCs and MkPs from a recently published single cell data set (Kucinski et al., Cell Stem Cell, 2024). Furthermore, the statistical analysis we have utilized for identification of differentially expressed genes accounts for the overdispersion associated with gene counts obtained from single cell RNA sequencing. Even though computational algorithms to impute or denoise data have been developed, we have after consultation with the single cell RNA sequencing experts co-authoring this manuscript (and since the reviewer anyhow did not request this) decided not to perform this on our dataset as they emphasize that such methods are currently not accepted as standard practice in the field. In light of the consistent findings among biological replicates, we have instead chosen to present the log2 normalized counts in the plots. We have updated the Methods (page 42, first paragraph) to point out that no denoising imputations were applied to the data presented in the manuscript.

Figure. Combined violin and box plots showing the distribution in gene expression for indicated genes within molecularly defined HSCs and MkPs replenished from both P-HSCs and Multi-HSC mice (pooled data of MkPs from both HSC types) based on single cell RNA sequencing data in our manuscript (selected genes also presented in **Fig. 4e** in the revised manuscript) and corresponding data for molecularly defined HSCs and MkPs from a recently published gene expression data set (Kucinski et al., Cell Stem Cell, 2024). Boxes show first and third quartiles of the normalized expression values, where the line within each box indicates the median and whiskers indicate the largest value within the $\pm 1.5 \cdot \text{IQR}$. Numbers below each gene name indicate the coefficient of variance (%) for each gene.

Reviewer #2 (All revisions have been underlined in the revised manuscript)

(Remarks to the Author)

In this study, the authors focus on a subset of HSC's which express Vwf, isolated using a Vwf-tdTomato reporter (P-HSC). These cells were found to be biased to reconstitute platelet, erythroid, and myeloid lineages relative to the Vwf- HSCs. Vwf- cells could not reconstitute Vwf+ (tdTomato+) cells, indicating that there is a hierarchical restriction in lineage specification to the platelet lineage. Next, scRNA-seq with index sorting for critical surface markers was performed on transplanted Vwf-tdTomato+ and Vwf-tdTomato- cells. The Vwf+ cells almost exclusively led to megakaryocyte production, while Vwf- cells made GMP, MkP, and CFU-E populations. The authors then focused on the MkP trajectory, excluding GMP and Erythroid cells and found dynamic genes along a pseudo-time trajectory. This is an exciting analysis that pointed to specifically differentially expressed genes between cells in the Vwf+ trajectory vs. the Vwf- trajectory. Flt3 and CD48 in particular are markers that a specific between the transitions, and could immediately followed with functional validations. The authors then elucidated some of the gene expression differences between MkPs derived from Vwf+ and Vwf- trajectories. Cyclophosphamide was used to perturb hematopoiesis and look for changes in platelet production, revealing that Vwf+ P-HSC play an important role in maintaining platelet numbers. Although platelet biased HSC populations have previously been identified in the literature, the elucidation of the gene expression dynamics between the Vwf+ and Vwf- differentiation trajectories is a valuable step in our understanding of hematopoiesis. This paper would be significantly improved if the below concerns are answered:

RESPONSE: We appreciate that the reviewer highlights that the findings are “*exciting*” and the value of our new findings towards an enhanced “*understanding of hematopoiesis*”. We also appreciate the many points raised by this reviewer to enhance the impact and clarity of the findings in the manuscript. We have addressed all the specific points raised by this reviewer in the revised manuscript, as outlined in detail in our point-by-point response below. We believe that these revisions (underlined in the revised manuscript), which include the presentation of new and interesting data (in response to this as well as the other reviewer), have considerably improved the clarity as well as novelty of the revised manuscript.

Major Concerns

1) The authors utilize a Gata1-eGFP reporter to look at Vwf-tdTomato expression but not explicitly state for what reason the Gata1 signal is cut (Figure 2). This information should be stated in the main text. The authors should also explicitly state the importance of CD201 for the identification of stem cell subsets and explain why it is used in this context. There should be a lot more detail in the text about our current understanding of Gata1 as a marker gene and its relevance in the studied populations

RESPONSE: We have, as suggested, clarified in the Results of the revised manuscript, the rationale for defining phenotypic HSCs as negative for the Gata1-eGFP reporter, reflecting that long-term repopulating HSCs have previously been shown to be negative for the Gata1-eGFP reporter (page 6, last paragraph; Drissen et al., Nature Immunology, 2016, reference #17). We also explain (Results pages 6-7 and Methods pages 35-37), why and how we use the Gata1-eGFP reporter to track platelets and erythrocytes from the transplanted single HSCs, since platelets and mature erythrocytes lack CD45 expression used to track the mature white blood cell lineages, and for the same reason the usage of the Gata1-eGFP reporter to identify erythroid progenitor cells. Likewise, we also clearly state and reference the literature for the rationale for using CD201 to enrich for HSCs (Results page 6, last paragraph; Purton, Experimental Hematology, 2022, reference #18).

2) The authors use Myeloid (M) cells for a label of cells characterized by NK1.1-CD4/CD8a-CD19-CD11b+CD45.1-CD45.2+. However, this label is confusing. Traditionally, myeloid refers to the branch of

hematopoietic cells that are not lymphoid and erythroid and megakaryocytes are thought to come from the common myeloid progenitor. What cell types are in this Myeloid label? Can the authors validate this label with further profiling using additional flow cytometry markers or cytopspins with microscopy?

RESPONSE: We agree that it might appear confusing that myeloid (M) cells in the HSC field typically refer to granulocytes/monocytes whereas erythrocytes and platelets are rather referred to separately. Therefore, to help clarify how the lineage reconstitution analysis was performed in the present manuscript, we have made the following revisions:

-The first time we talk about blood lineage reconstitution (page 6, first paragraph) we define myeloid as reflecting the granulocyte/monocyte lineages; “*myeloid (granulocyte/monocyte)*”

-To further clarify how we track the different blood cell lineages replenished by the transplanted single HSC we now also describe in the Results section (pages 6-7) that “*Donor-derived white blood cells were identified as CD45.2⁺CD45.1⁻, but since platelets, erythrocytes, and late-stage erythroid progenitors have little or no CD45 expression, platelets replenished by transplanted single HSCs were identified as Vwf-tdTomato⁺Gata1-eGFP⁺ and erythroid replenishment as Vwf-tdTomato⁻Gata1-eGFP⁺ (Carrelha et al., Nature, 2018, reference #9; Drissen et al., Nat Immunol, 2016, reference #17)*”. Further details of the analysis of different blood cell lineages are also provided in the revised Methods (pages 35-36).

-One aspect of the phenotypic definition of the different blood cell lineages is that, in addition to antigens that identify each blood lineage as being selectively/specifically expressed on that lineage, to enhance specificity we also defined each lineage as being negative for antigens defining other lineages. To make this clearer, we have in the revised manuscript and figures now consistently written the phenotypic definition of each lineage so that the lineage specific markers positively defining the lineage are listed first followed by the markers for other lineages for which they should be negative. Based on this, the phenotypic definition of the myeloid (granulocyte/monocyte) lineage replenished from the transplanted single HSCs is slightly reorganized to CD11b⁺NK1.1⁻CD19⁻CD4/CD8a⁻CD45.1⁻CD45.2⁺ (Methods page 36), although being defined by the same phenotype.

-Finally, as suggested by the reviewer, we have carefully validated the lineage identity of this definition of myeloid cells by sorting the CD11b⁺NK1.1⁻CD19⁻CD4/CD8a⁻CD45.1⁻CD45.2⁺ blood cells from 2 mice reconstituted by a Vwf⁻ Multi-HSC and 2 mice reconstituted with a Vwf⁺ P-HSC (with low myeloid reconstitution), made cytopspins of these cells, stained them with Eosin-Y/Azure-A/Methylene-Blue and performed differential counts of 100 cells per mouse. These results, demonstrating that cells with the cell surface phenotype of CD11b⁺NK1.1⁻CD19⁻CD4/CD8a⁻CD45.1⁻CD45.2⁺ represent pure myeloid (granulocyte/ monocyte) cells, are described in the revised Methods (page 36, first paragraph) and images of their morphology and differential counts are included in the new **Supplementary Fig. 4**.

3) In some experiments within the paper, the P-HSC seems to give rise to platelet cells uniquely (Figure 1A, 5A/B), while in other experiments (Ext. Figure 2B, Figure 2D/E), these cells have some erythroid and myeloid potential. The authors should carry out a high cell number experiment that firmly establishes which cell populations the Vwf⁺ HSC cells can generate and use more of the main text to explain these discrepancies.

RESPONSE: We thank the reviewer for making this point, as it highlights the importance of clarifying the distribution of lineage replenish patterns by transplanted single Vwf⁺ HSCs. This has been described previously in our referenced 2018 publication (Carrelha et al, Nature 554:106-11, 2018) in which >1,000 mice were transplanted with a single BM LSKCD34⁻CD150⁺CD48⁻ HSC. In the first paragraph of the Results of the revised manuscript (pages 5-6), we summarize the main findings from those studies with regard to the lineage reconstitution patterns observed from transplanted single Vwf⁺ HSCs; namely that a large fraction of these do not (at any time) contribute to B or T lymphocytes, and of these many replenish blood in a P-biased manner, of which a fraction are P-restricted, replenishing exclusively platelets in the

first four months after transplantation into primary recipients (Carrelha et al., Nature, 2018, reference #9). However, P-restricted HSCs can in some instances also replenish low levels of the erythroid and myeloid (granulocyte/monocyte) lineages, following long-term reconstitution in primary recipients or when transplanted into secondary recipients (Carrelha et al., Nature, 2018, reference #9). Nevertheless, they always remain highly P-biased and fail to contribute to the B and T lymphoid lineages (Carrelha et al., Nature, 2018, reference #9), suggesting that P-bias is a stable and HSC-intrinsic property. In contrast, most single *Vwf*-tdTomato⁻ HSCs replenish all lympho-myeloid blood lineages upon transplantation (Multi-HSC), and typically in a lineage-balanced or lymphoid-biased rather than P-biased manner (Carrelha et al., Nature, 2018, reference #9). Therefore, in the present study, *Vwf*-tdTomato⁺ P-biased HSCs (P-HSCs) were, as described (Results page 6, second paragraph, and Methods page 36), defined as single HSCs which upon transplantation stably (at multiple analysis time points) contribute ≥ 50 -fold to platelets than to both erythrocytes and myeloid cells, and with little or no ($\leq 0.01\%$) contribution to B and T lymphocytes.

4) Of particular interest, the authors should try to find out if the MkP cells from the *Vwf*⁺ HSC population go through a bipotent Erythroid/Megakaryocyte state, or if they are truly restricted to platelets.

RESPONSE: We thank the reviewer for this constructive suggestion, which resulted in inclusion of entirely new and clear data demonstrating that neither the MkPs nor any other progenitors replenished by single *Vwf*⁺ P-HSCs go through a bipotent Erythroid/Megakaryocyte state. To address this important point, we have now performed new single cell RNA sequencing data analysis of stem and progenitor cells replenished by *Vwf*⁺ P-HSCs. These new data provide further (in addition to the already included biological data through lineage replenishment analysis *in vivo*) compelling molecular evidence that MkPs replenished from *Vwf*⁺ P-HSCs are truly restricted to the platelet lineage and do not have erythroid potential. In fact, through analysis also of all other progenitor stages replenished by *Vwf*⁺ P-HSCs, we provide evidence that also other progenitor stages derived from *Vwf*⁺ P-HSCs lack E potential, whereas included new (although not specifically requested by the reviewer) corresponding data for *Vwf*⁻ Multi-HSCs show clear evidence for sustained E potential including bipotent progenitors with combined Mk and E potential. These data are shown in new **Fig. 3c-e**, and are described in the revised Results page 10, second paragraph.

5) There has been evidence of alternative platelet trajectories directly from HSC populations (See PMCID PMC6443046), so the authors should further clarify the state of our understanding of platelet biased stem cells from the literature in the manuscript.

RESPONSE: Although restricted by the extensively revised manuscript significantly exceeding the word limit, we have in the revised introduction, tried to more clearly explain the current status in the field with regard to platelet biased HSCs and potential alternative platelet replenishment mechanisms prior to our studies. As we state there, several reported findings have implicated distinct fast-track pathways for replenishment of megakaryocyte-restricted progenitors (MkPs) (Haas et al., Cell Stem Cell, 2015, reference #11; Psaila and Mead, Blood 2019, reference #12; Prins et al., Science Advances, 2020, reference #13; Morcos et al., Nature Communications, 2022, reference #14). However, as all of them rely on phenotypic or molecular rather than functional definitions of HSCs, it remains unclear whether the pathways proposed initiate from true HSCs or down-stream progenitors. Likewise, the identification of platelet (P)-biased and P-restricted HSCs (Sanjuan-Pla et al., Nature 2013, reference #8; Carrelha et al., Nature 2018, reference #9; Rodriguez-Fraticelli et al., Nature, 2018, reference #10), is also compatible with distinct pathways for platelet replenishment pathways. However, to date there has only been evidence provided for hierarchical relationships between HSCs with different lineage biases (Sanjuan-Pla et al., Nature 2013, reference #8; Müller-Sieburg et al., Blood, 2002, reference #15; Dykstra et al., Cell Stem Cell, 2007, reference #16), implicating shared rather than separate pathways for development and replenishment of blood cell lineages downstream of HSCs. Through single cell HSC fate mapping our studies provide the first definitive

evidence for alternative differentiation trajectories for the replenishment of platelets, initiating from functionally distinct and non-hierarchically related HSCs; a slower multilineage HSC pathway prevailing in steady-state, and a fast-track platelet-restricted HSC pathway bypassing many of the progenitor stages utilized by the multilineage pathway, which shows enhanced activity upon challenges resulting in loss of megakaryocyte progenitors.

Minor Concerns

1) In the first paragraph of the results, the authors state that they had previously shown a population of HSCs expresses Vwf. The authors should explicitly state what markers were used to classify these cells as HSCs.

RESPONSE: As requested, we have in the revised manuscript spelled out the cell surface phenotype of the Vwf⁺ HSCs previously studied (page 5, last paragraph). Moreover, in the revised manuscript (page 6, last paragraph) we also emphasize that HSCs can ultimately only be identified/defined through their functional properties (Purton, Experimental Hematology, 2022, reference #18). Therefore, single Vwf⁺ P-HSCs and Vwf⁻ Multi-HSCs were in our studies eventually identified/defined by their stable and distinct blood lineage output as assessed by serial analysis of their replenishment of different blood lineages over time.

2) Figure 1B, the gating scheme (LIN-SCA1+cKIT+(LSK)CD45.1-CD45.2+ gate) prior to the HSC/MPP gates presented here should be explicitly stated in the Figure itself, not just in the figure legend to improve clarity

RESPONSE: For clarity, the gating scheme has been revised/simplified in **Fig. 1b** and corresponding legend, and also in other FACS panels in the manuscript where the same is relevant (**Fig. 2b-e, 3m, 5b-c; Extended Data Fig. 1a, 1f**).

3) Figure 2B-E, the timepoint for the capture of these cells should be explicitly stated in the Figure itself to improve readability.

RESPONSE: As requested, the time points have been added to relevant panels of revised **Fig. 2**.

4) The authors should state how they are identifying P, E, M, B, and T cell types in Figure 2 legend or figure.

RESPONSE: The definitions of the P, E, M, B, and T cell lineages has been added in the legend of **Fig. 2** as requested.

5) A graphical summary for which cells get transplanted, what cells are used for secondary transplantation, and what time points the flow cytometry captures are profiled would greatly improve the ability to interpret this figure.

RESPONSE: We agree that such a graphical summary will aid the readers in understanding the design and experimental stages for the important hierarchical experiments in revised **Fig. 2** (and revised **Extended Data Fig. 1f**) and have therefore included this in the revised manuscript as new **Fig. 2a**. Moreover, we have

also introduced new “summary panels” to illustrate how the results were interpreted regarding (non-) hierarchical replenishment of Multi-HSCs and P-HSCs in revised **Fig. 2c-e** and **Extended Data Fig. 1f**.

6) Adding better titles to the flow cytometry plots and gates for the targeted populations would also help the reader understand the results better. For example, the authors should label the CD201 by Vwf gates in Figure 2E with specific names that they can refer to in the text.

RESPONSE: We have revised the flow cytometry panel in original Fig. 2e (revised **Extended Data Fig. 1f**) and also the labeling of other FACS panels in the manuscript where we also felt this would enhance clarity.

7) For the final statement in the first results section: "Thus, Vwf-tdTomato⁻ Multi-HSCs and Vwf-tdTomato⁺ P-HSCs are not hierarchically related and replenish platelets through distinct pathways." The authors should add something like Vwf-tdTomato⁻ cells are incapable of producing Vwf⁺ cells, and thus they must not be hierarchically related.

RESPONSE: We agree this helps to clarify the interpretation of the hierarchical experiments and have therefore, as suggested, revised the relevant sentence at the end of the first Results section (page 9, first paragraph) to read: *“Together with Vwf-tdTomato⁻ Multi-HSCs being incapable of producing Vwf-tdTomato⁺ P-HSCs, these findings demonstrate that Vwf-tdTomato⁻ Multi-HSCs and Vwf-tdTomato⁺ P-HSCs are not hierarchically related and therefore should replenish platelets through different pathways.”*

8) For Figure 3, again I would suggest the authors should explain the reason for using Gata1 reporter mice.

RESPONSE: As outlined in our response to Major points 1 and 2, we now already in the beginning of the revised Results (pages 6-7), in addition to the Methods (pages 35-37), explain the rationale for the usage of the Gata1-GFP reporter both for HSC sorting and blood replenishment analysis.

9) Perhaps the authors could use a statistical test to show that "Single cells replenished by Vwf-tdTomato⁺ P-HSCs and Vwf-tdTomato⁻ Multi-HSCs displayed similar quality control metrics".

RESPONSE: To address this question, we have in new **Supplementary Fig. 1b** (which we refer to in the revised manuscript page 9, last paragraph) added new data plots which show similar distribution of the quality control metrics for single cells replenished by Vwf-tdTomato⁺ P-HSCs and Vwf-tdTomato⁻ Multi-HSCs, as well as the mean and standard deviation values. These data demonstrate similar mean values (<1.22 fold differences) with overlapping interquartile ranges for these two groups. Furthermore, as described in the Methods (Smart-seq3 single cell library preparations and sequencing; page 40, first paragraph), each plate to be sequenced included the same internal control GMP and LSK cells sorted from a single pool of viably frozen BM MNCs, which we in **Supplementary Fig. 1e** demonstrate display similar clustering both with and without batch correction. These quality controls, together with consistent observations among biological replicates represented by different transplanted mice, allowed the control and correction for potential technical variation between experiments.

10) The authors should state the batch effect correction tool used in the main text results section for Figure 3.

RESPONSE: To address this, we have in the revised manuscript incorporated the following statement to the main text (page 9, last paragraph): “*After adjustment for batch effects from independent sequencing runs using the mutual nearest neighbors approach (Haghverdi et al., Nature Biotechnology, 2018, reference #24) (Supplementary Fig. 1d-g), we performed...*”

In addition, we have now also specified the batch correction method in the revised legend of the corresponding **Supplementary Fig. 1e**: “*Batch effect analysis, using the mutual nearest neighbour approach, of three independent sequencing runs...*”. This is also described in the Methods, page 42, second paragraph.

11) Can the authors comment on Figure 3H? Why do randomly selected genes have such a high correlation along pseudotime (close to 1)? Shouldn't randomly selected genes not be so highly correlated?

RESPONSE: We thank the reviewer for making us aware that we had not explained this clearly in the original manuscript. What was not specifically stated in the original legend for Fig. 3h (now **Fig. 3k** in the revised manuscript), and which we therefore now have clarified in the revised legend is that the data show the Pearson correlation in gene expression along pseudotime when comparing gene expression between cells replenished by Vwf+P-HSCs and Vwf- Multi-HSCs of the top 70 DEGs and 70 randomly selected genes, and therefore as expected the correlation of the randomly selected genes is high (close to 1) between the 2 pathways unlike the top DEGs. We now realize that the original description could be interpreted as if we were comparing expression between the different genes shown, and if so, the reviewer is of course correct that one would expect randomly selected genes not to be so highly correlated. To better clarify this the revised legend for **Fig. 3k** now reads: “*Pearson correlation along pseudotime comparing expression of the top 70 DEGs and 70 randomly selected non-DEGs between cells replenished by Vwf-tdTomato⁺ P-HSCs and Vwf-tdTomato⁻ Multi-HSCs. Shading indicates 95% confidence interval (CI).*”

We have also revised the relevant text in the manuscript (Results page 12, first paragraph): “*Pearson correlation of the 70 top ranked DEGs when comparing expression along pseudotime in cells replenished by Vwf-tdTomato⁺ P-HSCs and Vwf-tdTomato⁻ Multi-HSCs showed a more similar gene expression profile for the cells located at the trajectory start (HSCs) and end (MkPs) than for the intermediate stages (Fig. 3k; Supplementary Table 4). In contrast, 70 randomly selected genes demonstrated consistent and high correlation (close to 1) when comparing cells replenished by Vwf-tdTomato⁺ P-HSCs and Vwf-tdTomato⁻ Multi-HSCs. Taken together, this demonstrates that differential expression of DEGs along pseudotime defines the separation of the two pathways from HSCs to MkPs.*”

12) In Figure 5, it would be helpful for the authors to illustrate that the tdTomato expression here is reporting the Flt3-cre activity and not the Vwf activity (which is on GFP now).

RESPONSE: This has now been revised as suggested in all the relevant panels of original Fig. 5 (now **Fig. 7** in revised manuscript).

13) The authors should state how each of the P, E, M, B, and T subsets are identified in Figure 5, either in the legend or figure itself.

RESPONSE: This has been addressed in the revised legend of original Fig. 5 (now **Fig. 7** in revised manuscript).

Decision Letter, first revision:

13th Mar 2024

Dear Dr. Jacobsen,

Thank you for submitting your revised manuscript "Alternative platelet differentiation pathways initiated by non-hierarchically related hematopoietic stem cells" (NI-A35839A). It has now been seen by the original referees and their comments are below. The reviewers find that the paper has improved in revision, and therefore we'll be happy in principle to publish it in Nature Immunology, pending minor revisions to satisfy the referees' final requests and to comply with our editorial and formatting guidelines.

We will now perform detailed checks on your paper and will send you a checklist detailing our editorial and formatting requirements in about a week. Please do not upload the final materials and make any revisions until you receive this additional information from us.

If you had not uploaded a Word file for the current version of the manuscript, we will need one before beginning the editing process; please email that to immunology@us.nature.com at your earliest convenience.

Thank you again for your interest in Nature Immunology Please do not hesitate to contact me if you have any questions.

Sincerely,

Stephanie Houston, PhD
Senior Editor
Nature Immunology

Reviewer #1 (Remarks to the Author):

NA

Reviewer #2 (Remarks to the Author):

The authors have adequately addressed concerns.

Final Decision Letter:

Dear Dr. Jacobsen,

I am delighted to accept your manuscript entitled "Alternative platelet differentiation pathways

initiated by non-hierarchically related hematopoietic stem cells" for publication in an upcoming issue of Nature Immunology.

Over the next few weeks, your paper will be copyedited to ensure that it conforms to Nature Immunology style. Once your paper is typeset, you will receive an email with a link to choose the appropriate publishing options for your paper and our Author Services team will be in touch regarding any additional information that may be required.

Please note that *Nature Immunology* is a Transformative Journal (TJ). Authors may publish their research with us through the traditional subscription access route or make their paper immediately open access through payment of an article-processing charge (APC). Authors will not be required to make a final decision about access to their article until it has been accepted. Find out more about Transformative Journals.

Your paper will be published online soon after we receive your corrections and will appear in print in the next available issue.

You may wish to make your media relations office aware of your accepted publication, in case they consider it appropriate to organize some internal or external publicity. Once your paper has been scheduled you will receive an email confirming the publication details. This is normally 3-4 working days in advance of publication. If you need additional notice of the date and time of publication,

please let the production team know when you receive the proof of your article to ensure there is sufficient time to coordinate. Further information on our embargo policies can be found here: <https://www.nature.com/authors/policies/embargo.html>

Also, if you have any spectacular or outstanding figures or graphics associated with your manuscript - though not necessarily included with your submission - we'd be delighted to consider them as candidates for our cover. Simply send an electronic version (accompanied by a hard copy) to us with a possible cover caption enclosed.

If you have not already done so, we strongly recommend that you upload the step-by-step protocols used in this manuscript to the Protocol Exchange. Protocol Exchange is an open online resource that allows researchers to share their detailed experimental know-how. All uploaded protocols are made freely available, assigned DOIs for ease of citation and fully searchable through nature.com. Protocols can be linked to any publications in which they are used and will be linked to from your article. You can also establish a dedicated page to collect all your lab Protocols. By uploading your Protocols to Protocol Exchange, you are enabling researchers to more readily reproduce or adapt the methodology you use, as well as increasing the visibility of your protocols and papers. Upload your Protocols at www.nature.com/protocolexchange/. Further information can be found at www.nature.com/protocolexchange/about .

Please note that we encourage the authors to self-archive their manuscript (the accepted version before copy editing) in their institutional repository, and in their funders' archives, six months after publication. Nature Portfolio recognizes the efforts of funding bodies to increase access of the research they fund, and strongly encourages authors to participate in such efforts. For information about our editorial policy, including license agreement and author copyright, please visit www.nature.com/ni/about/ed_policies/index.html

Sincerely,

Stephanie Houston, PhD
Senior Editor
Nature Immunology